# THEORY, ANALYSIS, AND BEST PRACTICES FOR SIGMOID SELF-ATTENTION

**Jason Ramapuram,**[*] **Federico Danieli,**[*] **Eeshan Dhekane,**[*] **Floris Weers,**[*] **Dan Busbridge,**[*]
**Pierre Ablin,**[*] **Tatiana Likhomanenko,**[*] **Jagrit Digani, Zijin Gu, Amitis Shidani, Russ Webb**

Apple

## ABSTRACT

Attention is a key part of the transformer architecture. It is a sequence-to-sequence mapping that transforms each sequence element into a weighted sum of values. The weights are typically obtained as the softmax of dot products between keys and queries. Recent work has explored alternatives to softmax attention in transformers, such as ReLU and sigmoid activations. In this work, we revisit sigmoid attention and conduct an in-depth theoretical and empirical analysis. Theoretically, we prove that transformers with sigmoid attention are universal function approximators and benefit from improved regularity compared to softmax attention. Through detailed empirical analysis, we identify stabilization of large initial attention norms during the early stages of training as a crucial factor for the successful training of models with sigmoid attention, outperforming prior attempts. We also introduce FLASH-SIGMOID, a hardware-aware and memory-efficient implementation of sigmoid attention yielding a *17%* inference kernel speed-up over FLASHATTENTION2 on H100 GPUs [1]. Experiments across language, vision, and speech show that properly normalized sigmoid attention matches the strong performance of softmax attention on a wide range of domains and scales, which previous attempts at sigmoid attention were unable to fully achieve. Our work unifies prior art and establishes best practices for sigmoid attention as a drop-in softmax replacement in transformers.

## 1 INTRODUCTION

The success of modern machine learning can be largely attributed to the attention mechanism (Bahdanau et al., 2015; Vaswani et al., 2017). Attention uses a sequence-to-sequence (seq-to-seq) map to build context-aware token representations. Classically, attention relies on the softmax function (SoftmaxAttn) to recover token representations as data-dependent convex combinations of values.

Despite its widespread use and effectiveness, softmax in SoftmaxAttn is not without limitations. For instance, the softmax function can sometimes lead to a concentration of attention on just a few features (Yang et al., 2018; Ganea et al., 2019), potentially neglecting other informative aspects of the input data. Moreover, applying SoftmaxAttn requires performing a *row-wise* reduction along the length of the input sequence, which in the case of efficient attention kernels (Dao et al., 2022; Dao, 2023), slows down computations. In this work, we relax this constraint by substituting the *row-wise* softmax operation with an *element-wise* sigmoid nonlinearity. We highlight that the central problem with naïve sigmoid attention (SigmoidAttn) is that of large initial attention norms and propose solutions to alleviate it. **Our contributions are as follows:**

(1) We prove SigmoidAttn is a universal function approximator on seq-to-seq tasks (Sec. 3.1).

(2) We analyze SigmoidAttn's regularity and provide its worst-case Jacobian bound (Sec. 3.2).

(3) We extend FLASHATTENTION2 (Dao et al., 2022; Dao, 2023) with the sigmoid kernel, reducing kernel inference wall-clock time by up to 17% and real world inference by up to 8% (Sec. 4).

(4) We show that SigmoidAttn matches SoftmaxAttn in various tasks and domains (Sec. 5).

---

[*]Primary contributor. For a detailed breakdown of author contributions see Appendix H.
Correspondence to: Jason Ramapuram <jramapuram@apple.com>
[1]Code is available at https://github.com/apple/ml-sigmoid-attention.

## 2 SIGMOID ATTENTION

Let $\boldsymbol{X} \in \mathbb{R}^{n \times d}$ be the input sequence of $n$ vectors, where each vector has dimension $d$. We define three learnable weight matrices $\boldsymbol{W}_q \in \mathbb{R}^{d \times d_{qk}}$, $\boldsymbol{W}_k \in \mathbb{R}^{d \times d_{qk}}$, and $\boldsymbol{W}_v \in \mathbb{R}^{d \times d_v}$, which are used to compute the queries $\boldsymbol{Q} \in \mathbb{R}^{n \times d_{qk}}$, keys $\boldsymbol{K} \in \mathbb{R}^{n \times d_{qk}}$, and values $\boldsymbol{V} \in \mathbb{R}^{n \times d_v}$ as follows:

$$\boldsymbol{Q} = \boldsymbol{X}\boldsymbol{W}_q, \quad \boldsymbol{K} = \boldsymbol{X}\boldsymbol{W}_k, \quad \text{and} \quad \boldsymbol{V} = \boldsymbol{X}\boldsymbol{W}_v. \tag{1}$$

Self-attention (Bahdanau et al., 2015; Vaswani et al., 2017) can be compactly written as

$$\mathrm{SoftmaxAttn}(\boldsymbol{X}) = \mathrm{Softmax}(\boldsymbol{Q}\boldsymbol{K}^T/\sqrt{d_{qk}})\boldsymbol{V}, \tag{2}$$

where the $\mathrm{Softmax}$ function *normalizes each row* of the input matrix. We replace the $\mathrm{Softmax}$ with

$$\mathrm{SigmoidAttn}(\boldsymbol{X}) = \sigma(\boldsymbol{Q}\boldsymbol{K}^T/\sqrt{d_{qk}})\boldsymbol{V},$$
$$\text{with } \sigma : u \mapsto \mathrm{sigmoid}(u + b) := (1 + e^{-(u+b)})^{-1}. \tag{3}$$

Here, $\sigma$ is applied *element-wise* to the input matrix in (3). The activation function $\sigma$ has a hyper-parameter $b \in \mathbb{R}$. In App. E, we discuss an intuitive way to choose the order-optimal bias term, resulting in $b = -\log(n)$. This choice of $b$ allows us to make sense of $\mathrm{SigmoidAttn}$ for any sequence length. Indeed, letting $(\boldsymbol{y}_1, \ldots, \boldsymbol{y}_n) = \mathrm{SigmoidAttn}(\boldsymbol{X})$ be the output sequence, we have

$$\boldsymbol{y}_i = \sum_{j=1}^{n} \frac{\exp(\langle \boldsymbol{W}_q\boldsymbol{x}_i, \boldsymbol{W}_k\boldsymbol{x}_j \rangle)}{\exp(\langle \boldsymbol{W}_q\boldsymbol{x}_i, \boldsymbol{W}_k\boldsymbol{x}_j \rangle) + n} \boldsymbol{W}_v\boldsymbol{x}_j \xrightarrow[n \to +\infty]{} \int \exp(\langle \boldsymbol{W}_q\boldsymbol{x}_i, \boldsymbol{W}_k\boldsymbol{x} \rangle)\boldsymbol{W}_v\boldsymbol{x}d\mu(\boldsymbol{x}), \tag{4}$$

where $\mu = \frac{1}{n}\sum_{j=1}^{n}\delta_{\boldsymbol{x}_j}$ is the empirical measure corresponding to $\boldsymbol{X}$. Notably, (4) still makes sense in the infinite length limit, where the measure $\mu$ is not a sum of Diracs. Wortsman et al. (2023a) do not use a bias, and propose a $n^{-1}$ normalization for various attention activations, such as sigmoid and ReLU, but leave the reason as an open question. Our variable bias has a similar effect in the large $n$ limit, and we posit that recovering a finite output limit as $n$ increases is the why it works in practice.

A multi-head version of (3) is obtained by combining the outputs of several $\mathrm{SigmoidAttn}$, as follows:

$$[\mathrm{SigmoidAttn}_1(\boldsymbol{X}), \ldots, \mathrm{SigmoidAttn}_h(\boldsymbol{X})] \boldsymbol{W}_o, \tag{5}$$

for a learnable output weight matrix $\boldsymbol{W}_o \in \mathbb{R}^{hd_v \times d}$, where $h$ denotes the number of *heads*.

## 3 THEORETICAL PROPERTIES OF SIGMOID ATTENTION

We analyze $\mathrm{SigmoidAttn}$, with two objectives: (1) showing that a transformer architecture remains a universal function approximator when $\mathrm{SigmoidAttn}$ replaces $\mathrm{SoftmaxAttn}$, and (2) recovering a measure of regularity of $\mathrm{SigmoidAttn}$ by computing its Lipschitz constant.

### 3.1 ARE TRANSFORMERS WITH SIGMOID ATTENTION UNIVERSAL APPROXIMATORS?

Yun et al. (2020) demonstrate that classical transformers can approximate continuous sequence-to-sequence functions to arbitrary precision, a property known as the *Universal Approximation Property* (UAP). UAP is highly desirable as it provides proof of an architecture's generalizability and representation capability. As $\mathrm{SigmoidAttn}$ modifies the transformer architecture, it is crucial to theoretically guarantee that this modification does not impact the representation capability and that UAP is retained. We provide this guarantee with the following theorem.

**Theorem 3.1** (UAP for $\mathrm{SigmoidAttn}$)**.** *We denote with $\mathcal{T}_\sigma^{h,d_v,r}$ the class of transformer networks obtainable by combining an arbitrary number of $\mathrm{SigmoidAttn}$ layers (each of $h$ heads of dimension $d_v$) followed by FFN layers of hidden dimension $r$. For any given continuous, permutation-equivariant function $f : \Omega \subset \mathbb{R}^{n \times d} \to \mathbb{R}^{n \times d}$ with compact support $\Omega$, and for any arbitrarily small error $\varepsilon$, there exists a transformer network $g \in \mathcal{T}_\sigma^{4,1,4}$ such that*

$$\left(\int_\Omega \|f(\boldsymbol{X}) - g(\boldsymbol{X})\|_p^p d\boldsymbol{X}\right) \leq \varepsilon, \qquad \text{for} \quad 1 \leq p < \infty. \tag{6}$$

Theorem 3.1 is the exact counterpart of (Yun et al., 2020, Thm. 2), which shows UAP for classical transformers. Our proof largely follows the same path, an outline of the original proof provided in App. C. Here, we present an overview of the main adaptations required to prove Thm. 3.1 for SigmoidAttn, with further details in App. C.1 and C.2.

**Sigmoid Attention layers can implement contextual mappings:** A key step in proving Thm. 3.1 is showing that, even with SigmoidAttn, a sequence of transformer blocks can implement a *Contextual Mapping* (Yun et al., 2020, Def. 3.1). A contextual mapping characterizes a function that maps each input sequence element to an output *uniquely* dependent on the *whole* sequence. This property allows a transformer to capture and store global context within each token, even if each layer only performs pairwise comparisons. Subsequent layers can then use this global information to map individual tokens to the correct output, ultimately approximating any arbitrary sequence-to-sequence function.

In Yun et al. (2020), the contextual mapping is assembled by modifying individual transformer blocks: each block is tuned to react to a specific input token. By stacking a sequence of these blocks, a transformer can be turned into an accumulator, mapping a given input token sequence to a unique global index. This outcome is achieved via a *selective shift layer* (Yun et al., 2020, App. B.5):

$$\Psi(\boldsymbol{X}; b, b')_{i,1} := \begin{cases} \max_k \boldsymbol{X}_{k,1} - \min_k \boldsymbol{X}_{k,1} & \text{if} \quad b < \boldsymbol{X}_{i,1} < b' \\ 0 & \text{otherwise,} \end{cases} \tag{7}$$

and can be approximated using classic attention. Although SigmoidAttn cannot directly approximate (7), our accumulator definition relies on an equivalent selective shift operation:

$$\Psi_\sigma(\boldsymbol{X}; b, b')_{i,1} := \begin{cases} \sum_{k:\boldsymbol{X}_{k,1}>b'} \boldsymbol{X}_{k,1} & \text{if} \quad b < \boldsymbol{X}_{i,1} < b' \\ 0 & \text{otherwise,} \end{cases} \tag{8}$$

which can be approximated by SigmoidAttn (described in App. C.1). In App. C.2.4, we show that (8) shares similar properties with (7), allowing us to use the original proof framework in Yun et al. (2020) and demonstrate that UAP holds in our case as well.

Our proof is largely equivalent to that in Yun et al. (2020), with two relevant differences: to approximate (8), we require SigmoidAttn with *at least four heads* and shifts included in both query and key definitions. In contrast, SoftmaxAttn requires *at least two heads* to approximate (7), with shifts only in the query definition. However, this is primarily a theoretical requirement for the proof and does not affect performance. Notably, the total number of parameters required by both architectures for the approximation follows the same tight scaling of Yun et al. (2020).

## 3.2 Regularity of Sigmoid Attention

As with any layer in a neural network, the regularity of SigmoidAttn is important to study, as it gives insights into the robustness of the corresponding network and the ease of optimizing it. The most standard way to quantify the regularity of a layer function $\phi$ is to compute its *Lipschitz constant* over a set $\mathcal{X}$, that is a constant $C > 0$ such that for all $\boldsymbol{X}, \boldsymbol{Y} \in \mathcal{X}$, it holds $\|\phi(\boldsymbol{X}) - \phi(\boldsymbol{Y})\| \leq C\|\boldsymbol{X} - \boldsymbol{Y}\|$, where $\|\cdot\|$ is the standard Frobenius norm. The *local* Lipschitz constant is the spectral norm of the Jacobian of $\phi$ at $\boldsymbol{X}$. The two are related: the Lipschitz constant of $\phi$ over $\mathcal{X}$ is the greatest local Lipschitz constant for all $\boldsymbol{X} \in \mathcal{X}$. We turn to the theorem giving the regularity of SigmoidAttn:

**Theorem 3.2.** *Define $A = \{|\langle \boldsymbol{W}_q \boldsymbol{x}_i \boldsymbol{W}_k \boldsymbol{x}_j \rangle|, \ i, j \in \{1, \ldots, n\}\} \subset \mathbb{R}$ the set of attention weights, and the scaled activation norms $\sigma_\infty = n \times \sup_{u \in A} |\sigma(u)|$ and $\sigma'_\infty = n \times \sup_{u \in A} |\sigma'(u)|$. Then, the Jacobian of* SigmoidAttn *at $\boldsymbol{X} = (\boldsymbol{x}_1, \ldots, \boldsymbol{x}_n)$ has a spectral norm of at most:*

$$\|\boldsymbol{W}_v\|_2 \left( \sigma_\infty + 2\sigma'_\infty \|\boldsymbol{W}_q^T \boldsymbol{W}_k\|_2 \left( \frac{1}{n} \sum_{i=1}^n \|\boldsymbol{x}_i\|_2^2 \right) \right). \tag{9}$$

The proof is found in App. D. In SigmoidAttn, if we assume that the attention weights $\langle \boldsymbol{W}_q \boldsymbol{x}_i, \boldsymbol{W}_k \boldsymbol{x}_j \rangle$ are all bounded by a constant $\mu$ — this is true, e.g., if the activations are bounded — we get $\sigma_\infty \leq \exp(\mu)$ and $\sigma'_\infty \leq \exp(\mu)$ thanks to the choice of $b = -\log(n)$. The bound in Thm. 3.2 depends only on the *average* squared-norm of the input sequence $\boldsymbol{x}_i$, while classical results for the study of attention all rely on the largest value of $\|\boldsymbol{x}_i\|_2^2$ (Kim et al., 2021; Castin et al., 2023). This is another consequence of the simplicity of sigmoid attention and is due to the removal of the

normalizing constant in $\mathrm{SoftmaxAttn}$. Our result implies that if all $\boldsymbol{x}_i$ are within a ball of radius $R$ then the Lipschitz constant of $\mathrm{SigmoidAttn}$ grows at most like $R^2$, but it is stronger since we can apply this to unbounded distributions $\boldsymbol{x}_i$; it matters only that the second moment is bounded. This result contrasts sharply with the bounds obtained for $\mathrm{SoftmaxAttn}$: Castin et al. (2023, Thm. 3.4.) show that there exists a sequence $\boldsymbol{X} = (\boldsymbol{x}_1, \ldots, \boldsymbol{x}_n)$ with $\|\boldsymbol{x}_i\|_2 \leq R$ for all $i$ such that the spectral norm of the Jacobian of $\mathrm{Attn}$ at $\boldsymbol{X}$ is at least $cR^2 \exp(cR^2)$ for some constant $c > 0$. On the other hand, our bound scales in $R^2$: this means that the local Lipschitz constant of $\mathrm{SigmoidAttn}$ is much lower than the worst local Lipschitz constant of $\mathrm{SoftmaxAttn}$. Note that this result does not inform us of the practical average case Lipschitz constant, which is likely to be much lower for both Softmax and Sigmoid attention. Upper bounds on the Lipschitz constant of $\mathrm{SigmoidAttn}$ are of particular interest to study the dynamics of attention, as done, e.g., in (Geshkovski et al., 2024; 2023)

# 4 FLASHSIGMOID: HARDWARE-AWARE IMPLEMENTATION

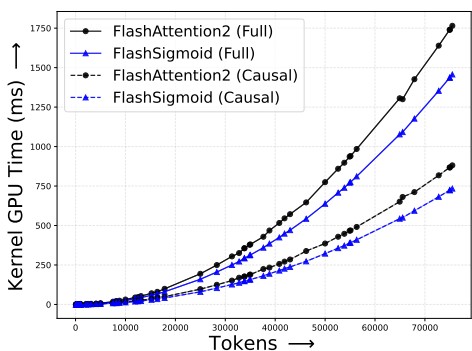
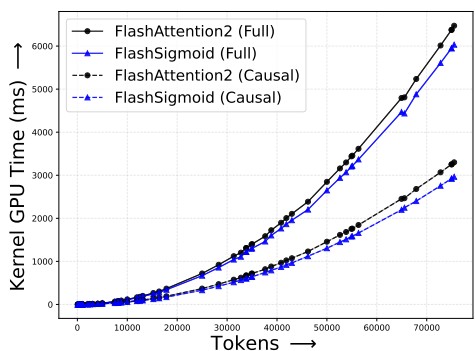

(a) Inference mode kernels on H100.

(b) Training mode kernels on H100.

Figure 1: Average kernel speed-up for FLASHSIGMOID over FLASHATTENTION2 for sequence lengths 64–78k. Inference is $17.39\%$ faster for self-attention and $18.76\%$ for causal attention. Training is $6.53\%$ faster for self-attention and $9.46\%$ for causal attention.

Memory speed has not kept pace with recent gains in computation speed (Choquette, 2023; Jouppi et al., 2017; Hannun et al., 2023). Consequently, attention computations on modern architectures have been IO-bound by memory accesses (Ivanov et al., 2021). FLASHATTENTION (Dao et al., 2022) and FLASHATTENTION2 (Dao, 2023) address these shortcomings by optimizing GPU memory hierarchy utilization to accelerate attention computations. Motivated by the speed boost provided by these approaches, we develop FLASHSIGMOID, a hardware-aware implementation of $\mathrm{SigmoidAttn}$. Like previous works, FLASHSIGMOID employs three core ideas:

**Tiling: Divide and Conquer Approach to Attention:** Similar to FLASHATTENTION and FLASHATTENTION2, FLASHSIGMOID processes input parts in parallel to compute attention outputs in blocks, efficiently combining partial results to generate the final attention output.

**Kernel Fusion:** Like FLASHATTENTION and FLASHATTENTION2, FLASHSIGMOID implements the computational steps of both forward and backward passes of $\mathrm{SigmoidAttn}$ as single GPU kernels, minimizing memory accesses and improving memory efficiency by avoiding materialization of intermediate activations on High-Bandwidth Memory (HBM).

**Activation Recomputation:** The backward pass of sigmoid attention requires the sigmoid activation matrix, which, if materialized on GPU HBM, results in slower implementation and memory inefficiencies. FLASHSIGMOID addresses this by retaining only query, key, and value tensors for re-computation of the sigmoid activation matrix during the backward pass. Despite increased FLOPs, this approach proves faster in wall-clock time as well as more memory-efficient than the alternative approach of materializing and retaining the attention matrix.

The forward and backward pass algorithms of FLASHSIGMOID can be found in App. F.1. Here, we highlight key differences between FLASHSIGMOID and FLASHATTENTION/FLASHATTENTION2. The point-wise nature of $\mathrm{SigmoidAttn}$ results in a faster and more memory-efficient implementation by removing the need to compute the softmax normalization and materialize it to HBM. A reduction

in the number of kernel dispatches also speeds up FLASHSIGMOID. Further, FLASHSIGMOID does not require accumulation and tracking of intermediate variables (row-sum and maximum of blocks) in the forward and backward passes which saves computation cost and reduces register pressure. We use $\text{sigmoid}(x) = 0.5 \cdot (1 + \tanh(0.5 \cdot x))$ to optimize the sigmoid computation on GPU. The speed up in FLASHSIGMOID compared to FLASHATTENTION arises from optimizing hardware bottlenecks; theoretically, SigmoidAttn is slower than SoftmaxAttn (App. G.6).

To measure the performance improvements of FLASHSIGMOID, we compare the timings of the kernels in its forward and backward passes against those of FLASHATTENTION2. The details of this benchmarking on H100 and A100 GPUs can be found in App. F.2. Measuring GPU computation time, we observe a $17.39\%$ speed-up during inference and a $6.53\%$ speed-up during training for attention over randomly initialized data on H100 GPU (Fig. 1). In practice, these gains may be affected by other bottlenecks, such as movement of tensors between CPU or GPU memory, computations in other layers, and communication overhead in distributed training and inference. However, we demonstrate that FLASHSIGMOID speeds up training by $\sim\textbf{4\%}$ and inference by $\sim\textbf{8\%}$ in a realistic end-to-end setup. The details of wall-clock time improvements with FLASHSIGMOID are in App. F.3. We also note that practical machine learning workflows are dominated by inference rather than training.

## 5 EXPERIMENTS

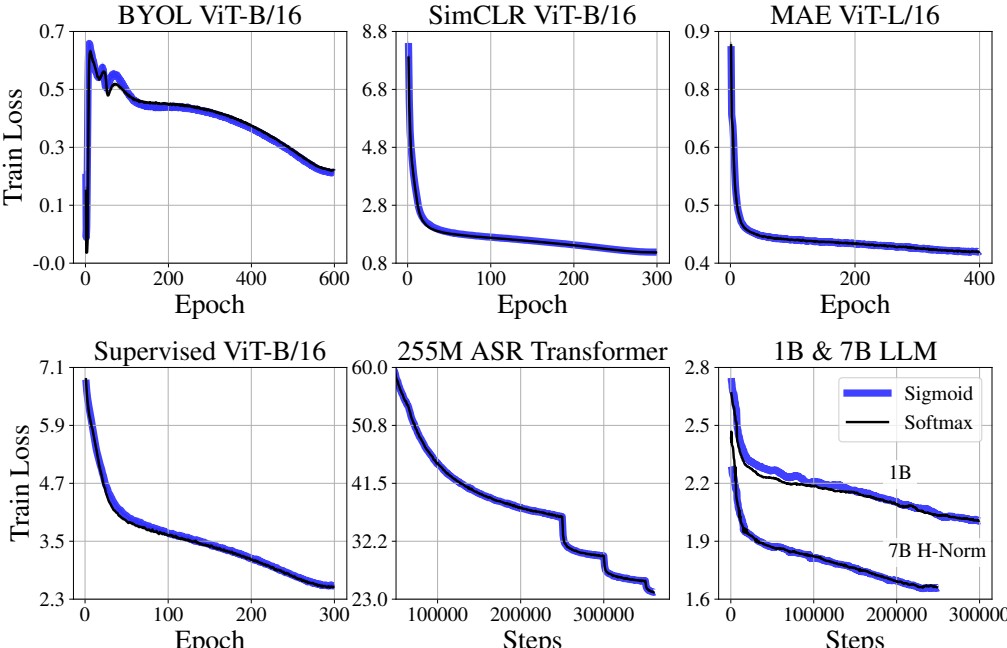

Figure 2: Train losses comparing SigmoidAttn with SoftmaxAttn.

To empirically validate SigmoidAttn, we evaluate across several domains: supervised image classification using vision transformers (Dosovitskiy et al., 2021), self-supervised image representation learning with SimCLR (Chen et al., 2020; Zhai et al., 2023a), Bootstrap Your Own Latent (BYOL) (Grill et al., 2020; Busbridge et al., 2023) and Masked AutoEncoders (MAE) (He et al., 2022) as well as automatic speech recognition (ASR) (Synnaeve et al., 2020; Gulati et al., 2020b) and auto-regressive language modeling (LM) (Brown et al., 2020). We also validate sequence length generalization on TED-LIUM v3 (Hernandez et al., 2018) for ASR and in small scale synthetic experiments in App. G.5.4. Across all these domains and algorithms, we demonstrate that SigmoidAttn matches the performance of SoftmaxAttn (Fig. 2 and 22), while offering training and inference speed-ups as highlighted in Sec. 4. Empirically we make the following observations:

(1) SigmoidAttn is effective for vision tasks without a bias (except MAE), but relies on Layer-Scale (Touvron et al., 2021) to match the performance of the baseline SoftmaxAttn (Fig. 10-a)

in a hyper-parameter free manner.[2] All results presented for SoftmaxAttn also fairly add LayerScale unless specified.

(2) LM and ASR are sensitive to the initial norm $||\sigma(\boldsymbol{QK}^T/\sqrt{d_{qk}})\boldsymbol{V}||$. Modulation is required via (a) relative positional embeddings like ALiBi (Press et al., 2022), which reduces the initial attention norm by shifting logit mass near zero under SigmoidAttn, (b) appropriate initialization of $b$ to achieve the same effect – enabling usage of any positional embedding, (c) using hybrid-norm (Appendices G.3.3, G.4 and G.7) at the expense of an extra normalization layer.

## 5.1 ABLATIONS

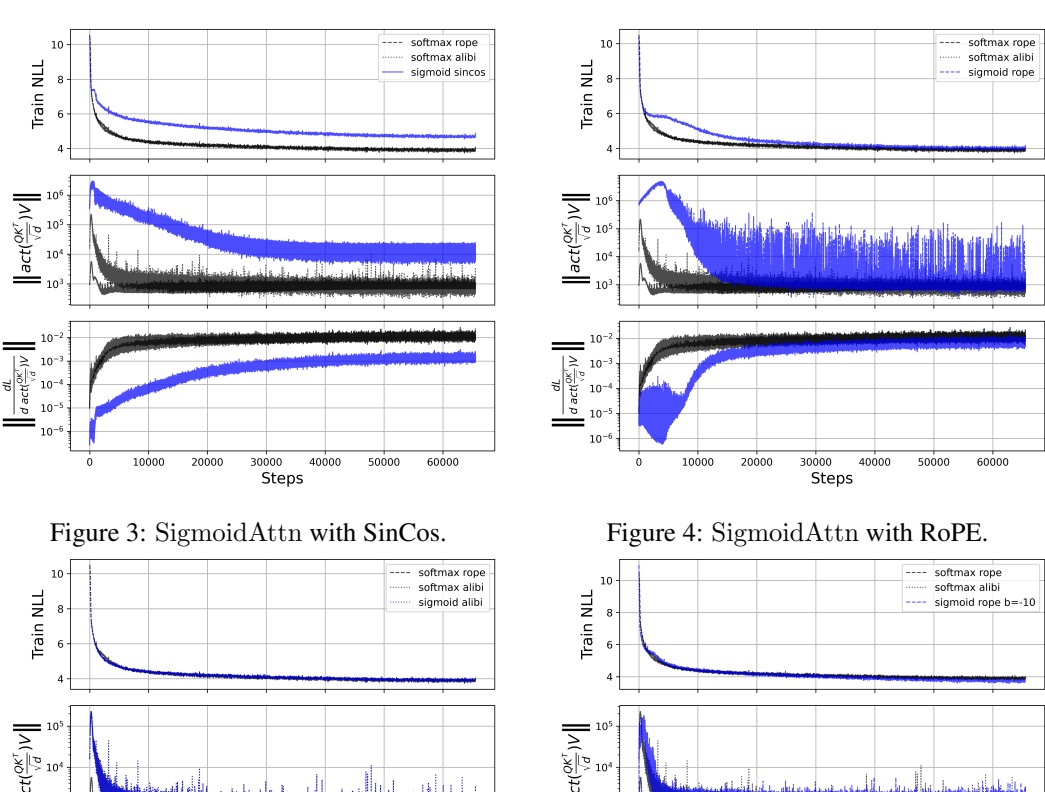

Figure 3: SigmoidAttn with SinCos.

Figure 4: SigmoidAttn with RoPE.

Figure 5: SigmoidAttn with ALiBi.

Figure 6: SigmoidAttn with RoPE, $b = -10$.

We begin with ablations to dissect the benefits of each of our introduced components. To gain intuition about SigmoidAttn, we developed a research-friendly auto-regressive (AR) LM training framework to measure all components of attention and validate the effects of LayerScale, LayerNorm applied to Q and K (QK norm), different positional embedding techniques, and initialization values for $b$.

**Mitigating Large Attention Norms** We train a single layer AR transformer block (E=3072, D_FF=12288) on the realnews split of C4 (Raffel et al., 2020). We train for $2^{16}$ steps using a batch size of 6 and max sequence length of 4096 using a single cycle cosine learning rate (LR) schedule without weight decay. SigmoidAttn initially underperformed SoftmaxAttn when using absolute sinusoidal (SinCos) (Fig. 3) or relative (Fig. 4) positional embeddings (PE), which we attribute

---

[2]Appendix G.2.2 demonstrates that supervised vision tasks using SigmoidAttn without LayerScale can match baseline SoftmaxAttn performance by relying on *learnable* scalar bias and temperature: $\{b, t\} \in \mathbb{R}$.

to high initial attention Frobenius norms, $\|\sigma(\boldsymbol{QK}^T/\sqrt{d})\boldsymbol{V}\|$. A corresponding evolution of the attention distribution and sparsity can be seen in Appendix Fig. 30 and Fig. 31 on a synthetic task. To address these larger attention norms, we propose: (a) using ALiBi (Press et al., 2022) whose relative bias moves initial attention logit mass to the zero region under the sigmoid activation, producing equivalent train negative log-likelihoods (Fig. 5); or (b) set the attention logit bias $b$ to a negative offset proportional to the sequence length, $b \propto -\ln n$ (see App. G.1.2 for an ablation on $b$). This enables the usage of other PE techniques like RoPE (Su et al., 2024) (Fig. 6).

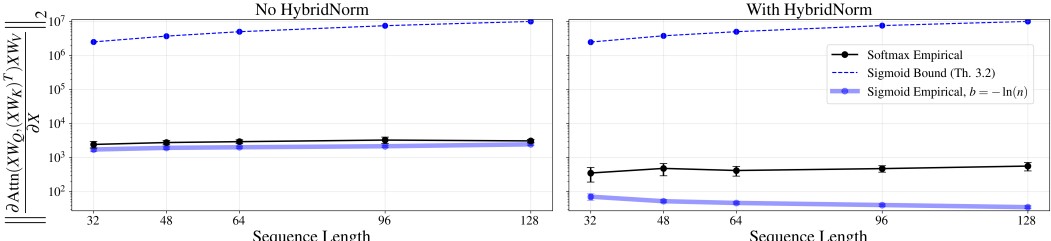

Figure 7: Regularity analysis comparing $\mathrm{SigmoidAttn}$ vs. $\mathrm{SoftmaxAttn}$ ($10\times$ trials per $n$). $\mathrm{SoftmaxAttn}$ theoretical bound is off scale and thus omitted.

**Empirical Analysis of Attention Regularity**    To validate our theoretical analysis (Section 3.2), we measure Jacobian norms of $\mathrm{SigmoidAttn}$ and $\mathrm{SoftmaxAttn}$ across sequence lengths (Figure 7). Using autograd, we compute exact Jacobian norms for both mechanisms, with and without HybridNorm (Appendices G.3.3 and G.7), comparing them to theoretical bounds ($\mathrm{SoftmaxAttn}$ bound omitted as it exceeds scale). Both variants show empirical norms (solid lines) well below their theoretical bounds (dashed lines). With our proposed bias initialization ($b = -\ln(n)$), $\mathrm{SigmoidAttn}$ achieves lower norms than $\mathrm{SoftmaxAttn}$ in both settings, suggesting improved regularity. This aligns with its strong task performance (Section 5). Additionally, HybridNorm (Figure 7, right) reduces norms for both mechanisms compared to baseline (left), highlighting normalization's role in attention stability at longer sequences.

**LayerScale**    To validate the need for LayerScale, we follow Wortsman et al. (2023b) to quantify the impact on stability. All models are trained with RoPE with $b \propto -\ln n$, using AdamW (Loshchilov & Hutter, 2017) on the realnews split of C4 with $(\beta_1, \beta_2) = (0.9, 0.95)$, $\epsilon = 10^{-8}$, $wd = 0$, batch size 24, maximum token sequence length of 512 from the T5 tokenizer (Raffel et al., 2020), cosine LR schedule of $2^{14}$ steps including a linear warmup of $2^{10}$ steps. Models have $n_{\mathrm{heads}} = \kappa$, $n_{\mathrm{layers}} = 2 \times \kappa$, $d_{\mathrm{model}} = 64 \times \kappa$ and $d_{\mathrm{feed\text{-}forward}} = 256 \times \kappa$ for a scaling value $\kappa \in \{1, 2, 4, 8, 16\}$ leading to models with $\{2.2, 4.9, 15.0, 67.0, 440.0\}M$ trainable non-embedding parameters. Following Wortsman et al. (2023b), we sweep learning rates $\eta \in \{3 \times 10^{-4}, 1 \times 10^{-3}, 3 \times 10^{-3}, 1 \times 10^{-2}, 3 \times 10^{-2}, 1 \times 10^{-1}, 3 \times 10^{-1}\}$. LR sensitivity is defined as $\mathbb{E}_{\eta \in [a,b]}[\min(\ell(\mathcal{A}(\eta)), \ell_0) - \ell^*]$ where $\ell(\mathcal{A}(\eta))$ is the loss achieved by the learning algorithm $\mathcal{A}$ with LR $\eta$, $\ell_0$ is the loss at initialization, and $\ell^*$ is the loss achieved by the best LR. LayerScale is initialized at $10^{-4}$. Unlike vision tasks, where LayerScale *improves performance* (Fig. 10-a), in LM, we observe that $\mathrm{SoftmaxAttn}$ slightly benefits from LayerScale, while the performance of $\mathrm{SigmoidAttn}$ remains largely unaffected.

**Stability with QK Norm**    To explore the stability of $\mathrm{SoftmaxAttn}$ vs. $\mathrm{SigmoidAttn}$ we repeat the analysis of Wortsman et al. (2023b), as described in the LayerScale analysis, to investigate the impact of QK norm (Dehghani et al., 2023). For language modeling, both $\mathrm{SigmoidAttn}$ and $\mathrm{SoftmaxAttn}$ exhibit sensitivity to learning rate changes without QK norm. However, incorporating QK norm significantly stabilizes performance (Fig. 9). In vision tasks, $\mathrm{SigmoidAttn}$ demonstrates robustness with and without QK norm (Fig. 10-a) and without the need for $n^{-\alpha}$ normalization from Wortsman et al. (2023a).[3]

**Multi-query attention (MQA)**    In Fig. 10-b we explore MQA (Shazeer, 2019) for vision using only one head for $\{\boldsymbol{K}, \boldsymbol{V}\}$. We find that both $\mathrm{SigmoidAttn}$ and $\mathrm{SoftmaxAttn}$ perform equally well with or without multiple heads even at the small scale of ViT-B/16.

**Activation Function Ablations**    As in Wortsman et al. (2023a), various activation functions, when combined with LayerScale and QK norm, perform equally well for vision tasks (Fig. 10-c). However, for sequence-critical tasks like ASR, activation functions such as ReLU pose instabilities

---

[3]We ablate multiplicative sequence length scaling in more detail in App. G.1.1.

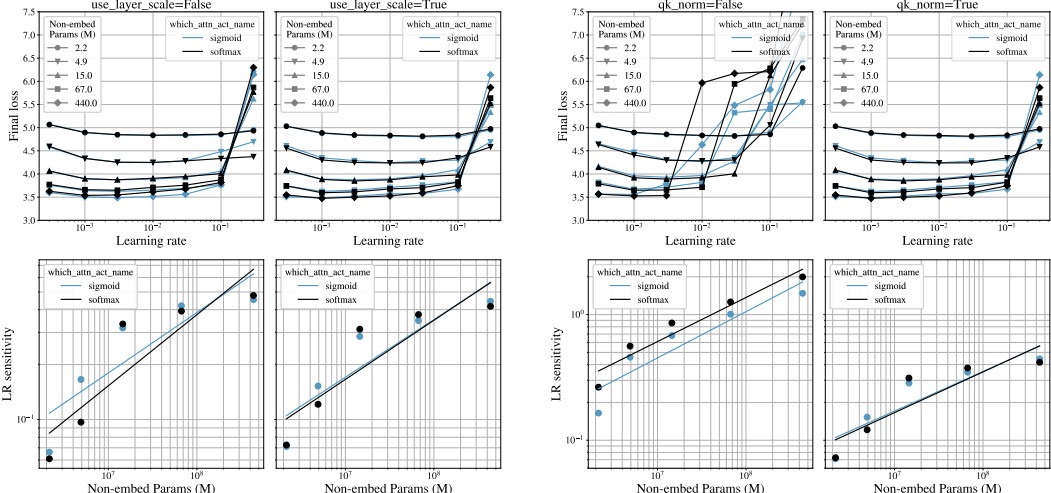

Figure 8: LR sensitivity LayerScale ablation.    Figure 9: LR sensitivity QK norm ablation.

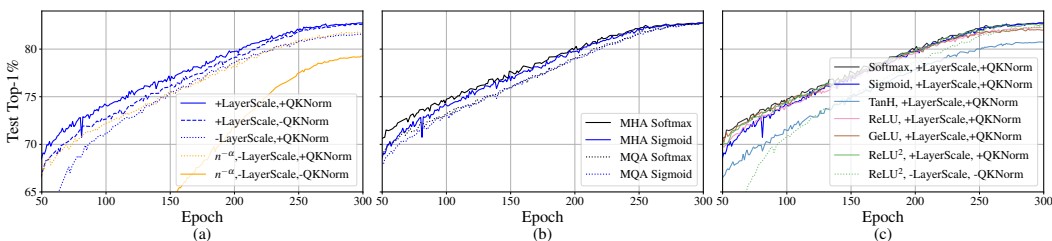

Figure 10: ImageNet1k ViT-B/16 classification. (a) SigmoidAttn is robust without QK norm (+LayerScale, -QKNorm). Removing LayerScale reduces accuracy by 1.0% (-LayerScale, +/-QKNorm). $n^{-\alpha}$ normalization (Wortsman et al., 2023a) underperforms without LayerScale. (b) SigmoidAttn multi-query attention (MQA) (Shazeer, 2019) with one head matches multi-head attention (MHA). (c) Sigmoid with LayerScale and QK norm performs comparably to other activations, except TanH. ReLU$^2$ (Hua et al., 2022) underperforms without LayerScale and QK norm.

and underperform. In the same figure, we also compare to the ReLU$^2$ proposal from Hua et al. (2022) and find that it underperforms without LayerScale and QK norm.

## 5.2 SUPERVISED IMAGE CLASSIFICATION

Vision transformers (Dosovitskiy et al., 2021) extend transformers (Vaswani et al., 2017) to treat $K \times K$ image grids as disparate tokens. All tokens are refined through sequential layers of self-attention, pooled using a CLS token or global average pooling layer, and optimized using the negative log likelihood, $\ln p(\boldsymbol{y}|\boldsymbol{x})$. We train ViT-B/16 models using $\mathbb{R}^{224 \times 224 \times 3}$ images for 300 epochs using the recipe provided in App. G.2.4. We use the same set of training hyper-parameters for both SoftmaxAttn and SigmoidAttn, changing only the activation function between trials. The train negative log-likelihood is reported in Fig. 2 and the test top-1% is reported in Fig. 22. We find that SigmoidAttn matches both the training dynamics and the evaluation performance of SoftmaxAttn.

## 5.3 SELF-SUPERVISED IMAGE REPRESENTATION LEARNING

Self-supervised representation learning (SSL) exploits vast quantities of unlabeled data to learn semantic representations based on inductive biases such as augmentation invariance (SimCLR Chen et al. (2020), BYOL (Grill et al., 2020)) or reconstruction from compressed representations (MAE (He et al., 2022)). We employ vision transformer training recipes from Zhai et al. (2023a) and Busbridge et al. (2023) (App. G.2.4) for SimCLR and BYOL. As with supervised learning, we use the same set of training hyper-parameters for both SoftmaxAttn and SigmoidAttn, changing only the activation function between trials. Figure 2 reports the train losses, and Fig. 22 highlights the linear probe

Table 1: Word error rate (%) on LibriSpeech test sets and TED-LIUM v3 (Hernandez et al., 2018) ("TED", joint validation and test sets split according to duration) for transformer (255M params) with either SoftmaxAttn or SigmoidAttn (LayerScale and QK norm are used with $b = -\log n$) trained on LibriSpeech 960h data (mean duration is 10-15s). Hyper-parameters are in App. G.4.

| ATTN | PE | TEST-CLEAN | TEST-OTHER | TED 0-10S | TED 10-20S | TED 20-30S | TED 30S+ |
|---|---|---|---|---|---|---|---|
| SOFTMAX | | 2.3 | 5.7 | 12.4 | 10.5 | 11.9 | 9.1 |
| SIGMOID | | 2.4 | 5.5 | 12.4 | 10.3 | 12.3 | 9.7 |
| - QK NORM | | | | UNSTABLE, GRADIENT NORM AND LOSS SPIKES | | | |
| - LAYERSCALE | CAPE | 2.5 | 6.1 | 13.6 | 11.5 | 13.4 | 8.9 |
| SIGMOID ($b = -10$, LEARNABLE) | | 2.3 | 5.5 | 12.1 | 10.5 | 13.0 | 9.3 |
| SIGMOID ($b = -5$ IN $Q$, LEARNABLE) | | 2.3 | 5.4 | 12.2 | 10.8 | 12.4 | 9.9 |
| - QK NORM | | | | UNSTABLE, GRADIENT NORM AND LOSS SPIKES | | | |
| SOFTMAX | | 2.2 | 5.5 | 12.7 | 10.6 | 12.8 | 9.5 |
| SIGMOID | | 2.3 | 5.4 | 12.3 | 10.1 | 12.3 | 8.6 |
| SIGMOID ($b = -10$, LEARNABLE) | RoPE | 2.2 | 5.2 | 12.4 | 10.5 | 12.3 | 21.8 |
| $+\,\alpha = 1$ | | 2.7 | 6.6 | 14.1 | 12.0 | 14.5 | 14.9 |
| SIGMOID ($b = -5$ IN $Q$, LEARNABLE) | | | | UNSTABLE, GRADIENT NORM AND LOSS SPIKES | | | |
| SOFTMAX | | 2.2 | 5.4 | 12.3 | 10.7 | 12.1 | 8.6 |
| SIGMOID | | 2.3 | 5.1 | 12.3 | 10.5 | 12.6 | 9.1 |
| SIGMOID ($b = -10$, LEARNABLE) | ALiBi | 2.2 | 5.2 | 12.4 | 10.4 | 11.7 | 9.1 |
| $+\,\alpha = 1$ | | 2.6 | 6.6 | 13.9 | 11.9 | 14.2 | 8.6 |
| SIGMOID ($b = -5$ IN $Q$, LEARNABLE) | | 2.2 | 5.2 | 12.1 | 10.4 | 12.0 | 8.2 |

and finetuned test top-1%. Despite the diverse training objectives in SSL, SigmoidAttn matches SoftmaxAttn while improving training and inference throughput (Sec. 4).

## 5.4 AUTOMATIC SPEECH RECOGNITION (ASR)

We benchmark ASR using LibriSpeech data (Panayotov et al., 2015) on 100h and 960h settings of paired speech and text transcriptions. Our PyTorch implementations of encoder-based vanilla transformer (Synnaeve et al., 2020) and conformer (Gulati et al., 2020a) are trained with Connectionist Temporal Classification (CTC) (Graves et al., 2006) w/ BF16 mixed precision, w/o QK norm and w/o LayerScale. After extensively tuning SoftmaxAttn baselines, we switch to SigmoidAttn per (3) without any other changes. We investigate the effects of post/pre-LayerNorm, model depth, optimizer type, small data regime, and connection to local attention, with details in App. G.4.

Our main findings are: i) CAPE (Likhomanenko et al., 2021) PE is the most unstable for SigmoidAttn; ii) post-LayerNorm models with SoftmaxAttn are hard to match with stable SigmoidAttn; iii) w/o QK norm SigmoidAttn is unstable and significant spikes happen in both gradient norms and training loss; iv) LayerScale is needed for generalization; v) learnable bias $b = -10$ gives no loss and gradient norms spikes while matching the SoftmaxAttn (which does not benefit from the improved throughput of FLASHSIGMOID); vi) adding a learnable bias, $b = -5$, to $Q$ instead of the attention logits also solves the initial large attention norms for CAPE and ALiBi but not for RoPE; vii) $b = -\log n$ gives rare (2-5 times) marginal gradient norms spikes with smooth loss while matching SoftmaxAttn.

Table 1 shows the main result for pre-LayerNorm transformers with CAPE, RoPE, and ALiBi, where SigmoidAttn uses LayerScale, QK norm, $b = -\log n$, and no sequence normalization. The bias is ablated with learnable bias (one per layer) in attention or $Q$ with or without sequence normalization. SigmoidAttn is stabilized with bias while matching SoftmaxAttn, and $b = -\log n$ works well. In most cases, bias allows generalization to longer sequences without sequence normalization, except for RoPE where it helps for longer sequences but hurts overall performance.

## 5.5 AUTOREGRESSIVE LARGE LANGUAGE MODELING

We train all models using the Llama2 recipe (Touvron et al., 2023) (with ALiBi instead of RoPE) and the RedPajama (Computer, 2023) dataset in JAX without FLASHATTENTION using the AXLearn framework[4] (App. G.3 for detailed hyper-parameters). Initial experiments at 85M parameters established basic stability requirements (App. G.3.1), with attention bias $b = -\log(n)$ (n = 4096) providing effective results. At 1B n = 2048 scale with $b = -\log(n)$, SigmoidAttn matches the train NLL and evaluation results of SoftmaxAttn (Tab. 2 top row) while improving throughput by **1.12×**.

---

[4]https://github.com/apple/axlearn

Table 2: LLM English evaluation. All models use ALiBi. Detailed ablations in Appendix G.3.3.

| MODEL | SIZE | SEQ. LEN. | ARC EASY | ARC CHAL. | HELLA-SWAG | PIQA | SCIQ | WINO-GRANDE | LAMBADA OPENAI | TRIVIAQA (1-SHOT) | WEBQS (1-SHOT) | AVG | STEP TIME (S) |
|---|---|---|---|---|---|---|---|---|---|---|---|---|---|
| SOFTMAX | 1B | 2K | 62.2 | 26.8 | 42.4 | 59.0 | 72.3 | 88.1 | 58.4 | 19.9 | 15.4 | 49.4 | 0.38 |
| SIGMOID | 1B | 2K | 62.8 | 28.8 | 42.5 | 59.7 | 70.3 | 88.6 | 59.7 | 19.1 | 13.8 | 49.5 | 0.34 |
| SOFTMAX | 1B | 4K | 62.6 | 27.7 | 42.4 | 58.6 | 71.1 | 88.2 | 58.6 | 18.9 | 14.7 | 49.2 | 0.84 |
| SIGMOID | 1B | 4K | 60.5 | 27.3 | 41.3 | 57.8 | 70.5 | 87.0 | 57.6 | 18.9 | 12.6 | 48.2 | 0.67 |
| SOFT (H-NORM) | 1B | 4K | 61.7 | 26.8 | 43.4 | 59.4 | 70.6 | 88.6 | 60.8 | 20.5 | 12.9 | 49.4 | - |
| SIGM. (H-NORM) | 1B | 4K | 63.5 | 28.1 | 43.5 | 60.7 | 70.8 | 88.9 | 59.0 | 20.9 | 16.0 | 50.2 | - |
| SOFT (H-NORM) | 7B | 4K | 71.2 | 39.9 | 53.2 | 65.5 | 75.6 | 91.8 | 67.2 | 37.7 | 21.8 | 59.0 | 3.85 |
| SIGM. (H-NORM) | 7B | 4K | 72.7 | 40.5 | 53.5 | 66.2 | 76.0 | 92.5 | 66.5 | 39.5 | 21.8 | 59.6 | 3.4 |

At the 1B n = 4096 scale, using just $b = -\log(n)$ we observe a **1.25×** speedup; however, slight instabilities prevent SigmoidAttn from matching the strong performance of SoftmaxAttn (Tab. 2 second row). We address these issues through hybrid-norm, an extra normalization layer applied on the output of the attention operation, $x + \text{norm}(\sigma(\boldsymbol{QK}^T/\sqrt{d_{qk}})\boldsymbol{V})$, more details in App. G.3 and G.7. With hybrid-norm, SigmoidAttn matches the train NLL and slightly outperforms SoftmaxAttn on English evaluation results (50.2% vs. 49.4% – Tab. 2 third row). In App. G.3.3 we ablate various design choices at 1B scale, including norm structures, position embedding techniques, and attention bias configurations.

At 7B n = 4096, SigmoidAttn with hybrid-norm demonstrates compelling advantages compared to SoftmaxAttn with hybrid-norm [5] : it matches the train NLL of SoftmaxAttn (Figure 2), while delivering a **1.13×** speedup (Tab. 2 bottom row). The model shows marginal improvements on challenging tasks, including both reasoning (ARC-Challenge: 40.5% vs 39.9%) and knowledge retrieval (TriviaQA: 39.5% vs 37.7%), with better average performance across all benchmarks (59.6% vs 59.0%). These results establish SigmoidAttn with hybrid-norm as an efficient alternative for large-scale language modeling, offering both computational and performance benefits.

## 6 RELATED WORK

Recent studies in supervised image classification (Wightman et al., 2021) and self-supervised learning (SSL), including approaches like SigLIP (Zhai et al., 2023b), are shifting large-scale machine learning training from output conditional categorical distributions, traditionally parameterized by softmax functions, to richer pointwise Bernoulli conditionals parameterized by sigmoid functions. In this study, our focus shifts to refining the model's internal mechanics, specifically by substituting the softmax component of the attention mechanism with a pointwise sigmoid function.

Previous work has explored the replacing softmax with the ReLU activation in both practical (Shen et al., 2023; Hron et al., 2020) and theoretical settings (Bai et al., 2023; Fu et al., 2023). Other works explores using the ReLU$^2$ activation (Hua et al., 2022), exploring purely linear attention (Katharopoulos et al., 2020; Lu et al., 2021; Koohpayegani & Pirsiavash, 2024) or cosine-similarity based attention (Luo et al., 2018; Liu et al., 2022). Our work builds upon these explorations, particularly Wortsman et al. (2023a), which replaces softmax with various activation functions scaled by $n^{-\alpha}$, where $n$ corresponds to the sequence length and $\alpha$, a hyper-parameter. However, we find that their formulation does not match expected performance without proper $b$ initialization and the use of LayerScale (Fig. 10-a, App. G.1.1).

## 7 CONCLUSION

In this work, we present a comprehensive theoretical and empirical study of sigmoid attention as an alternative to softmax attention in transformers. We prove that transformers with sigmoid attention are universal function approximators with improved regularity, and identify LayerScale and prevention of large initial attention norms as key factors for successful training. We introduce FLASHSIGMOID, a memory-efficient variant providing a 17% inference kernel speed-up. Extensive experiments across language, vision, and speech demonstrate that properly normalized sigmoid attention matches softmax attention performance on various tasks and scales. Our findings establish sigmoid attention as a viable alternative, unifying prior work and establishing best practices for its application in transformers.

---

[5]One-to-one trained 7B Sigmoid and 7B Softmax weights are available.

## 8 ACKNOWLEDGEMENTS

We thank Zakaria Aldeneh, Samy Bengio, Navdeep Jaitly, David Koski, Pau Rodriguez Lopez, Hadi Pouransari, and Skyler Seto for their helpful feedback and critical discussions throughout the process of writing this paper; Okan Akalin, Hassan Babaie, Michael Brooks, Brian Gamp, Denise Hui, Mubarak Seyed Ibrahim, Li Li, Rajat Phull, Evan Samanas, Guillaume Seguin, and the wider Apple infrastructure team for assistance with developing and running scalable, fault tolerant code. Names are in alphabetical order by last name within group.

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

# Appendices

## A    LIMITATIONS

While our work demonstrates that SigmoidAttn can serve as a viable drop-in replacement for SoftmaxAttn in many domains and scales, there are a few key limitations to note:

(1) In large-scale (1B parameter, 4096 context length) language modeling, we observed some gradient norm spikes and a slight performance gap between SigmoidAttn and SoftmaxAttn (Table 2). While runs at smaller context lengths (1B parameter, n=2048) were stable and matched SoftmaxAttn performance, we required the use of hybrid-norm to stabilize n=4096 sequence length models. Hybrid-norm does incur a slight extra performance penalty which we quantify in Appendix G.7.

(2) Our theoretical analysis proves that transformers with SigmoidAttn are universal function approximators and have improved regularity compared to SoftmaxAttn. However, the bounds we derive, while tighter than those for SoftmaxAttn, may not be maximally tight. There could be room for further theoretical refinements.

(3) We focused our empirical evaluation on standard benchmarks in language, vision, and speech domains. Performance on more niche or emerging applications remains to be validated.

(4) In automatic speech recognition experiments, we observed that SigmoidAttn can be sensitive to the choice of positional embeddings and may require careful initialization of the attention bias term to ensure stable training. Specifically, we found that the CAPE positional embedding was the most unstable for SigmoidAttn. Further work is needed to develop robust initialization schemes that work well across different positional embeddings. Moreover we found that w/o QK norm or with post-LayerNorm SigmoidAttn is unstable and can underperforms SoftmaxAttn, thus further investigation is needed.

(5) FLASHSIGMOID demonstrates promising inference and training speed-ups by exploiting SigmoidAttn's simpler kernel structure compared to SoftmaxAttn. However, realizing these gains at scale in distributed training setups may require additional engineering to optimize communication bottlenecks.

Despite these limitations, we believe this work establishes a strong foundation for SigmoidAttn, unifying prior art and demonstrating its potential as a drop-in SoftmaxAttn replacement. We hope our theoretical grounding and empirical results motivate further research into this simple yet effective architectural variation.

## B    BROADER IMPACT

The development of efficient and theoretically grounded attention mechanisms has the potential for significant positive impact across a range of applications. By establishing SigmoidAttn as a viable alternative to SoftmaxAttn, our work expands the toolkit of architectural choices available to researchers and practitioners. Positive impacts of this work may include:

(1) Improved computational efficiency: FLASHSIGMOID's faster kernel implementation could lead to more efficient training and inference for attention-based models, reducing energy consumption and enabling deployment on resource-constrained devices. This could democratize access to powerful models.

(2) Theoretical understanding: Our universal approximation results and tighter bounds on the regularity of SigmoidAttn contribute to a deeper theoretical understanding of this key component. A stronger theoretical foundation can guide principled model design and architectural search.

(3) Application-specific benefits: Across language, vision, and speech domains, SigmoidAttn's performance could translate into improved user experiences, such as more natural language interactions, enhanced image understanding, and robust speech recognition. These advancements could have positive societal impacts, such as improved accessibility tools and more effective educational technologies.

However, as with any foundational machine learning advance, there are also risks of negative impacts that must be considered and mitigated:

(1) Fairness and bias considerations: As with any machine learning model, it is important to carefully evaluate SigmoidAttn based models for fairness and potential biases when applied to sensitive use cases. The unique properties of SigmoidAttn may have unexpected interactions with data biases. Researchers and practitioners should follow best practices for auditing and mitigating unwanted biases to ensure equitable outcomes.

(2) Environmental impact: While FLASHSIGMOID is more computationally efficient than FLASHATTENTION, the overall trend of scaling up attention-based models has significant energy costs. Further efficiency improvements and the use of renewable energy sources are important to mitigate environmental harms.

We believe that the benefits of SigmoidAttn outweigh the risks, but it is crucial for the research community to actively consider and address these potential negative impacts. By doing so, we can work towards a future where the efficiency and expressivity of SigmoidAttn are used for societal benefit.

## C  UNIVERSAL APPROXIMATION PROPERTY FOR SIGMOID ATTENTION

This section is dedicated to the proof for the Universal Approximation Property for attention equipped with sigmoid nonlinearity. The proof follows closely the one provided in Yun et al. (2020, Sec. 3), of which we inherit much of the notation, and we encourage the interested reader to refer to the original source for a more comprehensive understanding of its details. Here we first provide context by outlining the main steps in the original proof, before proceeding to adapt its key components to the SigmoidAttn case.

The proof aims at showing that a transformer network can approximate to arbitrary accuracy any continuous, permutation-equivariant function with compact support. The proof is constructive in nature, in that it explicitly defines the architecture (and particularly, the sequence of self-attention and feed-forward layers) that can approximate a given target function. To do so, it proceeds in steps (see Yun et al. (2020, Sec. 3.2)):

(1) prove that any continuous function with compact support can be approximated to arbitrary accuracy by a piecewise constant function

(2) prove that an aptly-constructed *modified* transformer network, (where the softmax nonlinearity is substituted with a hardmax nonlinearity), can exactly represent such piecewise constant function. This step is further divided into three sub-steps (see Yun et al. (2020, Sec. 4)):

 (a) prove that a series of feed-forward layers can quantize any input to a specific discretization grid in the compact domain

 (b) prove that a series of self-attention layers can implement a *contextual mapping* (see Yun et al. (2020, Def. 3.1))

 (c) prove that a series of feed-forward layers can map the output of the contextual mapping to the desired output of the target piecewise-constant approximation

(3) prove that a (classical) transformer network can approximate such modified transformer network to arbitrary accuracy

Fortunately, some of the steps outlined above do not rely on a specific nonlinear function being used within the attention mechanism, and can be directly reused in our proof, virtually unchanged. Notice however that Steps (2-b) and (3) are directly impacted by modifications to the attention layer, and hence require adaptation in our case. This is the focus of the next sections.

### C.1  PROOF OF STEP (3): SIGMOID TRANSFORMERS CAN APPROXIMATE MODIFIED SIGMOID TRANSFORMERS

In Yun et al. (2020), to implement contextual mappings, the authors rely on a *modified* version of transformers, for the sake of simplifying the analysis. In their modified version, the (row-wise) softmax operation is substituted with a (row-wise) hardmax operation. This substitution is valid

because a classical transformer can still be made arbitrarily close to such modified transformer, in light of the fact that

$$\text{softmax}(\lambda \boldsymbol{X}) \xrightarrow{\lambda \to \infty} \text{hardmax}(\boldsymbol{X}). \tag{10}$$

In our proof, we follow a similar strategy to define our modified sigmoid transformer (and in particular, its self-attention mechanism). We have that

$$\sigma(\lambda \boldsymbol{X}) \xrightarrow{\lambda \to \infty} H(\boldsymbol{X}), \tag{11}$$

where $\sigma(x) = (1 + e^{-x})^{-1}$ is the (elementwise) sigmoid function, while

$$H(x) = \begin{cases} 1 & x > 0 \\ \frac{1}{2} & x = 0 \\ 0 & x < 0 \end{cases} \tag{12}$$

denotes the (elementwise) Heaviside step function. This allows us to define our modified sigmoid self-attention layer, as follows.

**Definition C.1** (Modified sigmoid self-attention layer). Given an input $\boldsymbol{X} \in \mathbb{R}^{d \times n}$, the action of a modified sigmoid self-attention layer with shifts and a single one-dimensional head is defined as $\boldsymbol{X} \mapsto \boldsymbol{X} + \psi(\boldsymbol{X}; \boldsymbol{q}, \boldsymbol{b}_q, \boldsymbol{k}, \boldsymbol{b}_k, \boldsymbol{v}, \boldsymbol{o})$, where

$$\psi(\boldsymbol{X}; \boldsymbol{q}, \boldsymbol{b}_q, \boldsymbol{k}, \boldsymbol{b}_k, \boldsymbol{v}, \boldsymbol{o}) = \boldsymbol{o} \left( \boldsymbol{v}^T \boldsymbol{X} \right) H \left( \left( \boldsymbol{q}^T \boldsymbol{X} - \boldsymbol{b}_q^T \right)^T \left( \boldsymbol{k}^T \boldsymbol{X} - \boldsymbol{b}_k^T \right) \right) \tag{13}$$

with $\boldsymbol{q}, \boldsymbol{k}, \boldsymbol{v} \in \mathbb{R}^d$ representing the query, key, and value vectors, $\boldsymbol{b}_q, \boldsymbol{b}_k \in \mathbb{R}^n$ the corresponding query and key bias vectors, while $\boldsymbol{o} \in \mathbb{R}^d$ denotes the output vector.

Analogously to (10), (11) guarantees that sigmoid attention can approximate modified sigmoid attention by simply increasing the magnitude of its inner parameters.

Here and in the following, the length of the input sequence is denoted as $n$, while $d$ represents the dimensionality of the tokens. Notice that we are considering the input tensor $\boldsymbol{X} \in \mathbb{R}^{d \times n}$, (as opposed to $\in \mathbb{R}^{n \times d}$) to better align out notation with the one used in Yun et al. (2020).

## C.2 PROOF OF STEP (2-B): MODIFIED SIGMOID TRANSFORMERS CAN IMPLEMENT CONTEXTUAL MAPPINGS

The core of the proof consists in showing how, by opportunely combining the operations in (13), one can build an architecture capable of implementing a *contextual mapping*. For completeness, we report next the definition of such a map (see also Yun et al. (2020, Def. 3.1)).

**Definition C.2** (Contextual mapping). A map $\boldsymbol{q} : \mathbb{L} \to \mathbb{R}^n$ from a finite set $\mathbb{L} \subset \mathbb{R}^{d \times n}$ is said to be a *contextual mapping* if both the following conditions hold:

(i) $q_i(\boldsymbol{X}) \neq q_j(\boldsymbol{X}), \forall i \neq j$ and $\forall \boldsymbol{X} \in \mathbb{L}$

(ii) $q_i(\boldsymbol{X}) \neq q_j(\boldsymbol{X}'), \forall i, j$ and $\forall \boldsymbol{X}, \boldsymbol{X}' \in \mathbb{L}$, with $\boldsymbol{X} \neq \boldsymbol{X}'$

where $q_i(\boldsymbol{X})$ denotes the $i$-th component of $\boldsymbol{q}(\boldsymbol{X})$.

Namely, a contextual mapping is such that it transforms each token in an input sequence to a value depending *uniquely* on the *whole* sequence. By satisfying this property, we can ensure that any element of the quantization of the input domain (achieved by Step (2-a)) can be mapped to a unique identifying value (depending on the whole input) via a sequence of modified sigmoid self-attention layers. It is then up to the MLP (in Step (2-c)) to correctly map this value to the corresponding output value in the piece-wise constant approximation.

In particular, after defining a uniform discretization (characterized by the parameter $\delta$) of the unitary hypercube $[0, 1]^d \subset \mathbb{R}^d$, namely

$$\mathbb{G}_\delta := \{ \boldsymbol{g} : g_i \in \{0, \delta, 2\delta, \dots, 1 - \delta\}, \quad \forall i = 1 \dots d \}, \tag{14}$$

we consider as input a tensor $\boldsymbol{X}$ (composed of columns $\boldsymbol{X} = [\boldsymbol{x}_i]_{i=1}^n$) such that

$$\boldsymbol{X} \in \mathbb{L} := \{ \boldsymbol{X} : \boldsymbol{x}_i \in \mathbb{G}_\delta \; \forall i = 1 \dots n, \quad \text{and} \quad \boldsymbol{x}_i \neq \boldsymbol{x}_j \; \forall i \neq j \} \subset \mathbb{R}^{d \times n}, \tag{15}$$

that is, a 2D tensor whose columns are element of the discretization $\mathbb{G}_\delta$, and that all differ from each other (at least for one element). We want to build a contextual mapping acting on $\mathbb{L}$, by stacking layers parameterized according to Def. C.1. In App. C.2.1 we define the basic building blocks of our architecture; in App. C.2.2 we describe how to stack them, and the effect the architecture has on a given input; finally, in App. C.2.4 we prove that this architecture indeed implements a contextual mapping.

### C.2.1 BASIC BUILDING BLOCKS OF CONTEXTUAL MAPPING

The strategy we follow to assemble a contextual mapping consists in sequentially looking at each column of the input, progressively updating and storing information regarding its content in a uniquely identifiable manner, and finally broadcasting this information back to every element in the sequence. The difficulty lies in the fact that each of these updates must be carried on while relying solely on applications of the modified $\mathrm{SigmoidAttn}$ layer in Def. C.1. In the following, we describe how we can tweak its parameters to achieve exactly this.

**From $d$-dimensional quantized vectors to scalars** As a first simplification, we can get rid of the $d$-dimension in the $\boldsymbol{X}$ tensor by mapping each of its columns to a corresponding identifying scalar, uniquely defined by the specific column components. This step is also performed in Yun et al. (2020, App. B.5), and can be achieved rather straightforwardly, by defining

$$\boldsymbol{v} \equiv \boldsymbol{q} \equiv \boldsymbol{k} \equiv \boldsymbol{u} := [1, \delta^{-1}, \delta^{-2}, \ldots, \delta^{-d+1}]^T. \tag{16}$$

Notice in fact that, since each column $\boldsymbol{x}_i$ belongs to $\mathbb{G}_\delta$, it can equivalently be written in the form $\boldsymbol{x}_i = \delta \cdot [\mathrm{id}_{0,i}, \mathrm{id}_{1,i}, \ldots, \mathrm{id}_{d-1,i}]^T$, where $\mathrm{id}_{j,i} \in \{0, 1, 2, \ldots, \delta^{-1} - 1\}$ represents the (indexed) coordinate of the discretization along the $j$-th dimension. Scalar-multiplying $\boldsymbol{X}$ by $\boldsymbol{u}$ in (16), then, turns this tuple of indices into a single one, in a bijective fashion[6].

This allows us to equivalently consider a single vector $\boldsymbol{u}^T \boldsymbol{X} \in \mathbb{R}^n$, rather than the whole tensor $\boldsymbol{X} \in \mathbb{R}^{d \times n}$ in the remainder of our analysis. Analogously, choosing $\boldsymbol{o} \equiv \boldsymbol{e}_0 := [1, 0, \ldots, 0]^T$ in (13) constraints the effect of the layer application to impact only the first row of the tensor: the goal is then to store in this row the result of the target contextual mapping $\boldsymbol{q}$ in Def. C.2. To slim our notation, in the following we often refer to $\boldsymbol{u}^T \boldsymbol{X}$ as the vector $\boldsymbol{l} \in \mathbb{R}^n$, with components $l_i$.

In light of the simplification above, we can rewrite (13) more compactly, as follows:

$$\psi(\boldsymbol{X}; \boldsymbol{q} = \boldsymbol{k} = \boldsymbol{v} \equiv \boldsymbol{u}, \boldsymbol{o} \equiv \boldsymbol{e}_0; \boldsymbol{b}_q, \boldsymbol{b}_k) = \boldsymbol{e}_0 \boldsymbol{l}^T H\left((\boldsymbol{l} - \boldsymbol{b}_q) \otimes (\boldsymbol{l} - \boldsymbol{b}_k)\right) \tag{17}$$

Notice that, since the elements of both $\boldsymbol{X}$ and $\boldsymbol{u}$ are always non-negative, so are those of $\boldsymbol{l}$, too. Moreover, since we are interested in permutation-equivariant functions with respect to the columns of $\boldsymbol{X}$, without loss of generality we can consider the elements of $\boldsymbol{l} = \boldsymbol{u}^T \boldsymbol{X}$ to be ordered: $0 \le l_i < l_j$, $\forall i < j$.

**Selective shift operation for sigmoid attention** Since we aim to recover a contextual map by sequentially updating the elements of $\boldsymbol{l}$, we proceed by designing a modification of (17) which affects only a certain selected element at a time. This is were our second simplification comes into play, and this time it pertains the roles of the bias vectors $\boldsymbol{b}_q$ and $\boldsymbol{b}_k$. Since $\boldsymbol{l} \ge 0$, these vectors have the effect of tweaking the sign of the inner arguments of the Heaviside function in (17), hence directly impacting when its application outputs 0 or 1. By aptly selecting the values of $\boldsymbol{b}_k$ and $\boldsymbol{b}_q$, then, we can explicitly decide when a specific layer triggers an update, which elements are affected by the update, and what elements to consider to compute the update itself.

More in detail, take $\boldsymbol{b}_q = \mathbf{1}b_q$ and $\boldsymbol{b}_v = \mathbf{1}b_v$, for some scalars $b_q, b_v$, and with $\mathbf{1}$ being the all-one vector. Plugging this into (17), we have

$$\tilde{\psi}(\boldsymbol{X}; b_q, b_k) := \psi(\boldsymbol{X}; \boldsymbol{q} = \boldsymbol{k} = \boldsymbol{v} \equiv \boldsymbol{u}, \boldsymbol{o} \equiv \boldsymbol{e}_0, \boldsymbol{b}_q = \mathbf{1}b_q, \boldsymbol{b}_k = \mathbf{1}b_k)$$

$$= \boldsymbol{e}_0 \boldsymbol{l}^T H\left((\boldsymbol{l} - \mathbf{1}b_q) \otimes (\boldsymbol{l} - \mathbf{1}b_k)\right) = \boldsymbol{e}_0 \begin{cases} \sum_{i: l_i < b_v} l_i & \text{if } l_j < b_k \\ \sum_{i: l_i > b_v} l_i & \text{if } l_j > b_k \end{cases}; \tag{18}$$

---

[6]For example, consider $d = 3$ and the column defined as $\boldsymbol{x}_i = [3\delta, 10\delta, 2\delta]^T$, that is, the column identified by the *triplet* of indices $[3, 10, 2]$. Multiplying by $\boldsymbol{u}$ would then give the scalar $\boldsymbol{u}^T \boldsymbol{x}_i = (3 + 10N + 2N^2)\delta$, where $N = \delta^{-1}$, which is uniquely identified by the *single* index $(3 + 10N + 2N^2)$.

notice how $b_q$ determines what elements of $\boldsymbol{l}$ compose the update (as it impacts the indices considered in the sum), while $b_k$ defines the elements impacted by the update itself [7]. If we opportunely combine *four* modified sigmoid self-attention heads $\tilde{\psi}(\boldsymbol{X}; b_q, b_k)$, we recover, for a given index $i = 0 \ldots \delta^{-d} - 1$,

$$
\begin{aligned}
\Psi^{(i)}(\boldsymbol{X}) :=& \boldsymbol{X} + \frac{1}{2} c \begin{pmatrix} \tilde{\psi}\left(\boldsymbol{X}; b_q = 0, b_k = \left(i - \frac{1}{2}\right)\delta\right) \\ -\tilde{\psi}\left(\boldsymbol{X}; b_q = 0, b_k = \left(i + \frac{1}{2}\right)\delta\right) \\ -\tilde{\psi}\left(\boldsymbol{X}; b_q = b_k = \left(i + \frac{1}{2}\right)\delta\right) \\ +\tilde{\psi}\left(\boldsymbol{X}; b_q = \left(i + \frac{1}{2}\right), b_k = \left(i - \frac{1}{2}\right)\delta\right) \end{pmatrix} \\
=& \boldsymbol{X} + \frac{1}{2} c e_0 \boldsymbol{l}^T \begin{pmatrix} H\left(\boldsymbol{l} \otimes \left(\boldsymbol{l} - \left(i - \frac{1}{2}\right)\delta\right)\right) \\ -H\left(\boldsymbol{l} \otimes \left(\boldsymbol{l} - \left(i + \frac{1}{2}\right)\delta\right)\right) \\ -H\left(\left(\boldsymbol{l} - \left(i + \frac{1}{2}\right)\delta\right) \otimes \left(\boldsymbol{l} - \left(i + \frac{1}{2}\right)\delta\right)\right) \\ +H\left(\left(\boldsymbol{l} - \left(i + \frac{1}{2}\right)\delta\right) \otimes \left(\boldsymbol{l} - \left(i - \frac{1}{2}\right)\delta\right)\right) \end{pmatrix} \\
\Longrightarrow& \Psi^{(i)}_{1,j}(\boldsymbol{X}) = \boldsymbol{X}_{1,j} + c \begin{cases} \sum_{k: l_k > i\delta} l_k & \text{if } l_j = i\delta \\ 0 & \text{otherwise} \end{cases} \\
\Longrightarrow& \Psi^{(i)}_{k>1,j}(\boldsymbol{X}) = \boldsymbol{X}_{k,j},
\end{aligned}
\tag{22}
$$

where $c \equiv c(\delta, d, n)$ is a multiplicative constant which will be chosen later.

The operator assembled in (22) defines the basic layer of the architecture that we use in our proof. Notice $\Psi^{(i)}(\boldsymbol{X})$ has the effect of modifying only the column $\boldsymbol{x}_j$ which has index $l_j = \boldsymbol{u}^T \boldsymbol{x}_j = i\delta$ (if at all present in the input $\boldsymbol{X}$). This layer covers a similar role to the *selective shift operation* introduced in Yun et al. (2020, App. B.5), but it has been adapted to account for the presence of a sigmoid nonlinearity: notice this required us to use 4-headed attention, while in Yun et al. (2020) a 2-headed version is sufficient.

### C.2.2 RESULT OF APPLYING A SEQUENCE OF SELECTIVE SHIFTS

Ultimately we want to show how, by stacking a sequence of selective shift layers (22) for increasing $i = 0 \ldots \delta^{-d} - 1$ and one additional global shift, we can build an architecture capable of representing

---

[7] This can be better seen by considering independently the effects of the two parameters $b_k$, $b_q$ on the modified sigmoid attention matrix $H\left((\boldsymbol{l} - \mathbf{1}b_q) \otimes (\boldsymbol{l} - \mathbf{1}b_k)\right)$. We have in fact, with $b_q = 0$,

$$
H\left(\boldsymbol{l} \otimes (\boldsymbol{l} - \mathbf{1}b_k)\right) = \begin{array}{c} \begin{matrix} l_j < b_k \quad l_j > b_k \end{matrix} \\ \begin{bmatrix} 0 & \cdots & 0 & 1 & \cdots & 1 \\ \vdots & \ddots & \vdots & \vdots & \ddots & \vdots \\ 0 & \cdots & 0 & 1 & \cdots & 1 \end{bmatrix} \end{array}.
\tag{19}
$$

This shows how, by modifying $b_k$, one can decide which columns will receive an update: namely, all those with index $l_j > b_k$. By combining two such operators with $b_k = \left(i - \frac{1}{2}\right)\delta$ and $b_k = \left(i + \frac{1}{2}\right)\delta$, we then recover

$$
\begin{aligned} H\left(\boldsymbol{l} \otimes \left(\boldsymbol{l} - \mathbf{1}\left(i - \frac{1}{2}\right)\delta\right)\right) \\ -H\left(\boldsymbol{l} \otimes \left(\boldsymbol{l} - \mathbf{1}\left(i + \frac{1}{2}\right)\delta\right)\right) \end{aligned} = \begin{array}{c} \begin{matrix} l_j = i\delta \end{matrix} \\ \begin{bmatrix} 0 & \cdots & 0 & 1 & 0 & \cdots & 0 \\ \vdots & \ddots & \vdots & \vdots & \vdots & \ddots & \vdots \\ 0 & \cdots & 0 & 1 & 0 & \cdots & 0 \end{bmatrix} \end{array},
\tag{20}
$$

which allows us to limit the update to only one specific column: the one with index $l_j = i\delta$.

The parameter $b_q$ acts analogously, but varies the output of the Heaviside function as we move down the rows, rather than the columns. The same operator as in (20), but with $b_q = \left(i + \frac{1}{2}\right)\delta$ gives us in fact:

$$
\begin{aligned} H\left(\left(\boldsymbol{l} - \mathbf{1}\left(i + \frac{1}{2}\right)\delta\right) \otimes \left(\boldsymbol{l} - \mathbf{1}\left(i - \frac{1}{2}\right)\delta\right)\right) \\ -H\left(\left(\boldsymbol{l} - \mathbf{1}\left(i + \frac{1}{2}\right)\delta\right) \otimes \left(\boldsymbol{l} - \mathbf{1}\left(i + \frac{1}{2}\right)\delta\right)\right) \end{aligned} = \begin{array}{cc} \begin{matrix} l_j = i\delta \end{matrix} & \\ \begin{bmatrix} 0 & \cdots & 0 & -1 & 0 & \cdots & 0 \\ \vdots & \ddots & \vdots & \vdots & \vdots & \ddots & \vdots \\ 0 & \cdots & 0 & -1 & 0 & \cdots & 0 \\ 0 & \cdots & 0 & 1 & 0 & \cdots & 0 \\ \vdots & \ddots & \vdots & \vdots & \vdots & \ddots & \vdots \\ 0 & \cdots & 0 & 1 & 0 & \cdots & 0 \end{bmatrix} & \begin{matrix} l_j < i\delta \\ \\ \\ l_j > i\delta \end{matrix} \end{array}.
\tag{21}
$$

Finally, (22) can be recovered by combining (20) and (21): this has the effect of removing the $-1$'s in (21).

a contextual mapping. As a preliminary step, in this section we provide an explicit formula for the result of applying such an architecture. Once again, we are proceeding analogously to Yun et al. (2020, App. B.5.1).

**After the first selective shift application** Consider a quantized input sequence $\boldsymbol{X} \in \mathbb{L}$ as defined in (15), with its columns ordered according to their scalar indices $\boldsymbol{l} = \boldsymbol{u}^T \boldsymbol{X}$. The sequence of selective shift layers $\Psi^{(0)}, \Psi^{(1)}, \dots$ initially has no effect on the input itself, and it leaves it unchanged until we hit the layer corresponding to the index of the first column in the input, $\Psi^{(\hat{i})}$, where $l_1 = \boldsymbol{u}^T \boldsymbol{x}_1 = \hat{i}\delta$. At this point, following (22), the first column of the input is modified into

$$\boldsymbol{x}_1 \quad \mapsto \quad \Psi^{(\hat{i})}_{|,1}(\boldsymbol{X}) = \boldsymbol{x}_1 + c\boldsymbol{e}_0 \sum_{k:l_k > l_1} l_k = \boldsymbol{x}_1 + c\boldsymbol{e}_0 \left( \sum_{k=1}^{n} l_k - l_1 \right) \tag{23}$$

while the other columns are still left untouched. In the following, we compactly refer to the quantities $\sum_{k=1}^{n} l_k - l_i$ as $s_i$:

$$\boldsymbol{s} = [s_1, s_2, \dots, s_n]^T := \left[ \sum_{k=1}^{n} l_k - l_1, \sum_{k=1}^{n} l_k - l_2, \dots, \sum_{k=1}^{n} l_k - l_n \right]^T . \tag{24}$$

According to (23), the index $l_1$ of column $\boldsymbol{x}_1$ is then analogously mapped to

$$l_1 = \boldsymbol{u}^T \boldsymbol{x}_1 \quad \mapsto \quad \tilde{l}_1 := \boldsymbol{u}^T \Psi^{(\hat{i})}_{|,1}(\boldsymbol{X}) = \boldsymbol{u}^T \boldsymbol{x}_1 + cs_1 = l_1 + cs_1. \tag{25}$$

Notice that, by choosing $c > 1$, we can ensure

$$c > 1 \quad \implies \quad \tilde{l}_1 > \cancel{l_1} + \sum_{k=1}^{n} l_k - \cancel{l_1} > \sum_{k=1}^{n} > l_i \quad \forall i, \tag{26}$$

and particularly $\tilde{l}_1 > l_2$, implying that at the next (effective) application of the selective shift operation, this term, too, will contribute to the update.

**Subsequent selective shift applications** Following similar considerations, the next effective update will be applied by the layer $\Psi^{(\hat{i})}$ with $l_2 = \boldsymbol{u}^T \boldsymbol{x}_2 = \hat{i}\delta$. At this point, the second column index is updated as follows:

$$
\begin{aligned}
l_2 = \boldsymbol{u}^T \boldsymbol{x}_2 \quad &\mapsto \quad \tilde{l}_2 := \boldsymbol{u}^T \Psi^{(\hat{i})}_{|,2}(\boldsymbol{X}) = \boldsymbol{u}^T \boldsymbol{x}_2 + c \left( \sum_{k:l_k > l_2} l_k + \tilde{l}_1 \right) \\
&= l_2 + c \left( \sum_{k=1}^{n} l_k - l_2 - \cancel{l_1} + \cancel{l_1} + cs_1 \right) = l_2 + cs_2 + c^2 s_1
\end{aligned}
\tag{27}
$$

where $\tilde{l}_1$ is also included in light of (26), and we used the definitions (24) and (25). Continuing to apply $\Psi^{(i)}(\boldsymbol{X})$, for increasing $i$, and unrolling the recursion, we recover

$$
\begin{aligned}
\tilde{l}_3 &= l_3 + c \left( \sum_{k=1}^{n} l_k - l_1 - l_2 - l_3 + \tilde{l}_1 + \tilde{l}_2 \right) = l_3 + cs_3 + c^2(s_2 + s_1) + c^3 s_1 \\
\tilde{l}_4 &= l_4 + c \left( \sum_{k=1}^{n} l_k - l_1 - l_2 - l_3 - l_4 + \tilde{l}_1 + \tilde{l}_2 + \tilde{l}_3 \right) \\
&= l_4 + cs_4 + c^2(s_3 + s_2 + s_1) + c^3(s_2 + 2s_1) + c^4 s_1 \\
\tilde{l}_5 &= l_5 + c \left( \sum_{k=1}^{n} l_k - l_1 - l_2 - l_3 - l_4 - l_5 + \tilde{l}_1 + \tilde{l}_2 + \tilde{l}_3 + \tilde{l}_4 \right) \\
&= l_5 + cs_5 + c^2(s_4 + s_3 + s_2 + s_1) + c^3(s_3 + 2s_2 + 3s_1) + c^4(s_2 + 3s_1) + c^5 s_1 \\
&\vdots
\end{aligned}
\tag{28}
$$

which eventually allows us to write the general formula [8]

$$\tilde{l}_j := l_j + cs_j + \sum_{i=0}^{j-2} c^{i+2} \sum_{k=i}^{j-2} \binom{k}{i} s_{k-i+1}, \qquad j = 1 \dots n. \qquad (29)$$

### C.2.3 RESULT OF APPLYING ONE LAST *Global Shift* LAYER

After the last selective shift layer, the original input $\boldsymbol{X}$ has been mapped to a modified one $\tilde{\boldsymbol{X}}$ whereby each column $\tilde{\boldsymbol{x}}_j$ is characterized by the index $\tilde{l}_j = \boldsymbol{u}^T \tilde{\boldsymbol{x}}_j$ given in (29). Remember our goal is to recover a contextual mapping, but notice that these $\tilde{l}_j$ indices are *not* uniquely defined by the input[9]; in other words, they do not satisfy property (2) in Def. C.2. The only exception to this is the last index $\tilde{l}_n$, as (loosely speaking) it has "seen" all the previous updates - and indeed in App. C.2.4 we prove this rigorously, under some assumption on the yet-undefined coefficient $c(\delta, d, n)$.

A straightforward way to recover a one-to-one mapping for the whole sequence, then, is to update every index $\tilde{l}_j$ via a quantity directly depending on $\tilde{l}_n$. This is precisely what the last *global shift* layer $\bar{\Psi}(\boldsymbol{X})$ aims to accomplish. This last layer is also defined starting from the simplified modified sigmoid attention (18), by picking $b_k = 0$ and $b_q = \left(c(\delta, d, n)^n + \frac{1}{2}\right)\delta$: if, for any input, we can guarantee that

$$\tilde{l}_j \le c(\delta, d, n)^n \delta \quad j < n \qquad \text{and} \qquad \tilde{l}_n > c(\delta, d, n)^n \delta, \qquad (30)$$

then the application of the global shift layer would result in[10]:

$$\begin{aligned}
\bar{\Psi}(\tilde{\boldsymbol{X}}) :=& \tilde{\boldsymbol{X}} + c^{n+1} \tilde{\psi}\left(\tilde{\boldsymbol{X}}; b_q = \left(c^n + \frac{1}{2}\right)\delta, b_k = 0\right) \\
\Longrightarrow& \bar{\Psi}_{1,j}(\tilde{\boldsymbol{X}}) = \tilde{\boldsymbol{X}}_{1,j} + c^{n+1} \tilde{l}_n \\
\Longrightarrow& \bar{\Psi}_{k>1,j}(\tilde{\boldsymbol{X}}) = \tilde{\boldsymbol{X}}_{k,j}.
\end{aligned} \qquad (32)$$

The global shift (32) is the last layer we need to define our candidate contextual mapping. Collecting the results from this section together, our architecture is defined by sequentially composing the selective shift layers with the global shift one,

$$\Psi(\boldsymbol{X}) := \bar{\Psi} \circ \Psi^{(\delta^{-d}-1)} \circ \cdots \circ \Psi^{(2)} \circ \Psi^{(1)}(\boldsymbol{X}). \qquad (33)$$

After being scalar-multiplied by $\boldsymbol{u}$, this results in a sequence

$$q(\boldsymbol{X}) := \boldsymbol{u}^T \Psi(\boldsymbol{X}) = \tilde{l} + c^{n+1} \mathbf{1} \tilde{l}_n \qquad (34)$$

which we aim to prove is a contextual mapping. This is shown in the next section.

---

[8]From (28), we can notice that, for a given $\tilde{l}_k$, the coefficients $a_{i,j}^{(k)}$ appearing in front of the various $s_{k-i}$ for each of the $c^j$ terms, are first given by a list of ones, $a_{i,1}^{(k)} = 1$, then a list of increasing numbers $a_{i,2}^{(k)} = i \Longrightarrow a_{-,2}^{(k)} = \text{cumsum}(a_{-,1}^{(k)})$, then a list of triangular numbers $a_{i,3}^{(k)} = i(i+1)/2 \Longrightarrow a_{-,3}^{(k)} = \text{cumsum}(a_{-,2}^{(k)})$, and so on: $a_{-,j}^{(k)} = \text{cumsum}(a_{-,j-1}^{(k)})$. The result of iterated applications of cumsum, starting from an all-one vector, can be compactly described via the binomial coefficient: we have in fact

$$a_{i,j} = [\text{cumsum}^j([1, 1, \dots])]_i = \binom{i+j-2}{j-1}.$$

The actual formula (29) can be recovered after a few algebraic steps, by rearranging the summation indices.

[9]To convince ourselves of this, it suffices to look at the formula for (25): two sequences with different elements $\boldsymbol{l} \ne \boldsymbol{l}'$, but such that $l_1 = l_1'$ and $s_1 = s_1'$ (that is, with $\sum_{i=1}^n l_i = \sum_{i=1}^n l_i'$) would map to the same $\tilde{l}_1 = \tilde{l}_1'$.

[10]As in footnote 7, this is also better seen by considering the resulting modified sigmoid attention matrix. With $b_k = 0$ and $b_q = \left(c(\delta, d, n)^n + \frac{1}{2}\right)\delta$, in fact, if condition (30) is verified, this matrix is given by

$$H\left(\left(\tilde{l} - \mathbf{1}\left(c^n + \frac{1}{2}\right)\delta\right) \otimes \tilde{l}\right) = \begin{bmatrix} 0 & \cdots & 0 \\ \vdots & \ddots & \vdots \\ 0 & \cdots & 0 \\ \hline 1 & \cdots & 1 \end{bmatrix} \begin{matrix} \tilde{l}_j, j < n \\ \\ \tilde{l}_n \end{matrix}. \qquad (31)$$

### C.2.4 A SEQUENCE OF SELECTIVE SHIFTS FOLLOWED BY A GLOBAL SHIFT PRODUCES A CONTEXTUAL MAPPING

To complete the proof, it remains to show that the recovered sequence (34) represents a contextual mapping and, in particular, that it is *(i)* one-to-one in $\mathbb{L}$, and that *(ii)* all of its elements are distinct for different inputs. To do so, we need a few preparatory lemmas. The first few are needed to show that each of the basic components of (34) is indeed a one-to-one map.

**Lemma C.3.** *The map $\boldsymbol{l} \mapsto \boldsymbol{s}$ in (24) is one-to-one.*

*Proof.* The target map can be compactly represented as a linear operator $S$:

$$\boldsymbol{l} \mapsto \boldsymbol{s} := \mathbf{1} \sum_{k=1}^{n} l_k - \boldsymbol{l} = (\mathbf{1} \otimes \mathbf{1} - I)\boldsymbol{l} =: S\boldsymbol{l} \tag{35}$$

which is invertible[11], denoting that $\boldsymbol{l} \mapsto \boldsymbol{s}$ is bijective. $\square$

**Lemma C.4.** *The map $\boldsymbol{l} \mapsto \tilde{l}_n$ in (29) is one-to-one, under the condition*

$$c(\delta, d, n) > (n-1)(\delta^{-d} - 1)\binom{n-1}{\lceil \frac{n-1}{2} \rceil}. \tag{36}$$

*Proof.* Consider two vectors of column indices $\boldsymbol{l}, \boldsymbol{l}'$ differing for at least one element. We have by definition (29) that

$$\tilde{l}_n - \tilde{l}'_n = (l_n - l'_n) + c(s_n - s'_n) + \sum_{i=0}^{n-2} c^{i+2} \sum_{k=i}^{n-2} \binom{k}{i}(s_{k-i+1} - s'_{k-i+1}) \tag{37}$$

By absurd, assume $\tilde{l}_n - \tilde{l}'_n = 0$ even though $\exists i : l_i \neq l'_i$. We have then that it must hold

$$
\begin{aligned}
(l'_n - l_n) &= c(s_n - s'_n) + \sum_{i=0}^{n-2} c^{i+2} \sum_{k=i}^{n-2} \binom{k}{i}(s_{k-i+1} - s'_{k-i+1}) \\
&= c\left((s_n - s'_n) + \sum_{i=0}^{n-2} c^{i+1} \sum_{k=i}^{n-2} \binom{k}{i}(s_{k-i+1} - s'_{k-i+1})\right)
\end{aligned}
\tag{38}
$$

Notice that, for $c(\delta, d, n)$ large enough, the right-hand side does not have enough *granularity* to counter the left-hand side: in fact, since $l_n \in \{0, \delta, 2\delta, \ldots, \delta^{-d+1} - \delta\}$, the left-hand side can attain values

$$l'_n - l_n \in \{0, \pm\delta, \pm 2\delta, \ldots, \pm(\delta^{-d+1} - \delta)\} \tag{39}$$

while the former, in light of the presence of the $c(\delta, d, n)$ factor, can only attain values $\in \{0, \pm c\delta, \pm 2c\delta, \ldots\}$. Picking $c > \delta^{-d} - 1$, then, ensures that equality between the two sides of (38) can only be achieved if they are both 0. In this case, we need to impose

$$
\begin{aligned}
c(s'_n - s_n) &= \sum_{i=0}^{n-2} c^{i+1} \sum_{k=i}^{n-2} \binom{k}{i}(s_{k-i+1} - s'_{k-i+1}) \\
\iff s'_n - s_n &= c\left(\sum_{i=0}^{n-2} c^{i} \sum_{k=i}^{n-2} \binom{k}{i}(s_{k-i+1} - s'_{k-i+1})\right).
\end{aligned}
\tag{40}
$$

Similarly, notice that[12], $\forall i$,

$$|s_i - s'_i| = \left| \sum_{k=1}^{n}(l_k - l'_k) - (l_i - l'_i) \right| = \left| \sum_{k=1, k \neq i}^{n}(l_k - l'_k) \right| < (n-1)(\delta^{-d+1} - \delta), \tag{41}$$

---

[11]Indeed its inverse can be explicitly recovered by directly applying Sherman-Morrison formula.

[12]This is a direct consequence of the definition of operator $S$ in (35): since it has 1's everywhere but on its diagonal, its $\infty$-norm is simply $n-1$.

implying that $s_n' - s_n \in \{0, \pm\delta, \pm 2\delta, \ldots, \pm(n-1)(\delta^{-d} - 1)\delta\}$. Again, by picking $c(\delta, d, n) > (n-1)(\delta^{-d} - 1)$ we ensure that the right-hand side does not have enough granularity, and hence

$$c(\delta, d, n) > (n-1)(\delta^{-d} - 1) \qquad \Longrightarrow \qquad s_n' - s_n = 0, \tag{42}$$

implying

$$c\left(\sum_{i=0}^{n-2} c^i \sum_{k=i}^{n-2} \binom{k}{i}(s_{k-i+1} - s_{k-i+1}')\right) = 0$$

$$\Longleftrightarrow \quad \sum_{k=0}^{n-2}\binom{k}{0}(s_{k+1}' - s_{k+1}) = c\left(\sum_{i=1}^{n-2} c^{i-1} \sum_{k=i}^{n-2} \binom{k}{i}(s_{k-i+1} - s_{k-i+1}')\right) \tag{43}$$

$$\Longleftrightarrow \quad \sum_{k=0}^{n-2}(s_{k+1}' - s_{k+1}) = c\left(\sum_{i=1}^{n-2} c^{i-1} \sum_{k=i}^{n-2} \binom{k}{i}(s_{k-i+1} - s_{k-i+1}')\right).$$

Following a similar reasoning as the one applied above shows us that picking

$$c(\delta, d, n) > (n-1)^2(\delta^{-d} - 1) \qquad \Longrightarrow \qquad \sum_{k=0}^{n-2}(s_{k+1} - s_{k+1}') = 0, \tag{44}$$

and requires us to satisfy

$$c\left(\sum_{i=1}^{n-2} c^{i-1} \sum_{k=i}^{n-2} \binom{k}{i}(s_{k-i+1} - s_{k-i+1}')\right) = 0$$

$$\Longleftrightarrow \quad \sum_{k=1}^{n-2}\binom{k}{1}(s_k' - s_k) = c\left(\sum_{i=2}^{n-2} c^{i-2} \sum_{k=i}^{n-2} \binom{k}{i}(s_{k-i+1} - s_{k-i+1}')\right) \tag{45}$$

$$\Longleftrightarrow \quad \sum_{k=1}^{n-2} k(s_k' - s_k) = c\left(\sum_{i=2}^{n-2} c^{i-2} \sum_{k=i}^{n-2} \binom{k}{i}(s_{k-i+1} - s_{k-i+1}')\right).$$

Once again, then, by choosing

$$c(\delta, d, n) > \frac{(n-2)(n-1)^2}{2}(\delta^{-d} - 1) \qquad \Longrightarrow \qquad \sum_{k=1}^{n-2} k(s_k - s_k') = 0. \tag{46}$$

This reasoning can be repeated recursively: at each step $i$ of the recursion, by imposing a stricter and stricter bound on $c(\delta, d, n)$ we gain more and more conditions that the quantity $s' - s$ needs to satisfy:

$$c(\delta, d, n) > (n-1)(\delta^{-d} - 1)\sum_{k=i}^{n-2}\binom{k}{i} \qquad \Longrightarrow \qquad \sum_{k=i}^{n-2}\binom{k}{i}(s_{k-i+1} - s_{k-1+1}') = 0. \tag{47}$$

Notice that, every time we increase $i = 0 \ldots n-2$, these conditions involve one less term $s_{k-i+1} - s_{k-i+1}'$, $k = i \ldots n-2$: if we were to collect all these conditions within a single linear system, the system would have an upper-triangular structure, and hence be non-singular. This implies that for the set of $n$ independent conditions on $s - s'$ to hold (we have $n-1$ in (47), plus one more in (42)), the only possibility is that $s \equiv s'$. Because of Lemma C.3, though, this also implies $l \equiv l'$: we have finally reached a contradiction, and proven that indeed $l \mapsto \tilde{l}_n$ is one-to-one, under an opportune condition on $c(\delta, d, n)$. Such condition can be promptly recovered[13] by (47):

$$\max_{i=0\ldots n-2} \sum_{k=i}^{n-2}\binom{k}{i} = \max_{i=0\ldots n-2}\binom{n-1}{i+1} = \binom{n-1}{\lceil\frac{n-1}{2}\rceil}. \tag{48}$$

Substituting this in (47), we recover that it suffices to impose

$$c(\delta, d, n) > (n-1)(\delta^{-d} - 1)\binom{n-1}{\lceil\frac{n-1}{2}\rceil}. \tag{49}$$

$\square$

---

[13]This is a consequence of some useful properties of the binomial coefficient, namely the Hockey stick identity Jones (1994), and the symmetry of $\binom{k}{i}$ with respect to $i$.

The next few lemmas are needed to bound the elements in the $\tilde{l}_j$ sequence, which in turn are used to prove property *(ii)* in Def. C.2.

**Lemma C.5.** *$\tilde{l}_j$ in (29) is an increasing sequence.*

*Proof.* This can be proven directly: we have in fact, by definition (29),

$$
\begin{aligned}
\tilde{l}_j > \tilde{l}_{j-1} \quad &\Longleftrightarrow \quad l_j + cs_j + \sum_{i=0}^{j-2} c^{i+2} \sum_{k=i}^{j-2} \binom{k}{i} s_{k-i+1} \\
&\qquad\qquad > l_{j-1} + cs_{j-1} + \sum_{i=0}^{j-3} c^{i+2} \sum_{k=i}^{j-3} \binom{k}{i} s_{k-i+1} \\
\text{combine sums} \quad &\Longleftrightarrow \quad (l_j - l_{j-1})(1-c) + \sum_{i=0}^{j-2} c^{i+2} \binom{j-2}{i} s_{j-1-i} > 0 \\
\binom{j-2}{i} \geq 1,\, c^{i+2} \geq c^2 \quad &\Longleftarrow \quad (l_j - l_{j-1})(1-c) + c^2 \sum_{i=0}^{j-2} s_{j-1-i} > 0 \\
(24) \quad &\Longleftrightarrow \quad (l_j - l_{j-1})(1-c) + c^2 \sum_{i=0}^{j-2} \left( \sum_{k=1}^{n} l_k - l_{j-1-i} \right) > 0 \\
&\Longleftrightarrow \quad (l_j - l_{j-1})(1-c) + c^2 \left( (j-1) \sum_{k=1}^{n} l_k - \sum_{k=1}^{j-1} l_k \right) > 0 \\
&\Longleftrightarrow \quad (1-c)l_j + (c-1)l_{j-1} + c^2(j-2) \sum_{k=1}^{n} l_k + c^2 \sum_{k=j}^{n} l_k > 0 \\
&\Longleftrightarrow \quad (c^2 - c + 1)l_j + (c-1)l_{j-1} + c^2(j-2) \sum_{k=1}^{n} l_k + c^2 \sum_{k=j+1}^{n} l_k > 0
\end{aligned}
$$
(50)

Already with $c > 1$, all the coefficients are positive (and at least one is non-zero), implying that the condition above is always satisfied and that indeed $\tilde{l}_j$ is an increasing sequence. $\qquad\square$

**Lemma C.6.** *Under constraint (36), each term $\tilde{l}_j$, $j > 1$ in (29) is bounded from below by*

$$\tilde{l}_j > c^j \delta,$$

*and each term $\tilde{l}_j$, $1 < j < n$ is bounded from above by*

$$\tilde{l}_j < c^{j+1} \delta.$$

*Proof.* We start by proving the lower bound. By definition (29), we have

$$
\tilde{l}_j = l_j + cs_j + \sum_{i=0}^{j-2} c^{i+2} \sum_{k=i}^{j-2} \binom{k}{i} s_{k-i+1} = l_j + cs_j + c^j s_1 + \sum_{i=0}^{j-3} c^{i+2} \sum_{k=i}^{j-2} \binom{k}{i} s_{k-i+1}. \quad (51)
$$

Since by assumption $l_j$ is an ordered sequence without repetitions, for $j > 1$ we necessarily have $l_j > l_1 \geq 0$, and hence $l_j \geq \delta$. All the other terms in (51) are non-negative, so we can safely claim that

$$\tilde{l}_j \geq \delta + c^j \delta > c^j \delta \qquad \forall j > 1, \quad (52)$$

which confirms the lower bound.

For the upper bound, we start again from the definition of $\tilde{l}_j$:

$$\tilde{l}_j = l_j + cs_j + \sum_{i=0}^{j-2} c^{i+2} \sum_{k=i}^{j-2} \binom{k}{i} s_{k-i+1}$$

$$< (\delta^{-d} - 1)\delta + c(n-1)(\delta^{-d} - 1)\delta + s_1 \sum_{i=0}^{j-2} c^{i+2} \binom{j-1}{i+1} \tag{53}$$

$$\leq (n-1)(\delta^{-d} - 1)\binom{j-1}{\lceil \frac{j-1}{2} \rceil} \delta \sum_{i=0}^{j} c^i = (n-1)(\delta^{-d} - 1)\binom{j-1}{\lceil \frac{j-1}{2} \rceil} \delta \frac{1 - c^{j+1}}{1 - c},$$

where we used relationship (48) and collected all $c$ terms within the sum. Notice that, for a given $a > 1$ we have that

$$\frac{1 - c^{j+1}}{1 - c} \leq ac^j, \tag{54}$$

provided that $c \geq \frac{a}{a-1}$. In fact,

$$\frac{1 - c^{j+1}}{1 - c} \leq ac^j \iff \frac{1 - c^{j+1} - ac^j + ac^{j+1}}{1 - c} \leq 0$$

$$\Longleftarrow \frac{1}{a-1} + \left(c - \frac{a}{a-1}\right) c^j \geq 0 \iff \frac{1}{a-1} \geq 0 \tag{55}$$

which is always satisfied. After substituting (54) in (53), this allows us to write

$$\tilde{l}_j < a(n-1)(\delta^{-d} - 1)\binom{j-1}{\lceil \frac{j-1}{2} \rceil} \delta c^j. \tag{56}$$

To prove that $\tilde{l}_j < \delta c^{j+1}$, then, it remains to show that

$$c \geq a(n-1)(\delta^{-d} - 1)\binom{j-1}{\lceil \frac{j-1}{2} \rceil} \qquad \forall 1 < j < n. \tag{57}$$

Substituting condition (36) in the inequality above, we are left with proving

$$\binom{n-1}{\lceil \frac{n-1}{2} \rceil} \geq \max_{j=2\ldots n-1} a\binom{j-1}{\lceil \frac{j-1}{2} \rceil} = a\binom{n-2}{\lceil \frac{n-2}{2} \rceil}. \tag{58}$$

The outcome depends on the parity of $n$. For $n$ odd, we have

$$\binom{n-1}{\lceil \frac{n-1}{2} \rceil} \geq a\binom{n-2}{\lceil \frac{n-2}{2} \rceil} \qquad \Longleftrightarrow \qquad 2\frac{n-1}{n-1} \geq a, \tag{59}$$

to satisfy which it suffices to pick $a = 2$. This requires having $c \geq \frac{a}{a-1} = 2$, which is automatically satisfied. For $n$ even, on the other hand, the binomial coefficients simplify to

$$\binom{n-1}{\lceil \frac{n-1}{2} \rceil} \geq a\binom{n-2}{\lceil \frac{n-2}{2} \rceil} \qquad \Longleftrightarrow \qquad 2\frac{n-1}{n} \geq a. \tag{60}$$

To satisfy this, we need to pick $a = 2\frac{n-1}{n}$, which requires $c \geq \frac{a}{a-1} = 2\frac{n-1}{n-2}$; however, this too is automatically satisfied by (36) provided $n \geq 4$. This completes the proof. $\qquad\square$

**Lemma C.7.** *Under the constraint (36), condition (30) holds.*

*Proof.* We remind that condition (30) is necessary for the correct "functioning" of the global shift layer, and it composes of two parts. The first part requires that $\tilde{l}_j < c^n \delta \ \forall j < n$. Thanks to Lemma C.5, it suffices to show that $\tilde{l}_{n-1} < c^n \delta$, but this is already granted by the upper bound in Lemma C.6. Analogously, for the second part, we need to show that $\tilde{l}_n > c^n \delta$: for this too we can use the lower bound in Lemma C.6. $\qquad\square$

We finally have all the ingredients to prove the main theorem of this section:

**Theorem C.8.** *The map in (34), given by*

$$\boldsymbol{X} \mapsto \boldsymbol{q}(\boldsymbol{X}) = \boldsymbol{u}^T \Psi(\boldsymbol{X})$$

*represents a contextual mapping.*

*Proof.* As defined in Def. C.2, a contextual mapping must satisfy two conditions. The first one is that

$$q_i(\boldsymbol{X}) \neq q_j(\boldsymbol{X}), \quad \forall i \neq j \qquad \text{and} \qquad \forall \boldsymbol{X} \in \mathbb{L}. \tag{61}$$

This is directly proven by considering Lemma C.5: since $\tilde{l}_j$ is a (strictly) increasing sequence, all its elements are already distinct. The action of the last global shift layer merely translates all these elements by a same quantity, but they remain distinct nonetheless.

The second condition for a contextual mapping is given by

$$q_i(\boldsymbol{X}) \neq q_j(\boldsymbol{X}'), \quad \forall i, j \quad \text{and} \quad \forall \boldsymbol{X}, \boldsymbol{X}' \in \mathbb{L}, \quad \text{with} \quad \boldsymbol{X} \neq \boldsymbol{X}'. \tag{62}$$

We prove that this holds for (34) by directly considering the difference between two components $i, j$ for different inputs:

$$q_i(\boldsymbol{X}) - q_j(\boldsymbol{X}') = \tilde{l}_i - \tilde{l}'_j + c^{n+1} \left( \tilde{l}_n - \tilde{l}'_n \right) = 0 \quad \Longleftrightarrow \quad \tilde{l}_i - \tilde{l}'_j = c^{n+1} \left( \tilde{l}'_n - \tilde{l}_n \right). \tag{63}$$

Notice that, due to Lemma C.4, we have $\tilde{l}_n - \tilde{l}'_n \neq 0$ and particularly, $|\tilde{l}_n - \tilde{l}'_n| \geq \delta$. On the other hand, in light of the bounds in Lemma C.6, we have that the left-hand side $|\tilde{l}_j - \tilde{l}_i| < c^n \delta$. Consequently, the two sides can never cancel each other out, and the proof is complete. $\square$

# D   LIPSCHITZNESS OF SIGMOID ATTENTION

In the following, we report the proof for the recovering the Lipschitzness constant associated with SigmoidAttn, as stated in Thm. 3.2.

Letting $A = W_q^T W_k$, and calling $\sigma_{ij} = \sigma(\langle W_q x_i, W_k x_j \rangle)$ and $\sigma'_{ij} = \sigma'(\langle W_q x_i, W_k x_j \rangle)$, we find that the Jacobian of $\phi$ in the direction $(\delta_1, \ldots, \delta_n)$ for the sample $x_i$ is given by:

$$\text{Jac}_i = \left( \sum_{j=1}^{n} \sigma'_{ij} x_j x_j^T A^T \right) \delta_i + \sum_{j=1}^{n} \left( \sigma'_{ij} x_j x_i^T A + \sigma_{ij} I_p \right) \delta_j, \tag{64}$$

We see that this Jacobian is the sum of two terms. To control its norm, we can control each norm individually.

The first term, $\left( \sum_{j=1}^{n} \sigma'_{ij} x_j x_j^T A^T \right) \delta_i$ is of the form $U_i \delta_i$ with $U_i$ a matrix. Its squared-norm is therefore:

$$\sum_{i=1}^{n} \|U_i \delta_i\|^2 \leq \max_i \|U_i\|_2^2 \|\delta\|_F. \tag{65}$$

Hence, its squared spectral norm is bounded by $\max_i \|U_i\|_2^2$.

We now let $\sigma'_\infty$ be a bound on $n \times |\sigma'|$; We have:

$$\|U_i\|_2 \leq \sum_{j=1}^{n} \|\sigma'_{ij} x_j x_j^\top A\|_2 \tag{66}$$

$$\leq \sigma'_\infty \|A\|_2 \frac{1}{n} \sum_{j=1}^{n} \|x_j\|^2 \tag{67}$$

$$\leq \sigma'_\infty \|A\|_2 \mathbb{E}[\|x_j\|^2]. \tag{68}$$

We see that if the points $x_i$ have norm $\leq R$, then the Jacobian grows at most like $R^2$, because it is "quadratic" in $x$. However, we see that the quadratic term is likely to be mitigated by the $\sigma'(a_{ij})$ term that goes to 0 if $a_{ij}$ is large.

The second term, $\sum_{j=1}^{n} \left( \sigma'_{ij} x_j x_i^T A + \sigma_{ij} I_p \right) \delta_j$, is the sum of two terms. Here, too, we use the triangular inequality to control their norm individually. We get:

$$\| \sum_{j=1}^{n} \sigma_{ij} \delta_j \|^2 = \| \delta^T \sigma_i \|^2 \tag{69}$$

$$\leq \| \delta \|_F^2 \| \sigma_i \|^2, \tag{70}$$

where $\sigma_i \in \mathbb{R}^p$ is the $i$-th column of $\sigma_{ij}$, and $\delta \in \mathbb{R}^{n \times p}$. and by summing, letting $\sigma_\infty$ an upper bound on $n \times |\sigma(x)|$:

$$\sum_{i=1}^{n} \| \sum_{j=1}^{n} \sigma_{ij} \delta_j \|^2 \leq \sigma_\infty^2 \| \delta \|_F^2. \tag{71}$$

So that $\sigma_\infty$ upper bounds the spectral norm of the last term.

For the final term, $\sum_{j=1}^{n} \sigma'_{ij} x_j x_i^T A \delta_j$, define $\hat{\delta} = \delta A^T$. We get:

$$\sum_{j=1}^{n} \sigma'_{ij} x_j x_i^T A \delta_j = \sum_{j=1}^{n} \sigma'_{ij} \langle x_i, \hat{\delta}_j \rangle x_j. \tag{72}$$

Hence, letting $M$ the matrix of entries $M_{ij} = \sigma'_{ij} \langle x_i, \hat{\delta}_j \rangle$, we see that the previous term is simply $x^T M_i^T$, so that we get the upper bound on the norm of the term:

$$\sum_{i=1}^{n} \| x^T M_i^T \|^2 \leq \| x \|_F^2 \| M \|_F^2 \tag{73}$$

and $\| M \|_F^2 = \sum_{ij} (\sigma'_{ij})^2 \langle x_i, \hat{\delta}_j \rangle^2 \leq \frac{1}{n^2} \sigma'_\infty \| x \|_F^2 \| A \|_2^2 \| \delta \|_F^2$, giving overall:

$$\sqrt{\sum_{i=1}^{n} \| x^T M_i^T \|^2} \leq \sigma'_\infty \| A \|_2 \mathbb{E}[\| x_j \|^2] \| \delta \|_F. \tag{74}$$

Notice how this quantity matches the one in (68).

Finally, summing all together gives:

$$\| \text{Jac} \|_2 \leq 2 \sigma'_\infty \| A \|_2 \mathbb{E}[\| x_j \|^2] + \sigma_\infty, \tag{75}$$

which completes the proof.

**Remark**: The previous upper bound might not be tight. Indeed, intuitively, if the $x_i$ are large, then the term $\sigma'_{ij}$ should be exponentially small (provided, of course, that $W_q x_i$ and $W_k x_j$ are not orthogonal), which would even remove the dependency on the variance in the sigmoid attention.

## E   THE BIAS TERM OF SIGMOID ATTENTION

One of the differences between SigmoidAttn and SoftmaxAttn is the normalization constant. In SigmoidAttn, one way to emulate the effect of a normalization constant (which links all the elements of the input together and defines a distribution over them), is to include a bias term in the definition as proposed in (3).

For an input vector $z \in \mathbb{R}^n$, the output of the sigmoid with bias $b$ is

$$\sigma^b(z)_i := \frac{\exp(z_i)}{\exp(z_i) + \exp(-b)}$$

Contrary to the softmax, this output cannot always sum to one because there is no normalization. We therefore seek a value for $b$ that *approximately* normalizes $\sigma^b(z)$, i.e., such that $\sum_{i=1}^{n} \sigma^b(z)_i \simeq 1$. We have

**Proposition E.1.** *Let* $z \in \mathbb{R}^n$, *and take* $m, M \in \mathbb{R}$ *such that for all* $i$, *it holds* $m \leq z_i \leq M$. *Then, the equation* $\sum_{i=1}^{n} \sigma^b(z)_i = 1$ *with variable* $b$ *has a single solution* $b^*$ *with*

$$-\log(n-1) - M \leq b^* \leq -\log(n-1) - m \ .$$

*Proof.* The function $\phi : b \to \sum_{i=1}^{n} \sigma^b(\boldsymbol{z})_i$ is smooth and monotonically increasing, and we have $\phi(-\log(n-1) - M) \leq 1$ and $\phi(-\log(n-1) - m) \geq 1$. This shows the existence of $b^*$ as well as the advertised bound on $b^*$. $\qquad\square$

This suggests using a $b$ of the order of $-\log(n)$; in practice we use $b = -\log(n)$.

We can also look for a bias term $b$, which helps to approximate the softmax function by the sigmoid function.

We assume that softmax provides us with the true distribution $p^\star$, where $p_i^\star = \frac{e^{z_i}}{e^{z_i} + \sum_{j \neq i} e^{z_j}}$. The goal is to find the bias term $b$ such that sigmoid function with weights over all elements denoted by $p$, where $p_i = \sigma^b(\boldsymbol{z})_i$, approximates $p^\star$. Note that, as mentioned before, $p$ is not necessarily a distribution, i.e. $\sum_{i=1}^{n} p_i$ is not always equal to one.

In technical terms, we aim to estimate the normalizing factor $\boldsymbol{Z} = \sum_{i=1}^{n} e^{z_i}$. The existing approaches for estimating $\boldsymbol{Z}$ is compute-expensive for high dimensions and requires resampling methods. Also, the optimal value of $b$ would depend on the exact values of $\boldsymbol{z}$, which is unknown beforehand. Therefore, we propose a more intuitive way to estimate the order of bias but possibly with larger disparity. To distribute the independent masses in SigmoidAttn, we assume that each element has uniform weight for the model apriori, which means that none of the elements of the input vector $\boldsymbol{z}$ has any known importance over the others. In the simplest case when softmax is a uniform distribution, we ideally want to have the same order of values for sigmoid as of softmax, which should be $\frac{1}{n}$. Therefore, we can write down the following:

$$\forall i \qquad p_i = \frac{1}{1 + e^{-(z_i + b)}} \simeq \frac{1}{n} = p_i^\star \tag{76}$$

Ideally, we would like to have $1 + e^{-(z_i + b)} \simeq n$. Requiring that $p = p^*$ in the case where all the $z_i$ are 0 gives $\exp(-b) = n - 1$, i.e. $b \simeq -\log(n)$ for large $n$. In the case that all the $z_i$ are bounded, $|z_i| \leq M < \infty$ for some constant $M$, then $b \simeq -(M + \log(n)) \approx -\max\{M, \log(n)\}$. However, in most cases we do not know $M$. When the sequence length $n$ is large enough, the constant $M$ loses its importance while in short sequence length, it impacts distributing the weights over elements more. To resolve this issue, we assume that $z_i$ are sampled from a standard Gaussian distribution, i.e. $z_i \sim \mathcal{N}(0, \sigma^2)$ where $\sigma = 1$. Note that this assumption comes from the fact that $z_i$ in our problem is one of the elements of $\boldsymbol{Q}\boldsymbol{K}^T / \sqrt{d_{qk}}$, which is the sum of $d_{qk}$ random variables. Using Central Limit Theorem, we can assume that $z_i$ is sampled from a Gaussian distribution. The idea is to estimate $M$, such that with high probability, $|z_i| \leq M$, i.e. $\mathbb{P}(|z_i| > M) \leq \epsilon$ for a desired $\epsilon$. Therefore, we have

$$\mathbb{P}(|z_i| > M) = \mathbb{P}\left(|z_i| > \frac{M}{\sigma}\sigma\right) \leq \frac{1}{(\frac{M}{\sigma})^2} = \frac{\sigma^2}{M^2} \leq \epsilon, \tag{77}$$

where the inequality is resulted from Chebychev's inequality. Setting $\sigma = 1$, we have $M \simeq \sqrt{1/\epsilon}$. Therefore, the order-optimal value would be $b \simeq -\max\{\sqrt{1/\epsilon}, \log(n)\}$, and for long sequence length, $b \simeq -\log(n)$. For example, if we want 90% accuracy in our estimation, $M \approx 3\sigma = 3$, which means $b \simeq -\max\{3, \log(n)\}$. Note that this approximation also follows the intuition that as $n$ grows, we expect the SigmoidAttn without bias term overestimate the mass on each point, so we need to normalize the mass according to $n$ at each point as well.

On another side, one may be more interested in the gradients of $p^\star$ and $p$ with respect to $z_i$ to behave similarly. We show that $b \simeq -\log(n)$ is still a good choice in this scenario. Let us derive the derivative of SigmoidAttn and SoftmaxAttn with respect to the input. We note that for any $i$, both functions can be written as $\frac{e^{z_i}}{e^{z_i} + \boldsymbol{Z}_{-i}}$ where $\boldsymbol{Z}_{-i}$ is the share of normalization factor except element $i$ of $\boldsymbol{z}$. For SoftmaxAttn, $\boldsymbol{Z}_{-i} = \sum_{j \neq i} e^{z_j}$ and for SigmoidAttn, $\boldsymbol{Z}_{-i} = e^{-b}$. Now, we have

$$\frac{\partial}{\partial z_i} \frac{e^{z_i}}{e^{z_i} + \boldsymbol{Z}_{-i}} = \frac{e^{z_i}\boldsymbol{Z}_{-i}}{(e^{z_i} + \boldsymbol{Z}_{-i})^2}. \tag{78}$$

Therefore, we have the following

$$\frac{\partial p_i^\star}{\partial z_i} = p_i^\star(1 - p_i^\star) \tag{79}$$

$$\frac{\partial p_i}{\partial z_i} = p_i(1 - p_i). \tag{80}$$

We can see that if $p_i \simeq p_i^\star$, then $\frac{\partial p_i}{\partial z_i} \simeq \frac{\partial p_i^\star}{\partial z_i}$. So, the previous choice of bias term $b \simeq -\log(n)$ approximates the order of gradients as well. In fact, this is the only valid choice even though we have a quadratic term.

$$\frac{\partial p_i}{\partial z_i} \simeq \frac{\partial p_i^\star}{\partial z_i} \quad \Longleftrightarrow \quad p_i^\star(1 - p_i^\star) = p_i(1 - p_i) \tag{81}$$

$$\Longleftrightarrow \quad (p_i - p_i^\star)(p_i - (1 - p_i^\star)) = 0. \tag{82}$$

Which means either $p_i \simeq p_i^\star$ or $p_i \simeq 1 - p_i^\star$. The first one provides us with $b \simeq -\log(n)$ while the second one cannot happen since the nominator of $p_i$ is dependent on $z_i$ while the nominator of $1 - p_i^\star$ is independent of $z_i$.

## F   DETAILS OF FLASHSIGMOID

This appendix provides details of the FLASHSIGMOID algorithm. We begin by discussing the implementation details of FLASHSIGMOID, which we build as an extension of FLASHATTENTION2, followed by a benchmark of the performance of the involved kernels. We show that the kernels of FLASHSIGMOID provide a considerable performance boost in model inference over those of FLASHATTENTION2 and a modest performance boost for model training. Further, we demonstrate that the kernel speed boosts also reflect in a considerable performance gain in realistic end-to-end experiments, with an example of training vision transformers (Dosovitskiy et al., 2021) on the ImageNet dataset (Deng et al., 2009). Finally, we also provide kernel benchmarking details of FLASHSIGMOID implementation by taking into account ALiBi slopes (Press et al., 2022), which is one of the important components of SigmoidAttn as seen in the main text of the paper.

### F.1   DETAILS OF FLASHSIGMOID ALGORITHM

**Softmax vs. Sigmoid Attention:**   In this subsection, we discuss the implementation details of FLASHSIGMOID algorithm, which is a hardware-aware implementation of SigmoidAttn approach. We begin with the expressions of the forward and backward passes of softmax and sigmoid attention mechanisms. Let $Q$, $K$, and $V$ represent the query, key, and value tensors. Then, the desired forward and backward pass expressions are reported in Tab. 3.   The application of sigmoid and

| SOFTMAX | | SIGMOID | |
|---|---|---|---|
| FORWARD | BACKWARD | FORWARD | BACKWARD |
| $S = \dfrac{Q \cdot K^\top}{\sqrt{d}}$ | $dV = P^\top \cdot dO$ | $S = \dfrac{Q \cdot K^\top}{\sqrt{d}}$ | $dV = P^\top \cdot dO$ |
| $P = \text{SOFTMAX}(S)$ | $dP = dO \cdot V^\top$ | $P = \sigma(S)$ | $dP = dO \cdot V^\top$ |
| $O = P \cdot V$ | $dS = P \odot (dP - \text{ROWSUM}(dO \odot O))$ | $O = P \cdot V$ | $dS = P \odot (1 - P) \odot dP$ |
| | $dQ = \sqrt{d} \cdot dS \cdot K$ | | $dQ = \sqrt{d} \cdot dS \cdot K$ |
| | $dK = \sqrt{d} \cdot dS^\top \cdot Q$ | | $dK = \sqrt{d} \cdot dS^\top \cdot Q$ |

Table 3: Description of the forward and backward passes of softmax and sigmoid attention. With $\odot$, we denote Hadamard (element-wise) multiplication.

softmax activation functions, as highlighted in orange color in Tab. 3, is the only implementation difference in the forward passes. Similarly, the expressions for the gradients of the preactivation ($dS$), as highlighted in purple color in the table above, is the only implementation difference in the backward passes. In light of this, we implement the FLASHSIGMOID algorithm as an extension of the FLASHATTENTION2 (Dao, 2023) algorithm, which is a highly optimized hardware-aware implementation of SoftmaxAttn.

**Flash Attention in Brief:**   As pointed at in the main text, the FLASHATTENTION (Dao et al., 2022) and FLASHATTENTION2 (Dao, 2023) algorithms provide hardware-aware implementations of exact attention mechanism by optimizing for bottlenecks of modern accelerators (Choquette et al., 2021; Choquette, 2023). These GPUs possess massive amounts (e.g., $\sim$ 80GB) of High-Bandwidth Memory (HBM), which stores large tensors but is slow in moving the data to the accelerators. On the other hand, they have smaller amounts (e.g., $\sim$ 20 MB) of SRAM, which is often more than an

---

**Algorithm 1** FLASHSIGMOID Forward Pass

---

1: **procedure** FORWARD( $Q, K, V, B_r, B_c$ ):
2:         """
3:         **inputs:** Matrices $Q, K, V \in \mathbb{R}^{n \times d}$ are on HBM of the GPU.
4:         **inputs:** Integers $B_r$ and $B_c$ are the block size for queries and key-values respectively.
5:
6:         **outputs:** Matrix $O \in \mathbb{R}^{n \times d}$ on HBM of the GPU.
7:             # No need to output logsumexp vector $L \in \mathbb{R}^n$ on HBM.
8:         """
9:         Divide $Q$ into $T_r := \lceil \frac{n}{B_r} \rceil$ blocks: $Q_1, \cdots, Q_{T_r}$ with $Q_i \in \mathbb{R}^{B_r \times d}$.
10:         Divide $K$ into $T_c := \lceil \frac{n}{B_c} \rceil$ blocks: $K_1, \cdots, K_{T_c}$ with $K_i \in \mathbb{R}^{B_c \times d}$.
11:         Divide $V$ into $T_c$ blocks: $V_1, \cdots, V_{T_c}$ with $V_i \in \mathbb{R}^{B_c \times d}$.
12:         Divide $O$ into $T_r$ blocks: $O_1, \cdots, O_{T_r}$ with $O_i \in \mathbb{R}^{B_r \times d}$.
13:         **for** $i = 1, \cdots, T_r$ **do**
14:             Load block $Q_i$ from HBM to SRAM of the GPU.
15:             On chip, initialize $O_i$ with zeros: $O_i \leftarrow \mathbf{0}^{B_r \times d}$.
16:                 # No allocation of either row-sum $\ell_i \in \mathbb{R}^{B_r}$ or row-max $m_i \in \mathbb{R}^{B_r}$ on chip.
17:             **for** $j = 1 \cdots T_c$ **do**
18:                 Load blocks $K_j, V_j$ from HBM to SRAM of the GPU.
19:                 On chip, evaluate pre-activations: $S_{ij} \leftarrow Q_i \cdot K_j^\top / \sqrt{d} \in \mathbb{R}^{B_r \times B_c}$.
20:                 On chip, evaluate sigmoid attention: $P_{ij} \leftarrow \sigma(S_{ij})$.
21:                 On chip, update output block: $O_i \leftarrow O_i + P_{ij} \cdot V_j$.
22:                     # No need to update and track $\ell_i$ and $m_i$ vectors.
23:             **end for**
24:             Store $O_i$ from chip to HBM as the $i$−th block of $O$ matrix.
25:                 # No post-processing of $O_i$ or $L_i$ blocks on chip.
26:                 # No movement of $L_i$ block from chip to HBM.
27:         **end for**
28:         **return** matrix $O$.
29: **end procedure**

---

order magnitude faster for carrying out actual computations using the registers/tensor cores of the GPU. This trade-off between memory size and computation speed across hierarchies results in the attention mechanism computation being bottlenecked by memory accesses between the HBM and the SRAM (Ivanov et al., 2021). Consequently, flash algorithms optimize for memory accesses across the hierarchy of GPU memory types in order to accelerate computation of attention mechanism and its gradients. FLASHSIGMOID is no exception to this approach.

Algorithm 1 describes the forward pass and Alg. 2 describes the backward pass of the FLASHSIGMOID algorithm. We highlight in orange color the steps in the forward pass of FLASHSIGMOID that differ from those in FLASHATTENTION2 by virtue of sigmoid activation. Similarly, we highlight in purple color the differences in the backward pass. Finally, we highlight in blue color the salient points of FLASHSIGMOID that further help minimize bottlenecking factors on modern accelerators.

**Fewer Tensor Allocations, Fewer Memory Accesses, Fast-Tanh:** In FLASHATTENTION and FLASHATTENTION2, the attention mechanism is computed by splitting the attention matrix into blocks. Since softmax activation requires a row-wise reduction to compute its normalization factor (i.e., the denominator), one needs to properly compute and track such factor across blocks. Moreover, in FLASHATTENTION this normalization factor is stored after being computed in the forward pass, to have it easily accessible to further speed-up the backward pass. By contrast, substituting sigmoid to softmax eliminates the need to allocate and move across the GPU memory hierarchy the tensors related to the normalization factor (i.e., moving the logsumexp tensor $L \in \mathbb{R}^n$ on HBM in the forward and backward passes). In addition, applying softmax in a stable manner requires tracking the row-max variable $m_i$ on chip, which instead is not needed for sigmoid activation. This further helps reducing some on-chip operations and lowering register pressure in FLASHSIGMOID.

Moving on to the backward pass (described in Alg. 2), FLASHATTENTION2 requires computing rowsum $(dO \odot O)$, which is needed to backpropagate the gradients of softmax attention outputs

---

**Algorithm 2** FLASHSIGMOID Backward Pass

---

1: **procedure** BACKWARD( $Q, K, V, \mathrm{d}O, B_r, B_c$ ):
2:       """
3:     **inputs:** Matrices $Q, K, V, \mathrm{d}O \in \mathbb{R}^{n \times d}$ are on HBM of the GPU.
4:     **inputs:** Integers $B_r$ and $B_c$ are the block size for queries and key-values respectively.
5:       # No need of logsumexp vector $L \in \mathbb{R}^n$ to be saved for the backward pass.
6:
7:     **outputs:** Matrices $\mathrm{d}Q, \mathrm{d}K, \mathrm{d}V \in \mathbb{R}^{n \times d}$ on HBM of the GPU.
8:       """
9:     Divide $Q$ into $T_r := \lceil \frac{n}{B_r} \rceil$ blocks: $Q_1, \cdots, Q_{T_r}$ with $Q_i \in \mathbb{R}^{B_r \times d}$.
10:     Divide $K$ into $T_c := \lceil \frac{n}{B_c} \rceil$ blocks: $K_1, \cdots, K_{T_c}$ with $K_i \in \mathbb{R}^{B_c \times d}$.
11:     Divide $V$ into $T_c$ blocks: $V_1, \cdots, V_{T_c}$ with $V_i \in \mathbb{R}^{B_c \times d}$.
12:     Divide $O$ into $T_r$ blocks: $O_1, \cdots, O_{T_r}$ with $V_i \in \mathbb{R}^{B_r \times d}$.
13:     Divide $\mathrm{d}O$ into $T_r := \lceil \frac{n}{B_r} \rceil$ blocks: $\mathrm{d}O_1, \cdots, \mathrm{d}O_{T_r}$ with $\mathrm{d}O_i \in \mathbb{R}^{B_r \times d}$.
14:     Allocate $\mathrm{d}Q$ on HBM and divide into $T_r$ blocks: $\mathrm{d}Q_1, \cdots, \mathrm{d}Q_{T_r}$ with $\mathrm{d}Q_i \in \mathbb{R}^{B_r \times d}$.
15:     Allocate $\mathrm{d}K$ on HBM and divide into $T_c$ blocks: $\mathrm{d}K_1, \cdots, \mathrm{d}K_{T_c}$ with $\mathrm{d}K_i \in \mathbb{R}^{B_c \times d}$.
16:     Allocate $\mathrm{d}V$ on HBM and divide into $T_c$ blocks: $\mathrm{d}V_1, \cdots, \mathrm{d}V_{T_c}$ with $\mathrm{d}V_i \in \mathbb{R}^{B_c \times d}$.
17:       # No need to compute rowsum $(\mathrm{d}O \odot O)$ as sigmoid and its gradients are pointwise.
18:     **for** $j = 1, \cdots, T_c$ **do**
19:       Load blocks $K_j, V_j$ from HBM to SRAM of the GPU.
20:       On chip, initialize $\mathrm{d}K_j, \mathrm{d}V_j$ with zeros: $\mathrm{d}K_j \leftarrow \mathbf{0}^{B_c \times d}; \mathrm{d}V_j \leftarrow \mathbf{0}^{B_c \times d}$.
21:       **for** $i = 1 \cdots T_r$ **do**
22:         Load blocks $Q_i, \mathrm{d}O_i, \mathrm{d}Q_i$ from HBM to SRAM of the GPU.
23:         # No need of movement of blocks rowsum $(\mathrm{d}O \odot O)_i$ and logsumexp $L_i$.
24:         On chip, evaluate pre-activations: $S_{ij} \leftarrow Q_i \cdot K_j^\top / \sqrt{d} \in \mathbb{R}^{B_r \times B_c}$.
25:         On chip, evaluate sigmoid attention: $P_{ij} \leftarrow \sigma(S_{ij})$.
26:         On chip, update gradient of values: $\mathrm{d}V_i \leftarrow \mathrm{d}V_i + P_{ij}^\top \cdot \mathrm{d}O_j$.
27:         On chip, compute gradients of attention matrix: $\mathrm{d}P_{ij} \leftarrow \mathrm{d}O_i \cdot V_i^\top \in \mathbb{R}^{B_r \times B_c}$.
28:         On chip, compute gradients of pre-activations: $\mathrm{d}S_{ij} \leftarrow P_{ij} \odot (1 - P_{ij}) \odot \mathrm{d}P_{ij}$.
29:         Load query gradient block $\mathrm{d}Q_i$ from HBM to SRAM, and then on to chip.
30:         Update query gradient block on chip: $\mathrm{d}Q_i \leftarrow \mathrm{d}Q_i + \sqrt{d} \cdot \mathrm{d}S_{ij} \cdot K_j$.
31:         Store query gradient block $\mathrm{d}Q_i$ from chip back to HBM.
32:         On chip, update key gradient block: $\mathrm{d}K_j \leftarrow \mathrm{d}K_j + \sqrt{d} \cdot \mathrm{d}S_{ij}^\top \cdot Q_i$.
33:       **end for**
34:       Store $\mathrm{d}K_j, \mathrm{d}V_j$ from chip to HBM as the $j-$th blocks of $\mathrm{d}K, \mathrm{d}V$ matrices respectively.
35:     **end for**
36:     **return** matrices $\mathrm{d}Q, \mathrm{d}K, \mathrm{d}V$.
37: **end procedure**

---

to the preactivations. However, since sigmoid activation is applied element-wise, its gradients also backpropagate across sigmoid element-wise, eliminating the need of the row-sum variable and the movement of its blocks across the memory hierarchy. Another optimization of FLASHATTENTION and FLASHATTENTION2 consists of partially re-computing the forward pass of attention mechanism in the backward pass to avoid bottlenecks and speed-up the implementation. To keep the backward pass implementation fast, they require the logsumexp variable to be available and transferred between HBM and SRAM in the backward pass. FLASHSIGMOID, being an element-wise activation, eliminates the need of this variable from the backward pass, and consequently, from the entire algorithm. Finally, a major component in our implementation is the usage of GPU-based implementation of the tanh activation. Sigmoid activation is related to Tanh activation via the following relation: $\sigma(x) = 0.5 \cdot (1 + \tanh(0.5 \cdot x))$. We utilize the fast GPU-implementation of Tanh activation, which trades off some precision for better speed, in order to compute sigmoid activation in both the forward and the backward pass. This provides a considerable speed-boost in both the forward and backward passes of FLASHSIGMOID, while maintaining parity in performance with a naïve implementation of sigmoid attention. Based on these points of modification, we extend FLASHATTENTION2 to obtain FLASHSIGMOID, a hardware-aware implementation of SigmoidAttn.

### F.2 BENCHMARKING OF FLASHSIGMOID KERNELS

**Benchmarking Setup:** Having seen the details of the FLASHSIGMOID algorithm, we next consider the benchmarking of its kernels. For this, we create a small model in PyTorch (Paszke et al., 2019) that inputs query, key, and value tensors (all of shape [batch, tokens, heads, features]) and passes these through a number of attention layers. Mimicking the design of vision transformers (ViTB-16/224) (Dosovitskiy et al., 2021), we set the number of heads and per-head features as 12 and 64, respectively. We set a batch size of 32, and consider a 10-layer architecture. Then, for the number of tokens sampled from a wide range of [64, 78k], we compute the forward and backward passes of this model. For these computations, we measure the kernel GPU time using PyTorch's profiler. We carry out our experiments on both H100 (Choquette, 2023) and A100 (Choquette et al., 2021) GPUs.

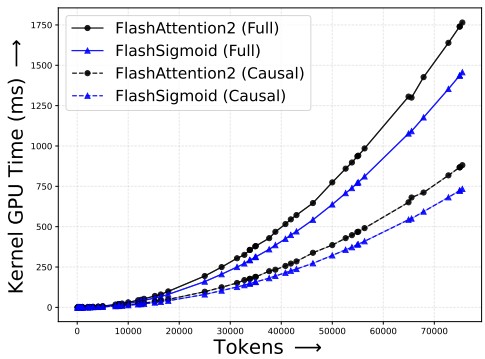

(a) Inference mode kernels on H100.

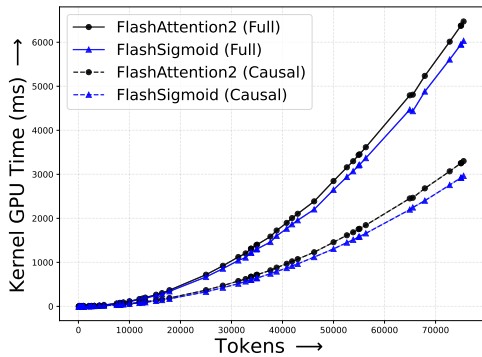

(b) Training mode kernels on H100.

Figure 11: On average, for sequence lengths between [64, 78k], the inference mode kernel of FLASHSIGMOID is 17.39% faster than FLASHATTENTION2 for self-attention and 18.76% for causal attention. The training mode kernels of FLASHSIGMOID are 6.53% faster than FLASHATTENTION2 for self-attention and 9.46% for causal attention. Note that inference involves only the forward pass of the model and training involves both the forward and the backward pass of the model.

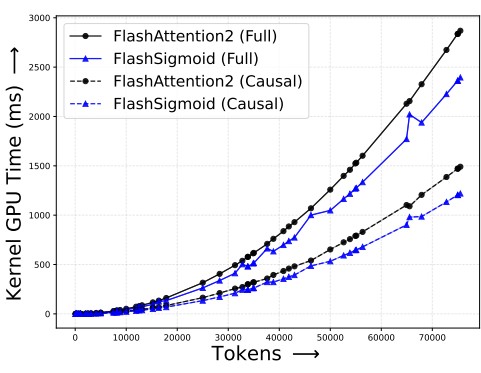

(a) Inference mode kernels on A100.

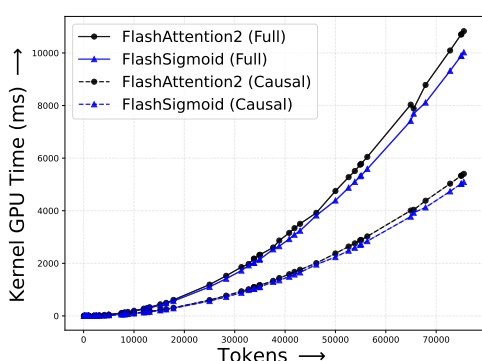

(b) Training mode kernels on A100.

Figure 12: On average, for sequence lengths between [64, 78k], the inference mode kernel of FLASHSIGMOID is 14.33% faster than FLASHATTENTION2 for self-attention and 16.92% for causal attention. The training mode kernels of FLASHSIGMOID are 6.02% faster than FLASHATTENTION2 for self-attention and 5.27% for causal attention. Note that inference involves only the forward pass of the model and training involves both the forward and the backward pass of the model.

**Results:** Figures 11 and 12 show the GPU time comparisons of kernels in inference mode and training mode of FLASHSIGMOID and FLASHATTENTION2 respectively. We observe that we obtain a large average speed-boost for inference and a modest average speed-boost for training. Note that

the speed-ups in all the subsequent figures are obtained by averaging the performances for tokens sampled in the range of $[64, 78\text{k}]$.

**Details of Individual Kernels:** Next, we also show the performance of individual flash kernels of FLASHSIGMOID and FLASHATTENTION2. Note that inference mode involves only the forward pas of the model, while training mode involves both the forward and the backward pass of the model. The forward pass of both these approaches involves one kernel, which we term flash_fwd_kernel, and the backward pass of both these approaches is made up of three kernels, which we term bwd_dq_dk_dv, bwd_dot_do_o, and bwd_convert_dq. In code, the real names of these kernels are as follows.

$$
\begin{aligned}
\text{fwd} &:= \text{flash\_fwd\_kernel} \\
\text{bwd\_dq\_dk\_dv} &:= \text{flash\_bwd\_dq\_dk\_dv\_loop\_seqk\_parallel\_kernel} \\
\text{bwd\_dot\_do\_o} &:= \text{flash\_bwd\_dot\_do\_o\_kernel} \\
\text{bwd\_convert\_dq} &:= \text{flash\_bwd\_convert\_dq\_kernel}
\end{aligned}
\tag{83}
$$

Here, we first provide a brief description of the tasks performed by each of these kernels; for a detailed explanation, we refer the reader to FLASHATTENTION2 (Dao, 2023) paper and code. The fwd kernel computes the full forward pass of the model as shown in Tab. 3. The bulk of computations of the backward pass happen in the bwd_dq_dk_dv kernel, which performs re-computation of attention matrix and reduction of key and value gradient tensors ($\mathbf{d}\boldsymbol{K}$, $\mathbf{d}\boldsymbol{V}$). Again, the exact steps carried out in the backward pass can be checked from Tab. 3. The bwd_convert_dq kernel performs the reduction of query gradient tensor ($\mathbf{d}\boldsymbol{Q}$). Finally, note that the bwd_dot_do_o kernel in FLASHATTENTION2 performs the task of computing the rowsum($\mathbf{d}\boldsymbol{O} \odot \boldsymbol{O}$) tensor along with clearing of the accumulators of query gradients ($\mathbf{d}\boldsymbol{Q}$). Although FLASHSIGMOID does not require this row-sum tensor, the clearing of accumulators of query gradients is still needed. For this reason, bwd_dot_do_o kernel also appears in the profiling of FLASHSIGMOID.

**Performance of Individual Kernels:** Figures 13 and 14 show the performance comparison of each flash kernel in FLASHSIGMOID with the corresponding kernel in FLASHATTENTION2 when tested on an H100 GPU and an A100 GPU respectively. We observe that on both the H100 and A100 GPU architectures, the fwd kernel of FLASHSIGMOID is significantly faster than that of FLASHATTENTION2 and the bwd_dq_dk_dv kernel of FLASHSIGMOID has a modest average speed boost over FLASHATTENTION2. The bwd_dot_do_o kernel in FLASHSIGMOID is significantly faster on A100 GPUs. Note that even though the bwd_dot_do_o kernel of FLASHSIGMOID appears to be slower on average on H100 GPUs, the kernel time of bwd_dot_do_o ($\sim 5\text{ms}$) is negligible compared to that of the main bwd_dq_dk_dv kernel ($\sim 5000\text{ms}$). Thus, the combined backward pass kernel in FLASHSIGMOID time does not suffer from this slowdown. Finally, note that for bwd_convert_dq, FLASHSIGMOID and FLASHATTENTION2 have identical performance. This is expected, since the task of this kernel is to reduce the gradient of the queries $\mathbf{d}\boldsymbol{Q}$, which is a common step in both the approaches and is not modified in FLASHSIGMOID.

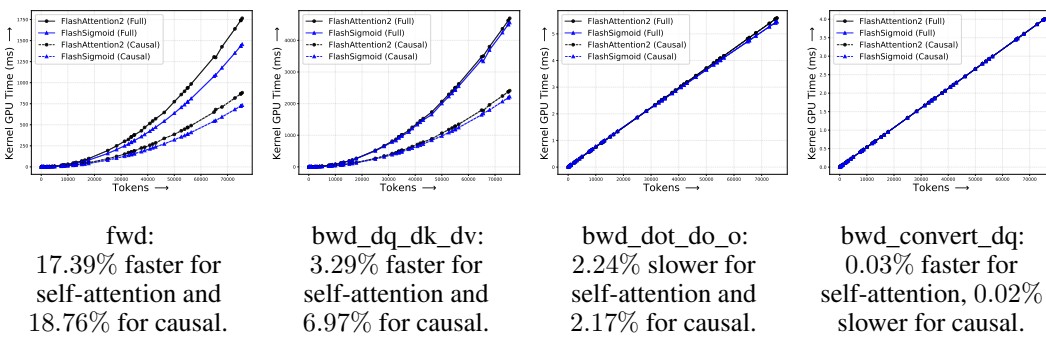

| fwd: 17.39% faster for self-attention and 18.76% for causal. | bwd_dq_dk_dv: 3.29% faster for self-attention and 6.97% for causal. | bwd_dot_do_o: 2.24% slower for self-attention and 2.17% for causal. | bwd_convert_dq: 0.03% faster for self-attention, 0.02% slower for causal. |

Figure 13: FLASHSIGMOID and FLASHATTENTION2 kernel comparison on H100 GPUs.

### F.3 SPEED BOOSTS OF FLASHSIGMOID IN REALISTIC SETTINGS

In this section, we demonstrate how the performance boosts measured in App. F.2 for the individual kernels of FLASHSIGMOID contributes to speeding-up realistic runs with end-to-end training.

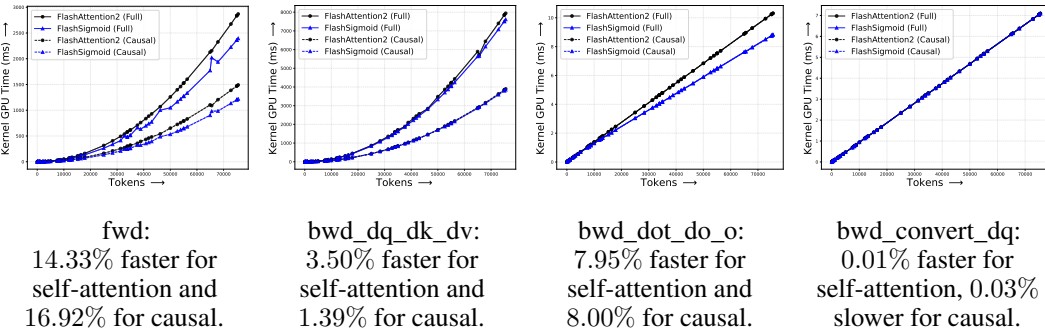

| fwd: | bwd_dq_dk_dv: | bwd_dot_do_o: | bwd_convert_dq: |
|---|---|---|---|
| 14.33% faster for self-attention and 16.92% for causal. | 3.50% faster for self-attention and 1.39% for causal. | 7.95% faster for self-attention and 8.00% for causal. | 0.01% faster for self-attention, 0.03% slower for causal. |

Figure 14: FLASHSIGMOID and FLASHATTENTION2 kernel comparison on A100 GPUs.

**Setup:** As a target experiment, we consider training a vision transformer (Dosovitskiy et al., 2021) on the ImageNet dataset (Deng et al., 2009). We create two vision transformer model variants– one with FLASHATTENTION2 attention and the other with FLASHSIGMOID attention. We carry out the training of these models with a distributed data-parallel (DDP) setup using PyTorch (Paszke et al., 2019). We perform two sets of experiments– *i.* the first performs DDP training on four nodes of H100 GPUs with eight GPUs per node and EFA/RDMA interconnect for the nodes, and *ii.* the second performs DDP training on four nodes of A100 GPUs with eight GPUs per node. In each set of experiments, we use three different image sizes ($64 \times 64$, $90 \times 90$, and $100 \times 100$), along with patch size of 1 to result in different number of tokens for the underlying attention mechanism in the vision transformer model ($64 \times 64 = 4096$, $90 \times 90 = 8100$, and $100 \times 100 = 10000$ tokens). For each of these configurations, we select batch sizes so that the GPU memory utilization would be greater than 80%. These considerations are in order to minimize, if not eliminate, other confounders that can unfairly affect estimation speed-ups in realistic runs. For instance, a low GPU utilization would lead to a larger number of updates, which in turn would incur unnecessary delays, variations, and slow-downs due to across-nodes communications.

**Results:** The results of the runs on H100 nodes and A100 nodes are shown in Tab. 4 and 5 respectively. There, we show how the kernel GPU times for forward and backward passes vary according to the number of tokens considered, and include the wall-clock time of the end-to-end runs as explained above. We observe that the kernel speed-up reflects significantly in the speed-up of inference of the models (during testing) and modestly in the training of the models. We observe $\sim 8\%$ speed-up in wall-clock time of inference and $\sim 4\%$ speed-up in wall-clock time of training.

| TOKENS | KERNEL GPU TIME COMPARISON | | | FULL RUN WALL-CLOCK TIME COMPARISON | | |
|---|---|---|---|---|---|---|
| | KERNELS | FLASHATTENTION2 (MS) | FLASHSIGMOID (MS) | MODE | FLASHATTENTION2 (S) | FLASHSIGMOID (S) |
| 4096 | FWD | 4.98±0.01 | 4.17±0.01 (−**16.31**%) | INFERENCE | 11.17±0.18 | 10.68±0.18 (−**4.42**%) |
| | FWD + BWD | 19.58±0.06 | 18.12±0.04 (−**7.45**%) | TRAINING | 1563.39±1.30 | 1521.68±2.27 (−**2.67**%) |
| 8100 | FWD | 20.46±0.05 | 16.73±0.05 (−**18.22**%) | INFERENCE | 28.21±0.18 | 25.93±0.17 (−**8.06**%) |
| | FWD + BWD | 77.63±0.13 | 72.70±0.12 (−**6.35**%) | TRAINING | 4282.75±2.14 | 4129.25±4.14 (−**3.58**%) |
| 10000 | FWD | 31.17±0.07 | 25.49±0.05 (−**18.20**%) | INFERENCE | 38.71±0.19 | 35.37±0.17 (−**8.62**%) |
| | FWD + BWD | 117.53±0.13 | 109.87±0.12 (−**6.52**%) | TRAINING | 5990.72±2.21 | 5751.43±5.77 (−**3.99**%) |

Table 4: FLASHSIGMOID vs. FLASHATTENTION2 on H100 nodes. The kernel GPU time for both the approaches is reported in milliseconds and wall-clock times is reported in seconds per epoch.

**Connection of Wall-Clock Time Speed-Up and Kernel Speed-Up:** From Tab. 4 and 5, it is clear that the speed-up in kernels is larger than that in the wall-clock times of the full runs. In fact, the speed-up in kernels is the upper bound for the speed-up that we would see in wall-clock times. To see why, let us denote by $\tau_{\text{sm}}$ and $\tau_\sigma$ the total kernel GPU time for softmax attention and sigmoid attention respectively. Then, the kernel speed-up is given by $s_{\text{kernel}} := 1 - \frac{\tau_\sigma}{\tau_{\text{sm}}}$. However, in a full run, the total wall clock time also incorporates the time required to load data, time taken by other layers of the underlying models, time required to communicate gradients and other data across GPUs and across nodes, and so on. For our corresponding sigmoid and softmax runs, these extra factors are designed to add, upon expectation, in the same extra time $\tau$. Thus, the wall-clock time speed-up of a full run

| TOKENS | | KERNEL GPU TIME COMPARISON | | FULL RUN WALL-CLOCK TIME COMPARISON | | |
|---|---|---|---|---|---|---|
| | KERNELS | FLASHATTENTION2 (MS) | FLASHSIGMOID (MS) | MODE | FLASHATTENTION2 (S) | FLASHSIGMOID (S) |
| 4096 | FWD | 8.32±0.02 | 7.84±0.03 (−**5.79**%) | INFERENCE | 19.05±0.22 | 18.74±0.19 (−**1.65**%) |
| | FWD + BWD | 31.81±0.08 | 31.11±0.08 (−**2.19**%) | TRAINING | 2795.03±2.35 | 2769.44±5.10 (−**0.92**%) |
| 8100 | FWD | 33.65±0.09 | 27.92±0.07 (−**17.04**%) | INFERENCE | 47.35±0.20 | 44.05±0.17 (−**6.96**%) |
| | FWD + BWD | 128.18±0.13 | 119.04±0.12 (−**7.13**%) | TRAINING | 7519.64±4.21 | 7254.84±12.64 (−**3.52**%) |
| 10000 | FWD | 51.17±0.07 | 42.49±0.06 (−**16.96**%) | INFERENCE | 64.61±0.32 | 59.55±0.18 (−**7.82**%) |
| | FWD + BWD | 194.54±0.14 | 180.59±0.15 (−**7.17**%) | TRAINING | 10455.64±8.85 | 10052.04±18.87 (−**3.86**%) |

Table 5: FLASHSIGMOID vs. FLASHATTENTION2 on A100 nodes. The kernel GPU time for both the approaches is reported in milliseconds and wall-clock times is reported in seconds per epoch.

with end-to-end training is $s_{\text{wall-clock}} := 1 - \frac{\tau_\sigma + \tau}{\tau_{\text{sm}} + \tau}$. Since we have faster sigmoid kernels, we have $\tau_\sigma < \tau_{\text{sm}}$, which in turn shows that $s_{\text{wall-clock}} = 1 - \frac{\tau_\sigma + \tau}{\tau_{\text{sm}} + \tau} < 1 - \frac{\tau_\sigma}{\tau_{\text{sm}}} = s_{\text{kernel}}$. This explains the speed boost trends in kernel time versus full run wall-clock time for each setting in Tab. 4 and 5. However, in particular, if a model performs attention mechanism over large number of tokens, the attention mechanism, and hence the corresponding kernel time, starts to dominate the other computations in the network. In that case, we see that the wall-clock time speed-boost is closer to the kernel speed-boost. Mathematically, if $\tau_\sigma, \tau_{\text{sm}} \gg \tau$, we have: $\tau_\sigma + \tau \approx \tau_\sigma$, $\tau_{\text{sm}} + \tau \approx \tau_{\text{sm}}$. Thus, $s_{\text{kernel}} \approx s_{\text{wall-clock}}$, thereby making $s_{\text{wall-clock}}/s_{\text{kernel}} \to 1$.

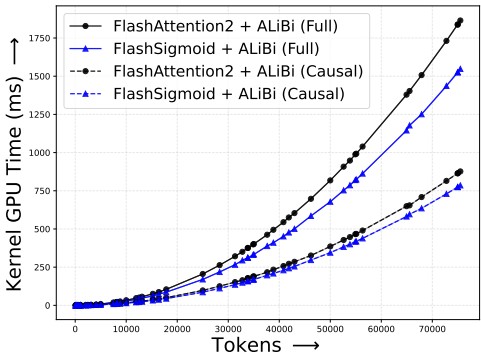

(a) Inference mode kernels on H100.

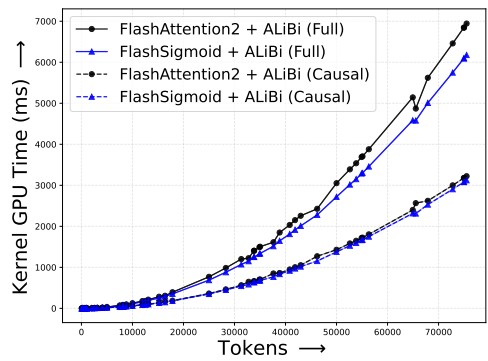

(b) Training mode kernels on H100.

Figure 15: On average, for sequence lengths between $[64, 78\text{k}]$, the inference mode kernel of FLASHSIGMOID is $17.04\%$ faster than FLASHATTENTION2 for self-attention and $10.87\%$ for causal attention. The training mode kernels of FLASHSIGMOID are $8.91\%$ faster than FLASHATTENTION2 for self-attention and $4.72\%$ for causal attention. Note that inference involves only the forward pass of the model and training involves both the forward and the backward pass of the model.

**Significance of Wall-Clock Speed-Up of Inference:** Although FLASHSIGMOID provides only modest gains during training, the speed-up in inference is significant ($> 15\%$ for underlying kernels and $5 - 10\%$ during inference of full runs). We posit that this speed-up in inference is extremely critical as well. Contemporary large-scale models, once trained, spend a huge portion of the rest their lifetime in inference mode (OpenAI, 2023). Thus, significant performance boosts in inference mode have immense potential for saving resources in deployment of large models for inference.

### F.4 FLASHSIGMOID WITH ALIBI

It is evident from the main text of the paper that improved positional embeddings, like ALiBi (Press et al., 2022), can be crucial for certain tasks and data modalities. Thus, we also provide a FLASH-SIGMOID implementation that incorporates ALiBi. We compare the FLASHSIGMOID with ALiBi implementation with the FLASHATTENTION2 with ALiBi implementation (Dao, 2023). Figures 15 and 16 show the kernel GPU time for the forward and backward pass kernels of FLASHSIGMOID with ALiBi implementation versus FLASHATTENTION2 with ALiBi implementation. Again, we observe that FLASHSIGMOID kernels for inference have significant speed-up in wall-clock time over those in FLASHATTENTION2 and the kernels for training also have modest wall-clock improvements.

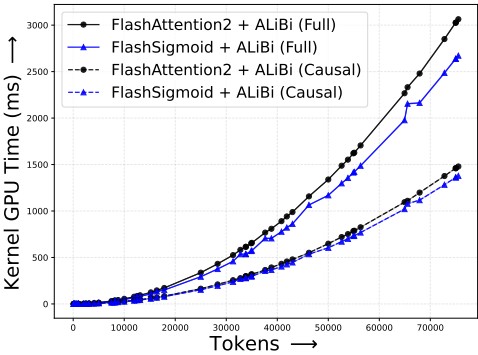 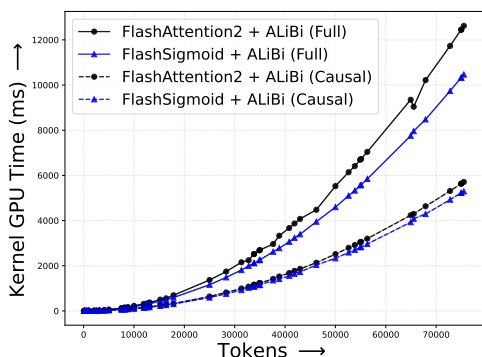

(a) Inference mode kernels on A100.  (b) Training mode kernels on A100.

Figure 16: On average, for sequence lengths between $[64, 78k]$, the inference mode kernel of FLASHSIGMOID is $12.28\%$ faster than FLASHATTENTION2 for self-attention and $5.30\%$ for causal attention. The training mode kernels of FLASHSIGMOID are $14.64\%$ faster than FLASHATTENTION2 for self-attention and $6.80\%$ for causal attention. Note that inference involves only the forward pass of the model and training involves both the forward and the backward pass of the model.

### F.5 DIRECTIONS FOR FUTURE WORK ON FLASHSIGMOID

In this section, we discussed FLASHSIGMOID, a hardware-aware implementation of the $\mathrm{SigmoidAttn}$ algorithm. Then, we demonstrated via kernel benchmarking and realistic setting runs that FLASHSIGMOID provides significant gains in inference as well as modest gains in training of models with attention mechanism. In this subsection we further discuss additional avenues for improving the implementation of FLASHSIGMOID, and point out some interesting directions for future work.

**Optimization of Block Shapes for Different Input and GPU Settings:**   As stated before, our FLASHSIGMOID implementation builds on FLASHATTENTION2 by adding functionality for forward and backward pass of sigmoid attention in place of the standard softmax attention. In particular, for all FLASHSIGMOID results discussed so far, we inherit directly from FLASHATTENTION2 the details of optimal block shapes, grid shapes, and other kernel launch parameters, and keep them unchanged in our implementation. For instance, this is the case for the block sizes $B_r, B_c$ in Alg. 1 and 2, which are identical in FLASHATTENTION2 and FLASHSIGMOID. This choice is dictated by the need to ensure a fair comparison between the two implementations, and allows us to demonstrate the speed-up of sigmoid attention by minimizing confounders associated with parallel computations on different GPU architectures for different input shapes.

Although FLASHSIGMOID kernels lead to speed-ups in inference and training for both H100 and A100 GPUs, we observe that the kernel timing speed-ups on A100 are not uniform across sequence lengths: for a small subset of these, our kernel provides significantly lower speed-up compared to the overall trend for other sequence lengths. Ideally, the implementation of attention mechanisms should not assume any information on the token count in input, and it is then desirable to have uniform speed-ups across all input lengths. Here, we show that this is achievable by simply updating the block shape information in FLASHSIGMOID to values that are different than those in FLASHATTENTION2. The implementation of FLASHATTENTION2 is templated according to block shapes, grid shapes, and other kernel launch parameters. Note that FLASHATTENTION2 provides various tailored implementations, optimized for different input shapes (e.g., different ranges of feature dimension per head), input types (e.g., causal attention vs. self-attention, ALiBi vs. no ALiBi in attention, etc.), and GPU types (e.g., A100 vs. H100 via checking shared memory size on GPUs). This is achieved by opportunely selecting the kernel template parameters defining block shapes, grid shapes, and other kernel launch parameters for parallel computation on GPUs. In our case, we create a variant of FLASHSIGMOID, denoted by FLASHSIGMOID$^{\dagger}$, where we update the block sizes for query and key tensors from $(B_r, B_c) = (128, 128)$ of FLASHSIGMOID to $(B_r, B_c) = (128, 64)$ of FLASHSIGMOID$^{\dagger}$ *only* for our input setting (template with features per head being 64).

**Experimentation and Results:**   For this variant, we perform kernel benchmarking as described in App. F.2, and report the corresponding results in Fig. 17. Comparing the plots for kernel timing

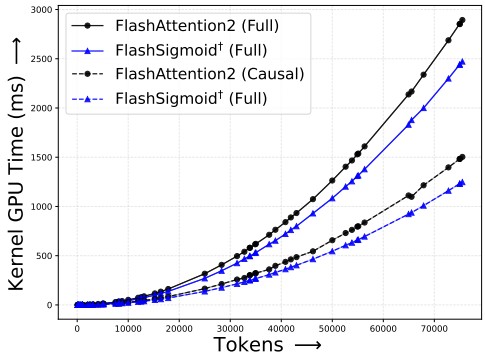
(a) Inference mode kernels on A100.

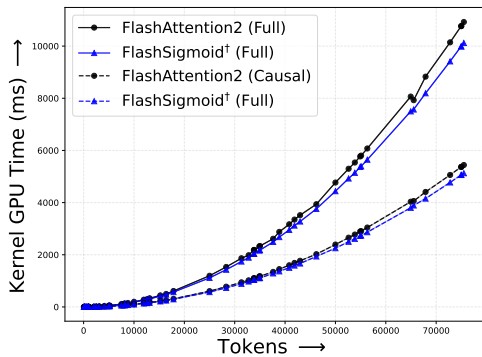
(b) Training mode kernels on A100.

Figure 17: On average, for sequence lengths between $[64, 78\text{k}]$, the inference mode kernel of FLASHSIGMOID[†] is $14.82\%$ faster than FLASHATTENTION2 for self-attention and $18.02\%$ for causal attention. The training mode kernels of FLASHSIGMOID[†] are $6.18\%$ faster than FLASHATTENTION2 for self-attention and $5.76\%$ for causal attention. Note that inference involves only the forward pass of the model and training involves both the forward and the backward pass of the model.

with FLASHSIGMOID plots from Fig. 12, we observe that FLASHSIGMOID[†] not only provides a more uniform inference and training kernel speed-up on *all* sequence lengths, but also improves the average of these speed-ups across all lengths. To further bolster our observations, Tab. 6 shows the inference mode and training mode kernel speed-ups for a subset of sequence lengths under consideration. This experiment indicates that it is possible to obtain higher and more uniform speed-ups in kernel timings across a wide range of tokens by investigating optimal block shape, grid shape, and other kernel launch parameters for each input setting and GPU type. We leave this optimization for future work.

| TOKENS | KERNEL GPU TIME COMPARISON | | | |
|---|---|---|---|---|
| | KERNELS | FLASHATTENTION2 (MS) | FLASHSIGMOID (MS) | FLASHSIGMOID[†] (MS) |
| 4096 | FWD | $8.32\pm0.02$ | $7.84\pm0.03\ (-\mathbf{5.79}\%)$ | $7.26\pm0.02\ (-\mathbf{13.21}\%)$ |
| | FWD + BWD | $31.81\pm0.08$ | $31.11\pm0.08\ (-\mathbf{2.19}\%)$ | $30.62\pm0.09\ (-\mathbf{4.03}\%)$ |
| 8100 | FWD | $33.65\pm0.09$ | $27.92\pm0.07\ (-\mathbf{17.04}\%)$ | $28.54\pm0.07\ (-\mathbf{15.50}\%)$ |
| | FWD + BWD | $128.18\pm0.13$ | $119.04\pm0.12\ (-\mathbf{7.13}\%)$ | $119.85\pm0.13\ (-\mathbf{6.81}\%)$ |
| 10000 | FWD | $51.17\pm0.07$ | $42.49\pm0.06\ (-\mathbf{16.96}\%)$ | $43.53\pm0.09\ (-\mathbf{15.32}\%)$ |
| | FWD + BWD | $194.54\pm0.14$ | $180.59\pm0.15\ (-\mathbf{7.17}\%)$ | $181.97\pm0.17\ (-\mathbf{6.87}\%)$ |
| 16384 | FWD | $134.19\pm0.12$ | $125.43\pm0.10\ (-\mathbf{6.53}\%)$ | $116.75\pm0.10\ (-\mathbf{13.40}\%)$ |
| | FWD + BWD | $494.65\pm0.28$ | $482.08\pm0.23\ (-\mathbf{2.54}\%)$ | $474.52\pm0.28\ (-\mathbf{4.48}\%)$ |

Table 6: FLASHATTENTION2 vs. FLASHSIGMOID vs. FLASHSIGMOID[†] on A100 nodes. The kernel GPU time for all three approaches are reported in milliseconds. We observe that FLASHSIGMOID[†] provides better and more uniform speed-ups across all example tokens.

# G EXPERIMENTS

## G.1 EXTRA ABLATIONS

### G.1.1 THE EFFECT OF MULTIPLICATIVE SEQUENCE LENGTH NORMALIZATION

Wortsman et al. (2023a) notes that models trained with sigmoid or ReLU attention require scaling by the sequence length, $n^{-\alpha}\sigma(\boldsymbol{QK}^T/\sqrt{d_{qk}})\boldsymbol{V}$. We ablate this by comparing the scaled solution to the one we propose in App. E. We also generalize the variant proposed in (Wortsman et al., 2023a) to variadic sequence lengths such that it works with auto-regressive (AR) training, for example for

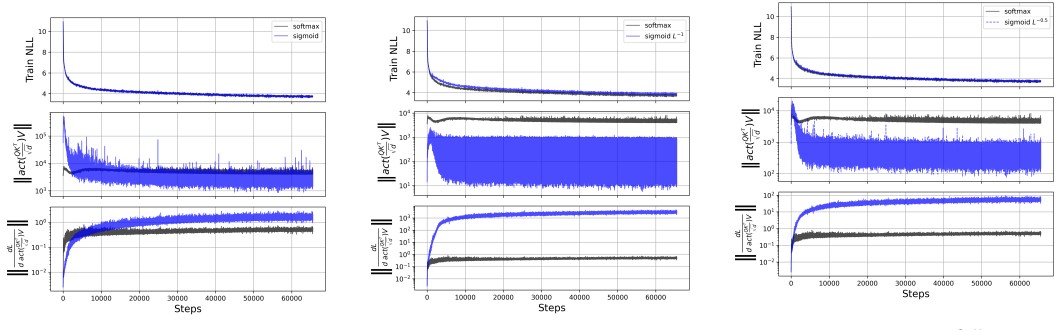

Figure 18: $b = -\ln n$.  Figure 19: $n^{-1}$ normalization.  Figure 20: $n^{-0.5}$ normalization.

$n = 3$:

$$\underbrace{\begin{bmatrix} 1 & 1 & 1 \\ 0.5^{-\alpha} & 0.5^{-\alpha} & 1 \\ 0.33^{-\alpha} & 0.33^{-\alpha} & 0.33^{-\alpha} \end{bmatrix}}_{n^{-\alpha}} \odot \underbrace{\begin{bmatrix} 1 & 0 & 0 \\ 1 & 1 & 0 \\ 1 & 1 & 1 \end{bmatrix}}_{\text{Causal Mask } \boldsymbol{M}} \odot \sigma(\boldsymbol{Q}\boldsymbol{K}^T/\sqrt{d_{qk}})\boldsymbol{V}. \tag{84}$$

We repeat the experiment from Fig. 5, using ALiBi positional embeddings for all trials. We apply $\alpha = \{1, 0.5\}$ AR normalization proposed in (84). While there is an observable difference in terms of the attention norm, $\|\sigma(\boldsymbol{Q}\boldsymbol{K}^T/\sqrt{d_{qk}})\boldsymbol{V}\|$, we find that the train NLL is slightly worse for both normalized variants (Fig. 19 and 20) in comparison to the $b = -\ln n$ variant in Fig. 18.

### G.1.2 ATTENTION BIAS STABILITY ABLATION

To validate the stabilizing effects of attention bias we repeat the experiment from Fig. 8 and 9, keeping all of the same hyper-parameters, while enabling QK norm and LayerScale (initialized at $10^{-4}$). We train with a range of constant bias offsets, $b \in \{-15, -10, -6, -4, -1\}$ and visualize the results below in Fig. 21. We observe a systematic increase in stability (and lower $\mathrm{SigmoidAttn}$ NLL) for

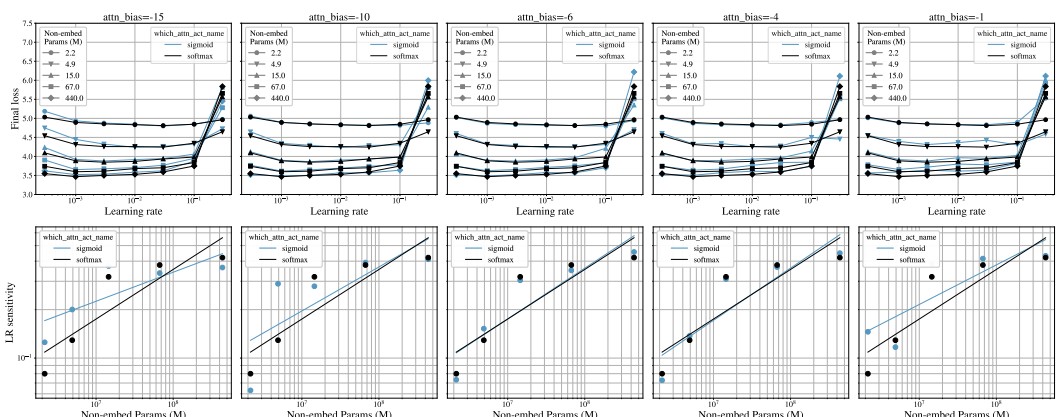

Figure 21: Attention bias ablation.

values less than $-1$ up till $-10$, after which the $-15$ plot shows an over-regularizing effect with decreased performance.

### G.2 VISION

### G.2.1 TEST IMAGENET1K TOP-1%

Fig. 22 reports the test linear probe results for the ViT-B/16 BYOL (Grill et al., 2020; Busbridge et al., 2023), ViT-B/16 SimCLR (Chen et al., 2020; Zhai et al., 2023a) and the finetuned performance

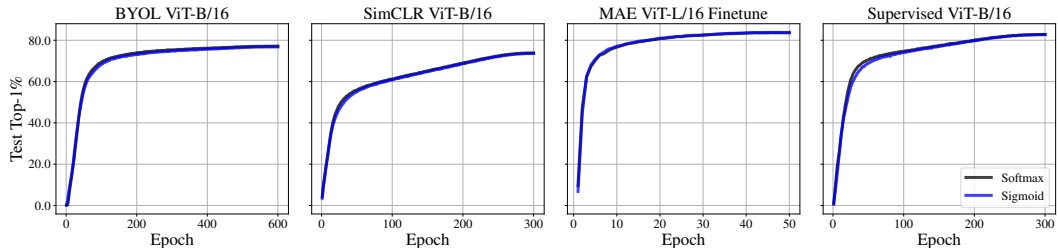

Figure 22: ImageNet1k test top-1% for SoftmaxAttn vs. SigmoidAttn using models from Fig. 2.

for the ViT-L/16 MAE (He et al., 2022) and the test top-1% results for for ViT-B/16 supervised model (Dosovitskiy et al., 2021). Across these wide range of SSL and supervised learning tasks, trained with contrastive (SimCLR), EMA distillation (BYOL) and reconstructive objectives (MAE), we find that SigmoidAttn not only matches the training dynamics (Fig. 2), but also the linear probe and finetuned performance of the baseline SoftmaxAttn.

## G.2.2 LAYERSCALE FREE SIGMOID ATTENTION

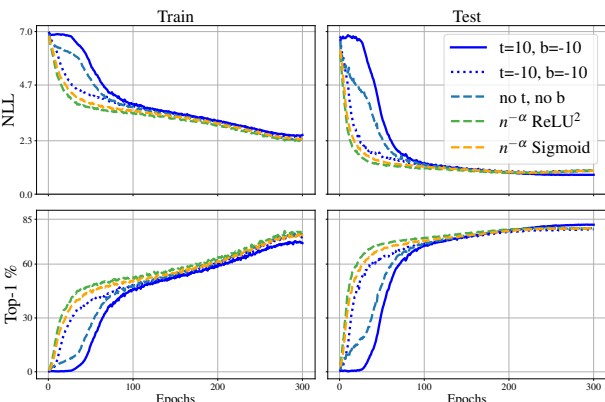

Figure 23: A competitive SigmoidAttn ViT-B/16 model can be learned without LayerScale or QK norm using a large initial *learnable* scalar temperature $t = 10$ and bias $b = -10$ (similar to SigLIP (Zhai et al., 2023b)): $\sigma(e^t[\boldsymbol{QK}^T/\sqrt{d_{qk}}] + b)\boldsymbol{V}, \{b, t\} \in \mathbb{R}$. This regularizes the model, as it must move the temperature to a learnable regime. The $t = 10, b = -10$ curve makes no progress in train NLL or test top-1 for ~25 epochs (near max LR), but ultimately outperforms baselines.

While Fig. 23 demonstrates the possibility of learning SigmoidAttn without LayerScale, it involves task specific tuning of $\{t, b\}$. We also explored gating attention from learning (through a simple multiply by zero) for ~25 epochs and were able to match the $t = 10, b = -10$ training curves from above. However, we opted for the LayerScale method due to its simplicity.

## G.2.3 SIGMOID ATTENTION VS. ATTENTION RELAXATIONS

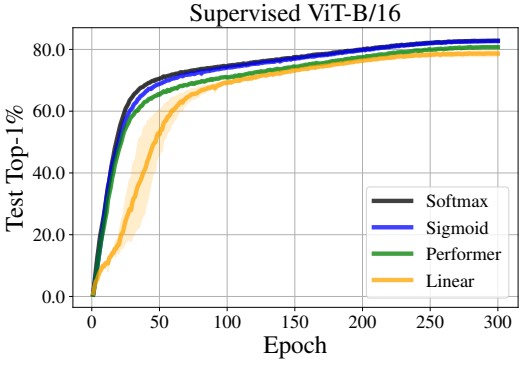

Figure 24: Supervised ViT-B/16 ImageNet1k classification. We contrast SigmoidAttn and SoftmaxAttn against (a) linear attention with no activation: $\boldsymbol{QK}^T/\sqrt{d_{qk}}$ and (b) fast attention via positive orthogonal random features, used in Performer (Choromanski et al., 2021). SigmoidAttn, like SoftmaxAttn, differs from attention relaxations like Performer which uses low-rank representations of the attention matrix. SigmoidAttn maintains performance parity with SoftmaxAttn, while outperforming other efficient attention variants.

### G.2.4 HYPER-PARAMETERS

Table 7: SigmoidAttn SimCLR and BYOL ViT-B/16 hyperparameters.

| Parameter | SimCLR | BYOL |
|---|---|---|
| Attention bias | None | None |
| LayerScale Init | $10^{-4}$ | $10^{-4}$ |
| QK Norm | Yes | Yes |
| Pos Embed | SinCos | Learnable |
| Freeze Patcher | Yes | No |
| Weight init | MocoV3 (Chen et al., 2021) | `trunc_normal(.02)` |
| Normalization | LayerNorm | LayerNorm |
| LR schedule | Single Cycle Cosine | Single Cycle Cosine |
| LR warmup | 10 Epochs | 40 Epochs |
| Min LR | $1 \times 10^{-6}$ | $1 \times 10^{-6}$ |
| Training duration | 300 Epochs | 600 Epochs |
| Optimizer | AdamW | AdamW |
| Optimizer scaling rule | Linear | Linear |
| Base Adam $(\beta_1, \beta_2)$ | (0.9, 0.95) | (0.9, 0.95) |
| Base LR | $2 \times 10^{-4}$ | $1 \times 10^{-4}$ |
| Base batch size | 256 | 256 |
| Total batch size | 4096 | 4096 |
| Base teacher momentum | - | 0.996 |
| Weight decay | 0.1 | 0.3 |
| Weight decay skip bias | Yes | Yes |
| Numerical precision | `bf16` | `bf16` |
| Stochastic depth | 0.0 | 0.2 |
| Augmentation stack | SimCLR (Chen et al., 2020) | DINO multicrop (Caron et al., 2021) |
| Color Jitter Scaling | 0.5 (Chen et al., 2021) | 1.0 |

Table 8: SigmoidAttn Supervised ViT-B/16 and MAE ViT-L/16 hyperparameters.

| Parameter | Supervised | MAE |
|---|---|---|
| Attention bias | None | $b = -\ln n$ |
| LayerScale Init | $10^{-4}$ | $10^{-4}$ |
| QK Norm | Yes | Yes |
| Pos Embed | Learnable | Learnable |
| Architecture | ViT-B/16 | ViT-L/16 |
| Mask Ratio | - | 0.75 |
| Freeze Patcher | No | No |
| Weight init | `trunc_normal(.02)` | `trunc_normal(.02)` |
| Normalization | LayerNorm | LayerNorm |
| LR schedule | Single Cycle Cosine | Single Cycle Cosine |
| LR warmup | 20 Epochs | 40 Epochs |
| Min LR | $1 \times 10^{-6}$ | 0.0 |
| Training duration | 300 Epochs | 400 Epochs |
| Optimizer | AdamW | AdamW |
| Optimizer scaling rule | Linear | Linear |
| Base Adam $(\beta_1, \beta_2)$ | (0.9, 0.95) | (0.9, 0.95) |
| Base LR | $1 \times 10^{-4}$ | $1.5 \times 10^{-4}$ |
| Base batch size | 256 | 256 |
| Total batch size | 4096 | 4096 |
| Weight decay | 0.3 | 0.05 |
| Weight decay skip bias | Yes | Yes |
| Numerical precision | `bf16` | `bf16` |
| Stochastic depth | 0.28 | 0.0 |
| Augmentation stack | RandAug (Cubuk et al., 2020) | RRC + HFLIP |

## G.3 LANGUAGE MODEL

### G.3.1 HYPER-PARAMETERS

Tab. 9 shows the hyper-parameters for the final comparison. MuP-simple (Wortsman et al., 2023b) is used, where the peak learning rate is set to 1e-2. Weight decay is decoupled, following Loshchilov & Hutter (2017). In addition, to confirm that applying QK-Norm does not hurt the baseline, we show training parity with and without QK-Norm in App. G.3.1.

Table 9: Training details for the Llama-style 1B and 7B LM training.

| Parameter | 1B Value | 7B Value |
|---|---|---|
| Params | 1B | 7B |
| Context Length | 2048 | 4096 |
| Total Tokens | 300B | 1T |
| Batch size | 4M tokens | 4M tokens |
| LR Schedule | Cosine | Cosine |
| LR Warmup Steps | 5000 | 5000 |
| Peak LR | 1e-2 | 1e-2 |
| Final LR | 10% of peak | 10% of peak |
| Optimizer | AdamW | AdamW |
| Optimizer momentum | 0.9, 0.95 | 0.9, 0.95 |
| Weight decay | 1e-4 | 1e-4 |
| Gradient clipping | 1.0 | 1.0 |
| Position encoding | ALiBi | ALiBi |
| Q/K Norm | Applied | Applied |
| Norm type | RMSNorm (Zhang & Sennrich, 2019) | RMSNorm (Zhang & Sennrich, 2019) |
| Norm structure | Pre-norm | Hybrid-norm |
| Num layers | 24 | 32 |
| Num heads | 32 | 32 |
| Hidden dim | 2048 | 4096 |

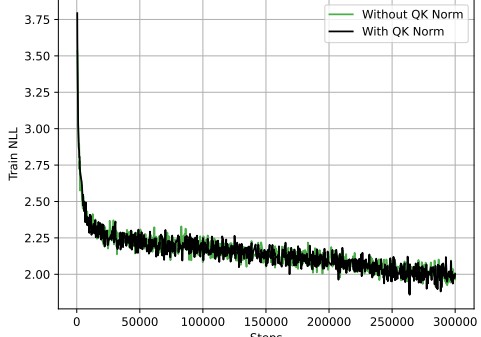 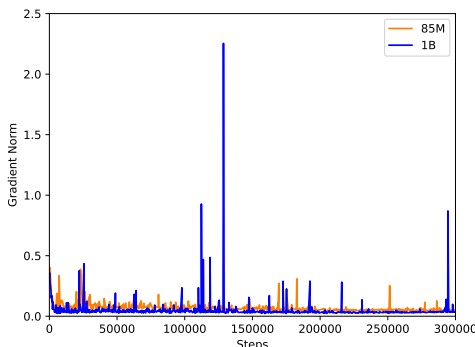

Figure 25: Left: 1B SoftmaxAttn LLM training with and without QK Norm converging to the same loss. Right: 85M and 1B LLM training using SigmoidAttn (n = 4096) without hybrid-norm. Smooth training loss curves, but gradient norm shows spikes. Hybrid norm mitigates gradient spikes.

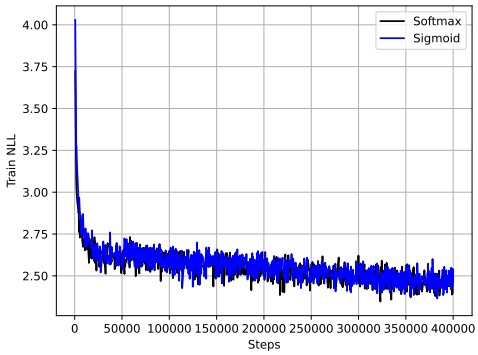 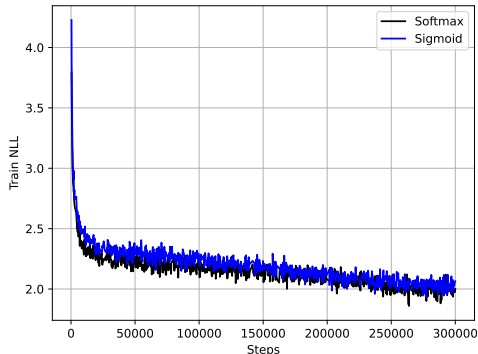

Figure 26: Left: 85M training using SigmoidAttn and SoftmaxAttn (n = 4096) without hybrid-norm. Training loss matches. Right: 1B training using SigmoidAttn (n = 4096). Higher sequence length with a larger model shows a slightly different loss curve. Hybrid norm resolves the issues enabling stable training up to 7B (Table 2 and Fig. 2).

### G.3.2 GRADIENT NORM

While a SigmoidAttn based LM using aforementation hyper-parameters has a smooth loss curve, we do see more gradient norm fluctuations. See App. G.3.1, where spikes larger than 0.5 are not visible in the SoftmaxAttn equivalent.

Table 10: Different norm structure ablations for SigmoidAttn with 1B language-modeling.

| Model | Seq. Len. | Attn. Bias | Pos. Encod. | Norm | ARC Easy | ARC Chall. | Hella-swag | PIQA | SciQ | Wino-grande | Lambada OpenAI | TriviaQA (1-shot) | WebQS (1-shot) | Avg |
|---|---|---|---|---|---|---|---|---|---|---|---|---|---|---|
| Soft. | 2k | No | AliBi | Pre | 62.2 | 26.8 | 42.4 | 59.0 | 72.3 | 88.1 | 58.4 | 19.9 | 15.4 | 49.4 |
| Soft. | 2k | No | RoPE | Pre | 64.5 | 30.4 | 43.9 | 61.0 | 71.9 | 88.7 | 59.3 | 21.1 | 15.0 | 50.6 |
| Sigm. | 2k | Yes | AliBi | Pre | 62.8 | 28.8 | 42.5 | 59.7 | 70.3 | 88.6 | 59.7 | 19.1 | 13.8 | 49.5 |
| Sigm. | 2k | Yes | RoPE | Pre | 62.2 | 26.9 | 41.4 | 57.9 | 71.1 | 87.8 | 57.3 | 17.3 | 12.8 | 48.3 |
| Sigm. | 2k | No | RoPE | Pre | 59.3 | 26.5 | 39.4 | 55.1 | 69.4 | 88.2 | 59.2 | 11.7 | 7.5 | 46.0 |
| Soft. | 2k | No | AliBi | Hybrid | 64.9 | 30.5 | 43.3 | 61.9 | 71.6 | 88.4 | 60.9 | 23.6 | 12.8 | 50.9 |
| Sigm. | 2k | Yes | AliBi | Hybrid | 60.5 | 26.9 | 42.2 | 59.2 | 70.8 | 89.6 | 57.9 | 17.7 | 13.4 | 48.7 |
| Sigm. | 2k | No | AliBi | Hybrid | 62.8 | 28.2 | 42.0 | 59.1 | 70.3 | 88.7 | 59.8 | 18.6 | 15.1 | 49.4 |
| Soft. | 4k | No | RoPE | Pre | 63.3 | 29.3 | 43.3 | 58.1 | 71.3 | 86.9 | 58.8 | 20.4 | 15.6 | 49.7 |
| Soft. | 4k | No | AliBi | Pre | 62.6 | 27.7 | 42.4 | 58.6 | 71.1 | 88.2 | 58.6 | 18.9 | 14.7 | 49.2 |
| Sigm. | 4k | Yes | AliBi | Pre | 60.5 | 27.3 | 41.3 | 57.8 | 70.5 | 87.0 | 57.6 | 18.9 | 12.6 | 48.2 |
| Soft. | 4k | No | RoPE | Hybrid | 64.1 | 27.2 | 43.3 | 61.4 | 71.2 | 88.5 | 60.0 | 21.4 | 15.3 | 50.3 |
| Soft. | 4k | No | AliBi | Hybrid | 61.7 | 26.8 | 43.4 | 59.4 | 70.6 | 88.6 | 60.8 | 20.5 | 12.9 | 49.4 |
| Sigm. | 4k | No | RoPE | Hybrid | 63.3 | 27.1 | 43.4 | 61.3 | 70.4 | 88.2 | 57.5 | 20.5 | 14.8 | 49.6 |
| Sigm. | 4k | Yes | AliBi | Hybrid | 63.5 | 28.1 | 43.5 | 60.7 | 70.8 | 88.9 | 59.0 | 20.9 | 16.0 | 50.2 |
| Sigm. | 4k | No | AliBi | Hybrid | 62.4 | 28.9 | 43.5 | 60.8 | 71.3 | 89.6 | 59.2 | 20.2 | 14.3 | 50.0 |

### G.3.3 Norm structure

Due to the slight performance difference observed at 4096 context length when using SigmoidAttn versus SoftmaxAttn, and marginally lower downstream results, we evaluated various norm structures to address potential instabilities (see Tab. 10). Some of these structures replace the required attention bias (in this case, column 'Attn. Bias' is 'No'). All use QK-norm with RMSNorm (Zhang & Sennrich, 2019), without LayerScale. We examined pre-norm and hybrid-norm (where we do both pre-norm and normalization of the output of the attention layer following Xiong et al. (2020)): $\text{norm}(\sigma(\boldsymbol{Q}\boldsymbol{K}^T/\sqrt{d_{qk}})\boldsymbol{V})$. Post-norm, which normalizes the combined residual data stream, $\text{norm}(x+\sigma(\boldsymbol{Q}\boldsymbol{K}^T/\sqrt{d_{qk}})\boldsymbol{V})$, is omitted from our analysis as it did not train stably for SigmoidAttn.

### G.4 Automatic Speech Recognition

### G.4.1 Training Details

All acoustic models are fed 80 channel log-mel filterbanks with a 25ms sliding window strided by 10ms.

The transformer-based encoder model has 255M parameters: 1D convolution of kernel 7 and stride 3 followed by CAPE positional embedding if it is used and 36 transformer blocks with pre-LayerNorm, an embedding dimension of 768, 4 heads, 3072 units in the MLP layers. The model is trained with CTC loss and a character vocabulary, including apostrophe ('). In additional experiments, we vary the depth to 12 and 24 layers, and change pre-LayerNorm to post-LayerNorm.

We implemented our own conformer-based encoder model, also trained with a CTC loss and a character vocabulary. The conformer model has 104M parameters and consists of 1D convolution of kernel 7 and stride 3 followed by 16 conformer blocks with an embedding dimension of 512, 4 heads, 2048 units in the MLP layers. Variational noise is not used and RoPE is used as a relative positional embedding instead of relative sinusoidal positional embedding.

For all models, SpecAugment (Park et al., 2019) is used for augmentation with 2 frequency masks (max width 30) and 10 time masks (max width 50, ratio 0.1). All models are trained with dynamic batching and mixed precision with BF16. Models are trained with different configurations of optimizers and hyperparameters to have diverse coverage of use-cases. ***We first optimize every configuration for*** SoftmaxAttn ***and then change only attention to the introduced configuration of*** SigmoidAttn ***while all other parameters are kept the same.*** Detailed configurations are shown in Table 11. We train models until the greedy WER stops improving on the validation sets (*dev-clean, dev-other*) and report final test sets (*test-clean, test-other*) greedy WER without integration of any external language model.

Table 11: Training details for the ASR models on LibriSpeech 100h (LS-100) and LibriSpeech 960h (LS-960) for transformers and conformers.

| Parameter | Transformer LS-960 | Conformer LS-960 | Transformer LS-100 | Transformer LS-100 |
|---|---|---|---|---|
| Params | 255M | 104M | 255M / 170M / 85M | 255M |
| LayerNorm | pre | pre + post | pre | post |
| Dropout | 0.1 | 0.1 | 0.3 | 0.3 |
| Layer drop | 0.1 | 0.0 | 0.3 | 0.3 |
| Training steps | 400k | 400k | 400k | 500k |
| Batch size | 3.56h | 4.44h | 1.1h | 1.1h |
| LR schedule | step-wise | step-wise | step-wise | step-wise |
| SpecAugment start | 0k | 10k | 0k | 0k |
| LR Warmup Steps | 64k | 10k | 64k | 64k |
| Peak LR | 1e-3 | 2e-3 | 0.1 | 0.03 |
| LR start decay | 250k | 250k | 200k | 330k |
| LR decay step | 50k | 50k | 30k | 50k |
| Optimizer | AdamW | AdamW | Adagrad | Adagrad |
| Optimizer momentum | 0.9, 0.999 | 0.9, 0.98 | - | - |
| Weight decay | 1e-6 | 1e-6 | 0 | 0 |
| Gradient clipping | 1.0 | 0.5 | 1.0 | 1.0 |
| Position encoding | CAPE / ALiBi / RoPE | RoPE | CAPE | CAPE / ALiBi / RoPE |
| Q/K Norm SoftmaxAttn | Not Applied | Not Applied | Not Applied | Not Applied |
| Q/K Norm SigmoidAttn | Applied | Applied | Not Applied | Applied |
| Num layers | 36 | 16 | 36 / 24 / 12 | 36 |
| Num heads | 4 | 4 | 4 | 4 |

Table 12: Word error rate (%) on LibriSpeech dev/test sets and TED-LIUM v3 (Hernandez et al., 2018) ("TED", joint validation and test sets with split according to audio duration) for pre-LayerNorm transformer (255M / 170M / 85M params) with CAPE and with either SoftmaxAttn or SigmoidAttn (w/ LayerScale, w/o QK norm, w/ $b = -\log n$) trained on LibriSpeech 100h data (average duration is 10-15s). Hyper-parameters can be found in Table 11.

| ATTN | # LAYERS | DEV-CLEAN | TEST-CLEAN | DEV-OTHER | TEST-OTHER | TED 0-10S | TED 10-20S | TED 20-30S | TED 30S+ |
|---|---|---|---|---|---|---|---|---|---|
| SOFTMAX | 36 | 6.7 | 7.1 | 20.0 | 20.4 | 26.4 | 22.4 | 23.3 | 21.8 |
| SIGMOID | 36 | 7.0 | 7.3 | 20.3 | 20.5 | 26.2 | 23.4 | 23.6 | 21.8 |
| $b = 0$ | 36 | 6.8 | 7.1 | 19.8 | 20.3 | | | | |
| SOFTMAX | 24 | 6.4 | 6.8 | 20.2 | 20.5 | 25.4 | 22.1 | 23.3 | 21.8 |
| SIGMOID | 24 | 7.1 | 7.3 | 21.0 | 21.3 | 26.6 | 23.3 | 24.0 | 22.0 |
| $b = 0$ | 24 | 6.7 | 6.9 | 20.2 | 20.7 | | | | |
| SOFTMAX | 12 | 8.2 | 8.7 | 25.0 | 25.4 | 29.0 | 25.6 | 27.1 | 27.4 |
| SIGMOID | 12 | 8.3 | 8.7 | 24.8 | 25.2 | 29.0 | 25.7 | 26.3 | 25.5 |
| $b = 0$ | 12 | 8.7 | 8.5 | 24.4 | 24.7 | | | | |

For the bias term $b = -\log n$ in SigmoidAttn, we do not use max sequence length as in language model experiments. Instead, for every audio sample we use its own duration as a bias terms resulting into non-trainable bias vector for the minibatch. For experiments with sequence normalization, we also use not the max sequence length in the minibatch but rather the ground truth sample duration to properly normalize encoder attention.

To evaluate behaviour for length generalization we use TED-LIUM v3 dataset Hernandez et al. (2018) as its validation and test sets have longer audio duration than LibriSpeech: LibriSpeech has in average 10-15s duration, while in TED-LIUM there are audio longer than 30s (the max duration of LibriSpeech). To perform evaluation on TED-LIUM v3, we combine together validation and test sets of TED-LIUM v3 (we don't use them for training and hyper-parameters search and just perform final evaluation) and split them into 4 datasets according to the duration: 0-10s, 10-20s, 20-30s, and 30s+.

For positional embeddings we use not only CAPE, but change it to ALiBi or RoPE. As ALiBi was originally introduced for the decoder only models and there is no official adoption of it yet[14] for the encoder models (without causal masking), we follow the best practices found in https://iclr-blogposts.github.io/2024/blog/alibi-mlm/ of nonsymmetric ALiBi with different slopes instead of symmetric version used by (Lee et al., 2022).

---

[14]See discussion in https://github.com/ofirpress/attention_with_linear_biases/issues/5.

Table 13: Word error rate (%) on LibriSpeech dev/test sets for post-LayerNorm transformer (255M) with either SoftmaxAttn (w/o QK norm) or SigmoidAttn (by default w/ LayerScale, w/ QK norm, w/ $b = -\log n$) trained on LibriSpeech 100h data. Hyper-parameters can be found in Table 11.

| ATTN | PE | DEV-CLEAN | TEST-CLEAN | DEV-OTHER | TEST-OTHER |
|---|---|---|---|---|---|
| SOFTMAX | CAPE | 6.4 | 6.5 | 18.4 | 18.2 |
| + QK NORM | | 6.1 | 6.3 | 18.2 | 18.1 |
| SIGMOID | | 8.0 | 8.4 | 22.7 | 22.7 |
| - QK NORM | | 7.5 | 7.9 | 22.1 | 27.6 |
| - LAYERSCALE | | UNSTABLE, GRADIENT NORM AND LOSS SPIKES | | | |
| - QK NORM - LAYERSCALE | | 6.5 | 6.9 | 19.9 | 20.1 |
| SIGMOID ($b = -10$, LEARNABLE) | | 8.7 | 9.4 | 23.5 | 24.0 |
| SOFTMAX | RoPE | 6.6 | 6.9 | 18.3 | 18.5 |
| SIGMOID | | 6.8 | 7.1 | 20.8 | 20.8 |
| SIGMOID ($b = -10$, LEARNABLE) | | 8.7 | 9.4 | 23.5 | 24.0 |
| SOFTMAX | ALIBI | 6.4 | 6.9 | 18.3 | 18.3 |
| SIGMOID | | 6.9 | 7.2 | 20.8 | 21.1 |
| SIGMOID ($b = -10$, LEARNABLE) | | 6.8 | 7.1 | 20.4 | 20.5 |

Table 14: Word error rate (%) on LibriSpeech dev/test sets and TED-LIUM v3 (Hernandez et al., 2018) ("TED", joint validation and test sets with split according to audio duration) for conformer (104M) with RoPE and with either SoftmaxAttn or SigmoidAttn (w/ LayerScale, w/ QK norm, w/ $b = -\log n$) trained on LibriSpeech 960h data (average duration is 10-15s). Hyper-parameters can be found in Table 11.

| ATTN | DEV-CLEAN | TEST-CLEAN | DEV-OTHER | TEST-OTHER | TED 0-10S | TED 10-20S | TED 20-30S | TED 30S+ |
|---|---|---|---|---|---|---|---|---|
| SOFTMAX | 2.2 | 2.5 | 5.4 | 5.6 | 13.0 | 11.1 | 13.2 | 7.1 |
| SIGMOID | 2.3 | 2.5 | 5.6 | 5.8 | 13.5 | 10.8 | 13.3 | 10.2 |
| SIGMOID ($b = -10$, LEARNABLE) | 2.4 | 2.7 | 5.8 | 5.8 | 12.9 | 11.1 | 14.1 | 54.9 |

### G.4.2 RESULTS AND ABLATIONS

Initial investigation on post-LayerNorm and pre-LayerNorm transformers on both LibriSpeech 100h and 960h revealed that SigmoidAttn without any bias is unstable resulting in huge and frequent gradient norm and training loss spikes throughout the training which in turn result in spikes of validation and test WER, see Figure 27. Neither LayerScale nor QK norm were able to stabilize the training, though we did not observe any model divergence.

Further experiments with bias term in the SigmoidAttn definition for post-LayerNorm transformers on LibriSpeech 100h reveal that training is now stable (only few marginal spikes in gradient norm occur, while train loss is smooth all the time). However, both LayerScale and QK norm restrict model capacity thus not matching SoftmaxAttn. Moreover, some combination of them is needed for the stable training, though w/o both of them we got the best performance for SigmoidAttn (still behind SoftmaxAttn), see Table 13. We believe, further adaptation and deeper investigation is needed for SigmoidAttn and post-LayerNorm, though recent advances in machine learning do not use post-LayerNorm models due to high training instability even for SoftmaxAttn.

Switching to pre-LayerNorm transformers and varying the depth of the models lead to stable training with SigmoidAttn and bias term $b = -\log n$ with few (2-5 times) spikes in the gradient norm and smooth loss. In this case, SigmoidAttn matches results for SoftmaxAttn and they both generalize to TED-LIUM data similarly, see Table 12. If the bias term is removed, SigmoidAttn can still match SoftmaxAttn but large spikes in gradient norm and loss can occur.

Finally, we experiment with a conformer model, in Table 14. Again, we found that bias term $b = -\log n$ stabilizes training. The learnable $b = -10$ works though we see significant gradient norm spikes while the train loss remains smooth. Besides, $b = -\log n$ generalizes well to longer sequences while learnable $b = -10$ fails to do so with RoPE for conformer. Overall, SigmoidAttn is able to match SoftmaxAttn having stable training with $b = -\log n$.

In experiments with different variants of bias term for SigmoidAttn, the bias $b = -\log n$ is found to be the most stable (only few marginal gradient norm spikes are observed with the train loss

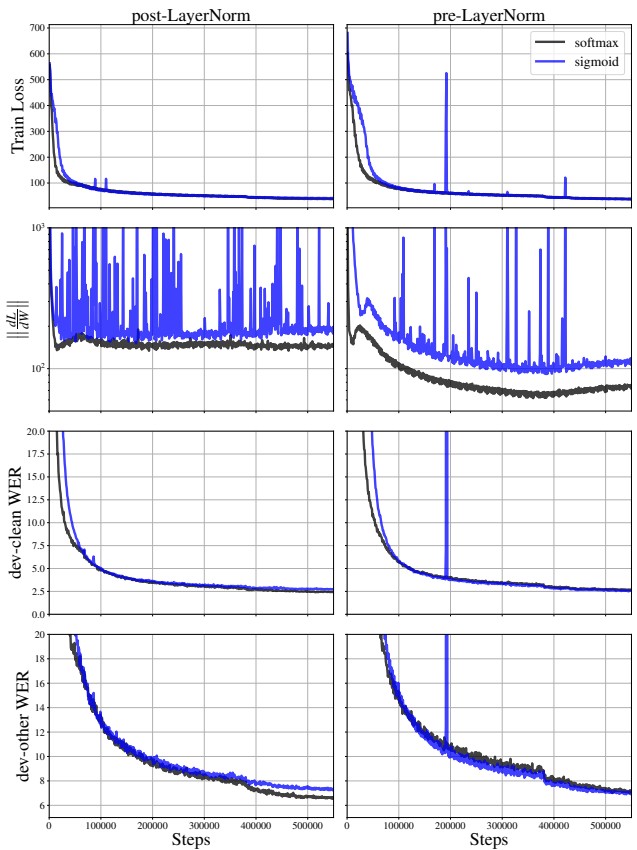

Figure 27: ASR Transformer model (255M) training with post-LayerNorm (left) and pre-LayerNorm (right) on LibriSpeech 960h with SigmoidAttn (w/ bias term, $b = 0$, w/o QK norm, w/ LayerScale) or with SoftmaxAttn. Huge gradient norms and training loss spikes are observed for SigmoidAttn which can result in worse final model performance hence models for SigmoidAttn are unstable.

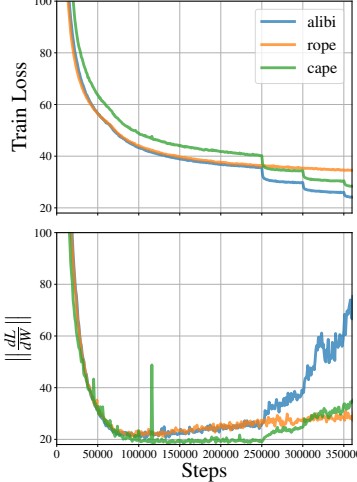

Figure 28: ASR Transformer model (255M) training with pre-LayerNorm on LibriSpeech 960h with SigmoidAttn (w/ bias term, $b = -\log n$, w/ QK norm, w/ LayerScale) and different positional embeddings CAPE, RoPE, ALiBi. The bias $b$ is able to stabilize SigmoidAttn training: smooth training loss and only marginal rare spikes in gradient norms are observed.

being smooth) and it provides similar performance as SoftmaxAttn in most settings. The source of instability is coming from the larger attention output norms (80k for CAPE, 40k for RoPE and 20k for AliBi while being 200 for SoftmaxAttn). This happens due to high attention weight of every token which can be biased towards zero with a bias term in SigmoidAttn definition. Preliminary results to connect this to the local attention property needed at the beginning of the training for stable training failed, as local attention did not converge well at all (it is deactivated after some initial training).

Table 15: Word error rate (%) on LibriSpeech dev/test sets and TED-LIUM v3 (Hernandez et al., 2018) ("TED", joint validation and test sets split according to duration) for transformer (255M params) with either SoftmaxAttn or SigmoidAttn (LayerScale and QK norm are used with $b = -\log n$) trained on LibriSpeech 960h data (mean duration is 10-15s). Hyper-parameters are in App. G.4. H-Norm corresponds to hybrid-norm, no LS-Attn corresponds to removing the LayerScale from the attention outputs, and no LS corresponds to removing the LayerScale from both the attention and MLP outputs.

| ATTN | PE | DEV-CLEAN | TEST-CLEAN | DEV-OTHER | TEST-OTHER | TED 0-10S | TED 10-20S | TED 20-30S | TED 30S+ |
|---|---|---|---|---|---|---|---|---|---|
| SOFTMAX | | 2.2 | 2.3 | 5.6 | 5.7 | 12.4 | 10.5 | 11.9 | 9.1 |
| SIGMOID | CAPE | 2.2 | 2.4 | 5.2 | 5.5 | 12.4 | 10.3 | 12.3 | 9.7 |
| SIGMOID, $b = -\log(\max_{batch} n)$ | | 2.1 | 2.3 | 5.2 | 5.3 | 12.2 | 10.6 | 12.0 | 9.3 |
| SOFTMAX | | 2.2 | 2.2 | 5.4 | 5.5 | 12.7 | 10.6 | 12.8 | 9.5 |
| SIGMOID | | 2.0 | 2.3 | 5.2 | 5.4 | 12.3 | 10.1 | 12.3 | 8.6 |
| SIGMOID, $b = -\log(\max_{batch} n)$ | RoPE | 2.1 | 2.3 | 5.0 | 5.1 | 12.3 | 10.1 | 12.1 | 10.4 |
| SIGMOID (H-NORM), NO QK-NORM, NO LS-ATTN | | 2.1 | 2.2 | 5.0 | 5.0 | 11.8 | 10.2 | 12.3 | 10.8 |
| SIGMOID (H-NORM), NO LS-ATTN | | 2.1 | 2.3 | 5.0 | 5.1 | 12.0 | 10.2 | 12.4 | 11.4 |
| SIGMOID (H-NORM), NO QK-NORM, NO LS | | 2.2 | 2.3 | 5.6 | 5.6 | 13.2 | 10.9 | 13.5 | 11.5 |
| SOFTMAX | | 2.1 | 2.2 | 5.3 | 5.4 | 12.3 | 10.7 | 12.1 | 8.6 |
| SIGMOID | ALiBi | 2.1 | 2.3 | 5.0 | 5.1 | 12.3 | 10.5 | 12.6 | 9.1 |
| SIGMOID, $b = -\log(\max_{batch} n)$ | | 2.0 | 2.3 | 5.2 | 5.2 | 12.3 | 10.5 | 11.9 | 10.2 |

To fully benefit from the improved throughput of FLASHSIGMOID, for the bias term $b = -\log n$ in SigmoidAttn, we experimented with configuration when the maximum audio duration in the minibatch is used as $n$ resulting into non-trainable bias scalar which changes between minibatches as we use dynamic batching. Comparison between the bias vector with per sample own duration normalization and the bias scalar as maximum duration in the minibatch is shown in Table 15: final model performance is similar and stability is same (only 2-3 minor spikes in CAPE for gradient norms are observed). Thus, per batch maximum audio duration can be used with $b = -\log n$ as the final configuration.

We also experimented with hybrid-norm (see Table 16) to check if it is able to stabilize the attention magnitudes as well as gradient norms. We did ablation with configuration similar to Table 16 with the following changes: LayerScale after attention is replaced to LayerNorm, only RoPE is used for positional embedding; we either keep or remove QK-norm and we either keep or remove LayerScale in MLP part of transformer block.

First, for all variants we observe that training loss and gradient norms are smooth without any spikes during training while we see abnormally large attention activations compared to all prior experiments. Second, while we observe that QK-norm or its removal behave similarly, the LayerScale on top of MLP output is necessary to get performance on par with SoftmaxAttn or with SigmoidAttn with bias term.

### G.5 SIMPLE EXPERIMENTS

#### G.5.1 k−SUMMATION PROBLEM DEFINITION

Here we look at a synthetic, simple task in order to investigate the behavior of softmax and sigmoid attention activations. The problem chosen is to minimize the MSE loss of a $\mathbb{R}^n \to \mathbb{R}$ target function. In the first half of each input are samples from a $\mathcal{N}(0, 1)$ distribution, and the second half is a $k$-hot binary vector indicating which values in the first half to sum.

The results presented here are for the $n = 40$ problem with various values for $k$. Where a transformer is used, the transformer is a single layer to aid visualization. In all cases (unless noted otherwise), the optimizer is Adam with a constant learning rate of 0.001, and the training data is continuously generated to preclude over-fitting.

A few examples for $n = 10$ (not drawn from $\mathcal{N}(0, 1)$) are shown below. Inputs in the second half of the input are show in orange only as a visual aid.

```
1 2 3 4 5 0 0 0 0 1 → 5
1 2 3 4 5 1 0 0 0 1 → 6
8 1 2 0 5 0 1 1 1 0 → 3
2 0 2 2 2 1 1 0 1 0 → 4
```

### G.5.2 COMPARISON TO SOFTMAX

In Figure 29, we see the performance of three architectures on the $k$-summation problem as $k$ increases. The sigmoid activated transformer has similar scaling to the softmax activation.

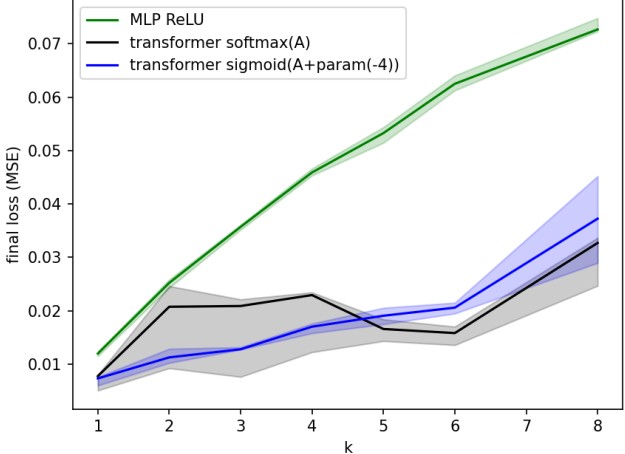

Figure 29: Final loss is shown after training convergence as k-summation problem complexity increases. The ReLU MLP has two hidden layers (900, 300) for 307k parameters, while the transformer has an embedding dimension of 120, 8 heads, and an MLP ratio of 4, giving 187k parameters. The SigmoidAttn is applied after a learned offset initialized to -4, `A+param(-4)`.

### G.5.3 ATTENTION EVOLUTION

In Figures 30 and 31, forty samples are used to monitor the single head, single layer post-activation attention matrix as training progresses. In Figure 30, the distribution of values is visualized over time; note the sigmoid attention is more variable but reaches comparable values at convergence. The main difference at convergence is that the sigmoid has fewer high magnitude values than softmax indicating a more distributed attention.

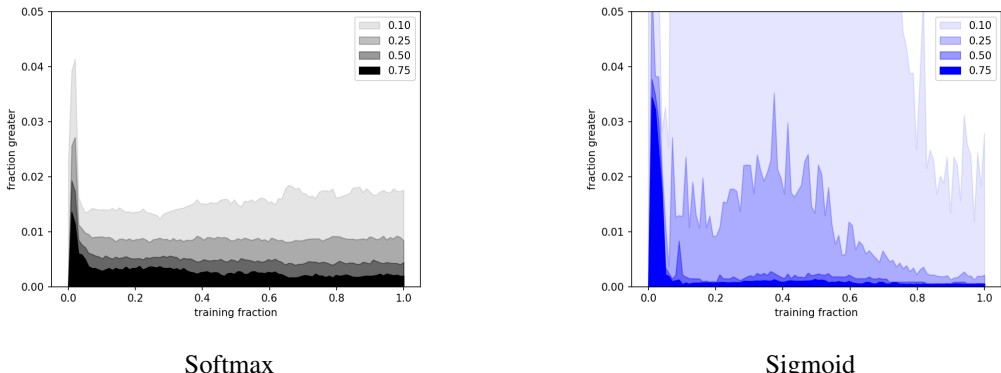

Figure 30: The post-activation attention evolves during training on the $k = 1, n = 40$ summation problem. The model has one head to simplify the visualization. Forty repeated test samples are used.

In Figure 31, metrics on the post-activation attention matrices are used and show comparable behavior in the first half of training. In the second half of training, the SigmoidAttn can be seen to reduce in norm and in sparsity. (see following discussion of Figure 32 for further insights).

In Figure 32, we see post-activation attention values for eight samples at training progresses. The most notable difference between the activations is, that by the end of training, the SigmoidAttn is

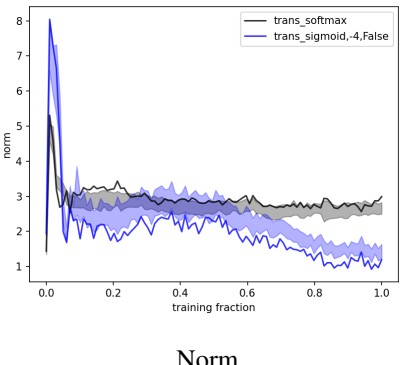
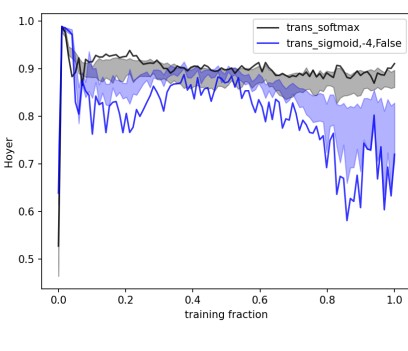

Norm                                 Hoyer Sparsity

Figure 31: Metrics on the post-activation attention evolve during training on the $k = 1, n = 40$ summation problem. The model has one head to simplify the visualization. Quartiles and mean from 40 repeated test samples are shown. On the right, the Hoyer Sparsity (Hurley & Rickard, 2009) is used to measure the change in sparsity as training progresses: $\text{Hoyer} := \left( \sqrt{n} - \frac{\sum_j c_j}{\sqrt{\sum_j c_j^2}} \right) (\sqrt{n} - 1)^{-1}$.

less sparse in the $\mathcal{N}(0, 1)$ self-attention in the upper-left quadrant. We can see that softmax tends to produce sparser values (as it is designed to) while sigmoid controls the magnitude and location of peak attention independently, leading to a less sparse attention at the end of training.

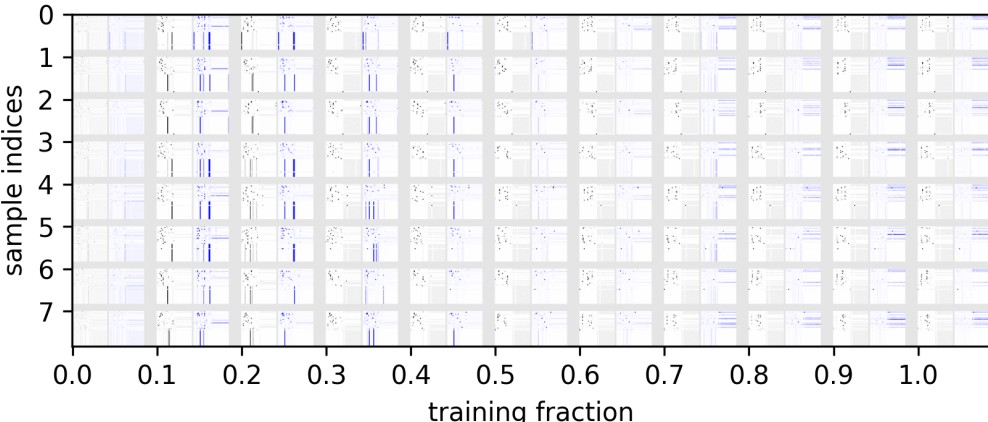

Figure 32: For 8 samples, the post-activation attentions is visualized as training progresses on the $k = 1, n = 40$ summation problem. The model has one head to simplify the visualization. The attention is shown in pairs for each sample with softmax attention is in black and sigmoid is in blue. A $2 \times 2$ block structure is evident in both cases, resulting from each halve of the input containing different information.

### G.5.4  PAIR REPEAT PROBLEM

We define a synthetic task of identifying if the first two symbols in a sequence repeat. The symbols, $s_i$ below come from a fixed vocabulary of size $K$, and the repeat location (when present) is uniformly distributed in the sequence.

$$f(s_0, s_1, s_2, ..., s_N) = \begin{cases} 1, & \text{if } \exists\, n > 1 \mid (s_0, s_1) = (s_n, s_{n+1}), \\ 0 & \text{otherwise} \end{cases}$$

A simple two layer transformer is trained on this problem. The model has an embedding dimension of 160, MLP ratio of 4, QK norm, and layers with eight heads. The results for different model architectures are shown in Figure Figure 33. The maximum input length is 22, $K = 9$, shorter lengths are padding with value $K$, and the training set only contains lengths 14 and 15. A cosine learning rate schedule with 5% linear warmup and a maximum learning rate of 1e-3 is used with the Adam optimizer.

In this result, we see the sigmoid activation has higher data efficiency and similar fall-off in the out of distribution cases. From shorter runs, we estimate that the softmax network would fit the training with 4–5x more data. Our conjecture is that the two layer transformer more easily learns the pair finding task with sigmoid because softmax is biased to focus on single values, though it is unclear why multiple heads are not able to compensate for this proposed cause in the softmax case.

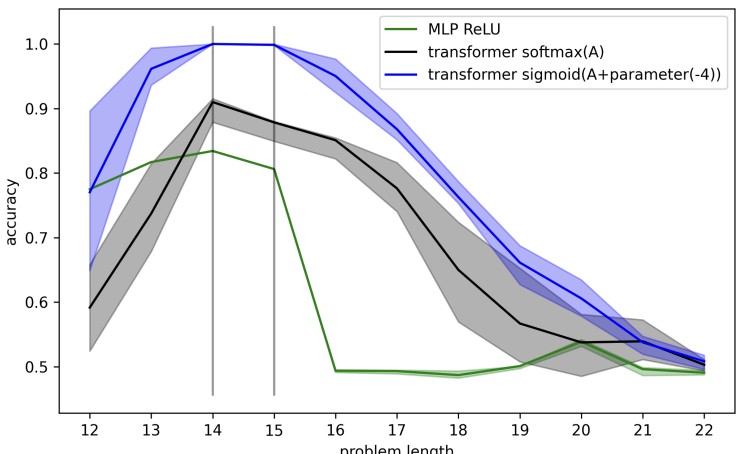

Figure 33: Validation accuracy for out of distribution sequence lengths is shows after 5M samples of training; trained lengths are shown with vertical lines. Quartiles and means are shown from six trials. The MLP has two hidden layers, ReLU activation, and a similar number of parameters. The sigmoid transformer has a learned offset initialized to -4.

### G.6 COMPUTATIONAL COMPLEXITY OF SIGMOID AND SOFTMAX.

Table 16: Forward floating operations per token per attention head. $n_{ctx}$ and $d_{head}$ are the context length and head dimension respectively. $\Delta$ measures the compute difference between sigmoid and softmax. $c$ accounts for causal ($c = (n_{ctx} + 1)/2n_{ctx} \sim 1/2$), or standard ($c = 1$) attention. Typical values from the 1B LLM results are $n_{ctx} = 2048$, $d_{head} = 64$. Sigmoid and softmax share the same number of floating operations (softmax: max-subtraction (2), exponentiation, summation, division; Sigmoid: bias-add, sign-flip, exponentiation, addition, division). Remaining differences are due implementation details, and are subleading ($\sim 1\%$) compared to other attention operations like computing attention logits $L$ (shown below). This analysis precludes hardware aware improvements (Section 4).

|  | $L = QK^T$ | Softmax $(L)$ | $\sigma(L + b)$ | $\Delta$ |
|---|---|---|---|---|
| Expression | $2\,c\,n_{ctx}\,d_{head}$ | $5\,c\,n_{ctx}$ | $5\,c\,n_{ctx}$ | $0$ |

### G.7 PRACTITIONER'S GUIDE

In Table 17 we summarize recommended settings for practitioners who aim to use $\mathrm{SigmoidAttn}$ for training in their respective domains / learning scenarios. While each setting has fully enumerated hyper-parameters listed in the Appendix, we highlight some sane $\mathrm{SigmoidAttn}$ choices below.

**Stabilizing larger models beyond sequence length 2048:** We propose a non-learned scalar bias to mitigate large attention norms with $\mathrm{SigmoidAttn}$ (Appendix E), but observe instabilities

Table 17: Simplified recipe for different domains and tasks with Sigmoid Attention. S/C/A/R refers to using any of SinCos, CAPE, ALiBi or RoPE positional encoding methods.

| DOMAIN | OBJECTIVE | MODEL SIZE | POS EMBED | QK NORM | LAYERSCALE | SIGMOID BIAS | NORM STRATEGY |
|---|---|---|---|---|---|---|---|
| VISION | SUPERVISED | 87M | LEARNABLE | YES | YES | NO | PRE-NORM |
| | BYOL | 87M | LEARNABLE | YES | YES | NO | PRE-NORM |
| | SIMCLR | 87M | SINCOS | YES | YES | NO | PRE-NORM |
| | MAE | 304M | LEARNABLE | YES | YES | YES | PRE-NORM |
| ASR | SUPERVISED (CTC) | 100M-250M | S/C/A/R | YES | YES | YES | PRE-NORM |
| | SUPERVISED (CTC) | 100M-250M | RoPE | NO | YES | NO | HYBRID-NORM |
| AR LANGUAGE | NEXT-TOKEN (<=2K SEQ LEN) | 1B | ALIBI | YES | NO | YES | PRE-NORM |
| | NEXT-TOKEN (<=2K SEQ LEN) | 1B | ALIBI | YES | NO | NO | HYBRID-NORM |
| | NEXT-TOKEN (>2K SEQ LEN) | 1B | ALIBI | YES | NO | YES / NO | HYBRID-NORM |
| | NEXT-TOKEN (>2K SEQ LEN) | 7B | ALIBI | YES | NO | YES / NO | HYBRID-NORM |

at sequence length $n = 4096$ for autoregressive language modeling (Section 5.5). Hybrid-norm (without learnable affine parameters) resolves these instabilities (Table 10). Hybrid-norm differs

| Post-norm | Hybrid-norm |
|---|---|
| $\mathrm{norm}(x + \sigma(\boldsymbol{Q}\boldsymbol{K}^T/\sqrt{d_{qk}})\boldsymbol{V})$ | $\mathrm{x} + \mathrm{norm}(\sigma(\boldsymbol{Q}\boldsymbol{K}^T/\sqrt{d_{qk}})\boldsymbol{V})$ |

from post-norm, which normalizes the combined residual data stream and attention block output. Hybrid-norm is used in models such as Grok-1 (xai-org, 2024) and frameworks such as Praxis (Google, 2024) under the normalization strategy "primer_hybrid". When both $\mathrm{SoftmaxAttn}$ and $\mathrm{SigmoidAttn}$ use hybrid-norm, we observe similar kernel speedup times as highlighted in Section 4. However, with LayerNorm (Lei Ba et al., 2016) only for $\mathrm{SigmoidAttn}$, a token length of 10,000 is needed to achieve a performance gain of $\sim 5.04\%$ for full self-attention and a token length of 1024 is needed to achieve a performance gain of $8.36\%$ for causal self-attention on H100 GPUs. For LayerNorm (with and without affine terms), we summarize approximate regimes for positive throughput gains in Table 18.

| ATTENTION TYPE | FLASHSIGMOID WITH LAYERNORM VERSUS FLASHATTENTION2 COMPARISON | | | |
|---|---|---|---|---|
| | A100 | | H100 | |
| | AFFINE PROJECTION | NO AFFINE PROJECTION | AFFINE PROJECTION | NO AFFINE PROJECTION |
| FULL | 16384 (5.22% ↑) | 12544 (5.08% ↑) | 10000 (4.82% ↑) | 10000 (5.04% ↑) |
| CAUSAL | 12544 (4.18% ↑) | 5184 (4.14% ↑) | 2048 (7.65% ↑) | 1024 (8.36% ↑) |

Table 18: FLASHSIGMOID along with LayerNorm vs. FLASHATTENTION2 on A100 GPUs. Based on benchmarking on a set of randomly sampled tokens from the range $[64, 60000]$, we report the token $T^*$ after which FLASHSIGMOID with normalization consistently outperforms FLASHATTENTION2, along with the total CUDA time speed-up averaged over subsequent tokens ($T > T^*$).

# H CONTRIBUTIONS

All authors contributed to writing this paper, designing the experiments, discussing results at each stage of the project.

**Preliminary work** Preliminary viability of $\mathrm{SigmoidAttn}$ done by Jason Ramapuram.

**Universal Function Approximation** Proof of UFA (Section 3.1 and App. C) sculpted by Federico Danieli.

**Lipschitzness of Sigmoid Attention** Lipschitzness analysis (Section 3.2 and App. D) molded by Pierre Ablin. Empirical regularity analysis (Figure 7) done by Jason Ramapuram.

**FlashSigmoid** Implementation and analysis driven by Eeshan Dhekane in collaboration with Jagrit Digani (Section 4 and App. F).

**Bias Analysis**  Theoretical grounding for bias (Appendix E) done by Amitis Shidani in discussion with Pierre Ablin.

**Language Modeling Results**  All large scale language model pretraining and evaluation (Section 5.5 and App. G.3) driven by Floris Weers.

**Stability Analysis**  QK norm (Figure 9), LayerScale (Figure 8) and bias (Figure 21) ablations crafted by Dan Busbridge using Attention Simulator. Attention Simulator written by Jason Ramapuram and used to validate norm growth (Figures 3 to 6, 19 and 20).

**ASR Results**  All ASR experiments (Section 5.4) and ablations (Appendix G.4.2) are conducted by Tatiana Likhomanenko in discussions with Jason Ramapuram and Zijin Gu. Baseline ASR models code is written by Zijin Gu and Tatiana Likhomanenko. Baseline models are optimized by Zijin Gu to be close to state-of-the-art results.

**Vision Results**  All vision experiments (Sections 5.2 and 5.3) and ablations (Figures 10, 23 and 24 and App. G.2.1) conducted and written by Jason Ramapuram.

**Simple Experiments**  Simple experiments to compare $\mathrm{SigmoidAttn}$ to $\mathrm{SoftmaxAttn}$, including visualizing attention evolution and simple sequence length generalization analysis (Appendix G.5) conducted by Russ Webb.

