# OpenReview forum: "Theory, Analysis, and Best Practices for Sigmoid Self-Attention"
_ICLR.cc/2025/Conference — ICLR 2025 Poster_

### Official Review · Reviewer_q17K · 2024-10-27

**Soundness:** 3
**Presentation:** 3
**Contribution:** 2
**Rating:** 5
**Confidence:** 5

**Summary:**

This paper gives an in-depth analysis of sigmoid attention for use on transformer architectures. It shows theoretically that sigmoid attention based transformers have the universal approximation property and that sigmoid attention has a better local Lipshitz constant than softmax based attention, namely that the former has Lipshitz constant that scales quadratically and that the latter has Lipshitz constant that scales exponentially. The authors conduct a variety of ablations and experiments giving an in depth analysis of when and where sigmoid attention should be used. Furthermore, they develop a hardware aware based implementation on sigmoid attention termed FlashSigmoid that has a much faster inference than the well known FlashAttention2.

**Strengths:**

**Originality:** The paper gives a good in depth analysis of the use of the sigmoid function as an activation for attention and provides an original perspective of how it should be used. I applaud the authors for taking the time to give such a detailed analysis.

**Quality:** The quality of the paper is good clearly showing that the authors have taken the time to give the reader a clear understanding of sigmoid based attention mechanisms in transformer architectures. Furthermore, they undertake a variety of experiments giving the reader insights into how sigmoid attention performs on practical tasks.

**Clarity:** The paper is well written and easy to follow with lots of details provided in the appendix.

**Significance:** I believe the real significance of this paper is in understanding how sigmoid attention compares to softmax attention and when and where it should be used, which I believe is something that is not found in current literature. Furthermore, the authors do develop a hardware aware implementation of sigmoid attention that does much better than the current state of the art for softmax attention.

**Weaknesses:**

**Novelty:** The main issue I have with the paper is its novelty. Although the authors give an in depth analysis of sigmoid attention I don't see it adding value to the community as their results often show that it only does comparable to softmax. The main novelty of the paper I feel is that they develop FlashSigmoid but I feel this is more of an engineering feat and is not enough for a paper to be accepted into ICLR. I applaud the authors for their in depth analysis and their various ablations but I am still questioning whether there is real quality in the paper. Please see my questions below. The proof of UAP is very nice but is essentially a slight change of the proof given by Yun et al., which the authors do say themselves. Furthermore, the authors provide worst case Jacobian bounds which I found very interesting. In section 3.2 they show that the Lipshitz constant of sigmoid attention grows quadratically w.r.t a ball of radius R and that softmax grows exponentially. This suggest that the gradients of softmax attention should explode causing issues with model convergence, yet I don't see this in their experiments. Softmax often performs on par with sigmoid. Thus I don't see how this theoretical analysis transfers into a useful statement for the reader to gain insight into transformers using sigmoid attention over softmax. I think it would be helpful if the authors could tell me in a few sentences what is novel about sigmoid attention apart from FlashSigmoid?

**Experiments:** I found some of the experimental results in the main paper a bit confusing as the authors tend to over do it with ablation after ablation. It would have been much better for the authors to put most of the ablations in the appendix and keep the main experimental results in the main paper. I felt this clouds the other experiments. For example, it might be fine to just put 6 of the graphs in the section 5.1 in the paper and then put the other 6 in the appendix. Similarly, in figure 7 and 8 I feel it would be enough to just put one of them, say layer scale in the main paper, and put qk_norm results in the appendix. This would free up space for you to really talk about in what way you see sigmoid attention yielding a better attention mechanism for the community to use.

Also in figure 2 I can't really see any difference between softmax and sigmoid. In figure 11 the authors compare sigmoid with layer scale and QK norm with other activations and I notice some of the other activations like GeLU and ReLU do on par with sigmoid on the vision tasks. Does this mean for vision tasks activations like ReLU and GeLU are just as good as softmax so therefore sigmoid is not offering any benefit?

My overall feeling is that this paper would make a great journal paper as it gives a very in depth analysis of just one activation and its use for attention based mechanisms. However, I will be willing to change my mind if the authors could answer my questions below.

**Questions:**

1). Putting aside your FlashSigmoid hardware aware attention. Could you explain what is the real benefit of using sigmoid over softmax? Is there any real benefit in terms of training/performance? Could you also clearly explain the disadvantages of using sigmoid attention?

2). I am very interested in your regularity results in section 3.2. I thank you for providing an in depth proof in appendix D. I found this result rather paradoxical though. Your proof shows that the Lipshitz constant of sigmoid attention scales quadratically, see line 159, and it is known that the Lipshitz constant of softmax scales exponentially, see line 162. This implies that sigmoid attention has much better Jacobian regularity and softmax has very bad Jacobian regularity. Wouldn't we see this in training in that it should therefore be very difficult to train transformers with softmax activation as the backpropagated weights would explode as they grow exponentially? However, we don't see this in practice. Most transformers use softmax as an activation and have no real problem with training. For example, most ViTs use softmax and their models can converge. Yet looking at figure 2 for example, I don't see this. Could you comment on this? I wonder if other architectural mechanisms like skip connections is in some sense negating the effect of the exponential scaling of the Lipshitz constant of softmax attention and that is why we still see such transformers being able to train well. Are the other parts of the transformer architecture somehow mitigating the bad Lipshitz scaling that softmax attention has?

3). I noticed in various experiments, see figure 2, table 2, table 3, that sigmoid attention is comparable to softmax attention. However, since sigmoid attention has a Lipshitz constant that has much better regularity than that of softmax (quadratic over exponential) shouldn't we see sigmoid attention doing much better than softmax?

4). You mention in your abstract, see line 023 to 024, that previous attempts at sigmoid attention have not been able to achieve results on par with softmax. However, in the main paper I don't see you comparing to any of those previous attempts. Could you cite those previous attempts for me so I can go and check those papers? Did you compare your methods with those papers? I think this would help your case as it would show just how your methodology does in comparison to those previous attempts you mention in the abstract. The paper is very dense so if this is somewhere in the appendix and I have missed it then I do apologize.

5). In section 5.5 on Autoregressive large language modelling, you mention in line 508 that a slight disparity between sigmoid and softmax is observed at 1B scale with sequence length 4096 which you are able to address using a hybrid-norm by adding an additional normalization layer. I checked appendix G.3 but was not able to follow the explanation. Could you explain why you need extra normalization and in what way it helps the efficiency of sigmoid attention? As the Jacobian of sigmoid attention has quadratic regularity I would expect that much less normalization should be needed for a transformer architecture employing such an attention layer and rather you would need it on the transformer using the softmax based attention. Also, what is the parameter increase by adding an extra normalization layer and does this add any practical disadvantages to sigmoid attention for such tasks?

---

> ### Author Response · Authors · 2024-11-14
> **Response 1/3 to Reviewer q17K: On SigmoidAttn's Novelty, Theoretical Foundations, and Performance Analysis**
>
> Thank you for your thoughtful review of our paper on sigmoid attention. We appreciate your recognition of our theoretical and empirical contributions. We'd like to address your concerns:
>
> * **Novelty & Performance:**
>     * We build upon the proof of Yun et al. and extend contextual mapping to the sigmoid activation (this is much more involved than softmax which tends towards hardmax — elaborated more in the point below).
>     * The gains reported by FlashSigmoid are not only due to it being a hardware-aware implementation, but rather by the fact that Sigmoid itself is a local operation (applied element-wise), as opposed to the non-locality of Softmax (applied row-wise) with direct implications on the parallelisability of the algorithm. Not all FLOPs are created equal.
>     * We empirically analyze practical training of SigmoidAttn and highlight stability issues and ways to mitigate them.
>     * We improve inference throughput of attention by 17% in non IO bound scenarios. Note that 17% inference speedup can result in 17,000 dollars per day in typical language model serving [4]. This can directly reduce energy requirements which has a real world impact. Even without FlashSigmoid, SigmoidAttn is faster than SoftmaxAttn (see JAX results for LM training in Table 3 where no Flash variant is used for either -- at sequence length 4096 we observe $\approx$ 20% speedup).
>     * **Empirically SigmoidAttn outperforms SoftmaxAttn on:**
>         * Language Modeling: 6/9 tasks on 1B n=2048 [truncated summary table below]
>         * Language Modeling:  5/9 tasks at 1B n=4096 [truncated summary table below]
>         * ASR: Sigmoid + RoPE  (“Sigmoid” in Table 2 under “RoPE” row — note that this is SigmoidAttn + QKNorm + b = -ln(n) ) outperforms SoftmaxAtttn on all sequence generalization tasks evaluated on TEDLIUM-v3. Lower WER is better here. We provide a transparent view of various changes here to highlight what works and what does not.
>         * And matches performance in all vision tasks (supervised and self-supervised learning).
> * **“The proof of UAP is very nice but is essentially a slight change of the proof given by Yun et al., which the authors do say themselves”**
>     * We do agree with the reviewer that the proof for UAP in SigmoidAttn largely follows the blueprint of that for SoftmaxAttn, which is why we did ensure to provide due credit to [Yun et al.] in the manuscript. At the same time, we believe it unfair to downplay the technical difficulty associated with our adaptation. Arguably, even for the original proof the main challenge consisted in identifying the proper way of combining selective shift layers to assemble a contextual mapping: SigmoidAttn modifies the action of a selective shift layer non-trivially (compare (7) with (8)), and forced us to re-think the proper way to combine them, as outlined in C2. On top of this, and regardless of the actual complexity of the proof, we believe this contribution positively impacts the quality of the paper, as it allows to highlight the theoretical equivalence of the expressive power of SigmoidAttn to SoftmaxAttn
> * **“In section 3.2 they show that the Lipshitz constant of sigmoid attention grows quadratically w.r.t a ball of radius R and that softmax grows exponentially.” + Q.2** :
>     * We want to clarify this statement: we show that for any input sequence in a ball of radius R, the Lipschitz constant of sigmoid attention is at most $O(R^2)$. On the other hand, [Castin et al. 23] show that there exists an input sequence in a ball of radius R such that the Lipschitz constant of softmax attention grows at least like $R^2 \exp(R^2)$. This result is therefore to be understood as an adversarial case, where a special input sequence is used. It says nothing about the average case behavior that we observe in practice, which is likely much better behaved (otherwise, at the reviewer points out, it would be practically impossible to train transformers). These Lipchitz constants are also of independent interest to study the dynamics of attention [1, 2]. We have clarified this in the revised pdf.
> * **“This implies that sigmoid attention has much better Jacobian regularity and softmax has very bad Jacobian regularity.”**
>     * As stated above, this statement should rather become “This implies that sigmoid attention has much better Jacobian regularity in the worst case than softmax in the worst case”. In practice, we are likely far from these worst cases.
> * **“Also in figure 2 I can't really see any difference between softmax and sigmoid.”**
>     * The goal of this figure is to higlight that on various domains and tasks SigmoidAttn matches SoftmaxAttn, while providing throughput improvements (Figure 1). Faster training for equivalent performance can lead to substantial energy savings as we keep scaling our modern models. We have also improved the clarity of this figure in the manuscript to better highlight this.

---

> > ### Comment · Reviewer_q17K · 2024-11-16
> > **Further questions**
> >
> > Thanks for your response. I didn't mark you down for UAP but I still believe there is not much in the proof you have provided. I did go an actually have a look at the paper by Yun et al. and I still believe your proof is really just an adaptation of theirs. In any case you have been honest about it so I thank you for that.
> >
> > To get back to my original issues with the paper. You have a shown a ton of experiments which show sigmoid doing better in some case and in others comparable. I have no issues with that. It seems you have access to a lot of compute and you are using it wisely which I commend you on. What I am still struggling to find in your work is an explanation as to why sigmoid performs better/comparable to softmax. UAP does not answer this and I hope we can agree on that. UAP shows that a network has the capacity to approximation any function arbitrarily closely but it does not say whether a gradient based algorithm like Adam will always be able to find that solution.
> >
> > Regularity does help in showing that an activation is better than another one. However, this is where I have a problem with your work. You are comparing your regularity to the worst case regularity with softmax and your paper seems to suggest to the reader that because you get a quadratic bound and the worst case for softmax is an exponential bound therefore sigmoid is better than softmax. This is actually not necessarily true. As an extreme example to try and get my point through. Imagine I had a quadratic polynomial and I have a cubic polynomial. Someone then comes and tells me they have a bound on the quadratic polynomial that is exponential. This is indeed a bound but it is a terrible bound. Then someone else comes and tells me that they have a bound for the cubic poly and it is a 10th order polynomial, again it is not a good bound. However, I can't conclude from this that therefore we should expect a cubic to have better regularity than a quadratic. In other words comparisons involving worst case bounds don't lead to conclusions that state "therefore one activation has better regularity than the other". This is a bit of a stretched example but I hope you can see my viewpoint.
> >
> > You have shown your regularity is better than the worst case bound for softmax. The point I am trying to get across to you is that it is a stretch then to conclude that therefore you get better regularity. A worst case bound is exactly what it states "it is worst case" it could well be that softmax has much better regularity than the worst case bound for some applications. And given your experiments it seems that would indeed be the case.
> >
> > It could be that I am missing something here so feel free to tell me I am wrong.
> >
> > Also, when you apply sigmoid to attention. Are you applying it just to the QK matrix as in the case of softmax or do you normalize the sigmoid in some way? This may have been explained somewhere in the paper but given it is so long and dense I have probably missed it. Plus I am going into some of the citations you mention in your work like Yun et al. so with all these papers I might be missing things here and there.

---

> > > ### Author Response · Authors · 2024-11-20
> > > **Clarification on Theory, Implementation, and Empirical Advantages**
> > >
> > > Thank you for your thoughtful feedback. First, regarding the UAP proof - while we acknowledge it builds on Yun et al.'s framework, we respectfully disagree with characterizing it as "just an adaptation". The proof required significant technical work to handle SigmoidAttn's fundamentally different selective shift behavior (compare equations (7) and (8) ), necessitating 4 attention heads versus 2 for SoftmaxAttn and other key modifications. This theoretical foundation importantly establishes equivalent expressive power to SoftmaxAttn, a novel contribution of this work.
> > >
> > > Regarding your insightful point about theoretical bounds not necessarily implying better regularity in practice: first, we want to highlight that we provide empirical support of better regularity through various diverse experiments on different domains and tasks, showing that SigmoidAttn outperforms SoftmaxAttn or on par with it across LM, vision and speech tasks; second, to support these empirical results even further, we conducted an empirical investigation comparing actual Jacobian norms vs theoretical bounds (lines 352-360 and Figure 7 in updated manuscript). The latter results validate your intuition: the empirical bound is far from predicted (worst case) theoretical, however, SigmoidAttn with a proper bias initialization ($b = -\ln(n)$, one of the core contributions of our work), actually achieves a lower Jacobian norm than SoftmaxAttn, both with and without HybridNorm (HybridNorm is only needed for very long sequence lengths like the $n=4096$ language modeling results). This analysis, detailed in Section 5.1, provides preliminary evidence for improved regularity for SigmoidAttn in practice.
> > >
> > > We have also framed our theoretical results more carefully as worst-case guarantees (see contribution (2) and lines 163-166) and clarified that improved Lipschitz regularity doesn't necessarily guarantee more stable training. We've updated portions of Sec3.2 accordingly.
> > >
> > > To directly answer your question about sigmoid application: we apply sigmoid directly to the $QK^T$ matrix scaled by $\sqrt{d}$ plus a bias term, without any additional normalization: $\sigma(QK^T/\sqrt{d} + b)$ (Eq 3 in the paper). The single scalar constant bias term $b = -ln(n)$, where n is the sequence length, helps control output magnitude (as detailed in Appendix E). A key contribution of our work is developing practical setups for training sigmoid attention stably across different domains (see ablations in Figures 3-6 for language modeling, Table 2 for ASR, and Table 11 for comprehensive normalization studies).
> > >
> > > Most importantly, empirically SigmoidAttn delivers strong practical benefits - outperforming SoftmaxAttn in 6/9 tasks at 1B scale (n=2048) while enabling ~20% faster computation with or without FlashSigmoid optimizations (eg: Table 3 shows 20% speedup in JAX implementations without any Flash variant).
> > >
> > > Duplicated lines from lines 352-360 for reference: "Empirical Analysis of Attention Regularity To validate our theoretical analysis (Section 3.2), we measure Jacobian norms of SigmoidAttn and SoftmaxAttn across sequence lengths (Figure 7). Using autograd, we compute exact Jacobian norms for both mechanisms, with and without HybridNorm (Appendices G.3.3 and G.6), comparing them to theoretical bounds (SoftmaxAttn bound omitted as it exceeds scale). Both variants show empirical norms (solid lines) well below their theoretical bounds (dashed lines). With our proposed bias initialization ($b=−ln(n)$), SigmoidAttn achieves lower norms than SoftmaxAttn in both settings, suggesting improved regularity. This aligns with its strong task performance (Section 5). Additionally, HybridNorm (Figure 7, right) reduces norms for both mechanisms compared to baseline (left), highlighting normalization’s role in attention stability at longer sequences."

---

> > > > ### Comment · Reviewer_q17K · 2024-11-26
> > > > **Response to authors**
> > > >
> > > > Thanks for your response. I understand you show empirically that sigmoid attention does outperform or perform comparable to softmax attention. I have no issues with that. However, for ICLR I believe it's not enough to simply give empirical results. A understanding of how those empirical results arise and the mechanics behind them is needed on some level. Your initial position was that sigmoid attention has better regularity and I already explained to you why your argument was incorrect on that front as you were comparing to worse case softmax bounds which is not a logical argument. You have updated your paper to include this reflection which I thank you for doing.
> > > >
> > > > Interestingly you mentioned you have added Jacobian plots in Figure 7 which I have looked at. If you look at that figure you can clearly see softmax empirical has better regularity than the sigmoid bound given in Thm. 3.2 showing that your initial viewpoint that softmax has exponential regularity and sigmoid has quadratic therefore sigmoid is better was completely false. Empirically you can see from your plots softmax has much better regularity than an exponential bound.
> > > >
> > > > I believe that you have still not been able to answer why sigmoid has been able to do better than softmax and your initial argument based on worse case bounds was flawed which I concisely explained to you and now your figure 7 even shows why.
> > > >
> > > > Given that your work has some nice empirical results I have increased my score. Though I don't believe running a mass of experiments and showing something is empirically better than something else is at the level of ICLR and I do believe your work falls short of the ICLR mark. I thank you for the discussions you have had with me and greatly appreciate you taking the time to explain your work.

---

> > > > > ### Author Response · Authors · 2024-11-26
> > > > > **Follow up on Fig 7 and contributions**
> > > > >
> > > > > We sincerely thank the reviewer for their detailed engagement throughout the review process, which has helped improve the manuscript.
> > > > >
> > > > > > If you look at that figure you can clearly see softmax empirical has better regularity than the sigmoid bound given in Thm. 3.2...
> > > > >
> > > > > Figure 7 shows, for the example considered, that **SigmoidAttn achieves better average empirical regularity (lower Jacobian norms) than SoftmaxAttn [the solid blue line is below the solid black line]**. While both empirical regularities are below their theoretical worst-case bounds (as we now clarify in the manuscript), the empirical evidence supports our claims about improved regularity.
> > > > >
> > > > > The **theoretical worst case bound for SoftmaxAttn is omitted from Figure 7 as it is significantly above the the y-scale, and corresponds to worst case bound much worse than that of SigmoidAttn**. The theoretical bounds, though not tight in practice, provide valuable guarantees for worst-case scenarios and for the theoretical analysis of the corresponding mean-field regime [1, 2]. We agree with the reviewer that this is not sufficient to tell the whole story regarding training stability and have appropriately clarified this in the manuscript.
> > > > >
> > > > > > I believe it's not enough to simply give empirical results. A understanding of how those empirical results arise and the mechanics behind them is needed on some level.
> > > > >
> > > > > We appreciate this perspective and highlight that our work provides many theoretical results that aid understanding:
> > > > > - New universal function approximation theorems for SigmoidAttn [ Sec 3.1 ]
> > > > > - Worst-case regularity bounds [ Sec 3.2 ]
> > > > > - Principled analysis of the bias term explaining stability properties [ App E.]
> > > > > - Mathematical foundations for FLASHSIGMOID's improved efficiency [ App F.]
> > > > >
> > > > > While we share the reviewer's interest in theoretical understanding, a complete theoretical explanation for every empirical advantage of SigmoidAttn over SoftmaxAttn may be beyond the scope of a single paper. Can the reviewer let us know of any specific theoretical properties already proven for SoftmaxAttn that would be valuable to also establish for SigmoidAttn? We would be very interested in addressing this in the paper.
> > > > >
> > > > > > Though I don't believe running a mass of experiments and showing something is empirically better than something else is at the level of ICLR
> > > > >
> > > > > We respectfully suggest that our paper offers more than empirical comparisons. Specifically:
> > > > > 1. The theoretical foundations outlined above
> > > > > 2. Significant practical improvements in inference/training efficiency (17% speed-up)
> > > > > 3. Comprehensive empirical validation across multiple domains (vision, speech, language)
> > > > > 4. An implementation that will be open-sourced, enabling immediate practical impact
> > > > >
> > > > > The combination of theoretical guarantees, practical improvements, and broad empirical validation not only makes this work particularly relevant for ICLR but also aligns directly with the standards set out in the ICLR 2025 reviewer guidelines. While we value the reviewer's emphasis on complete theoretical understanding, our paper balances theoretical insights with demonstrated practical utility that is valuable for the ML community.
> > > > >
> > > > > We thank the reviewer again for their careful consideration and welcome specific suggestions for further improving the manuscript.
> > > > >
> > > > >
> > > > > [1]: Geshkovski, Borjan, et al. "A mathematical perspective on transformers." arXiv preprint arXiv:2312.10794 (2023).
> > > > >
> > > > > [2]: Geshkovski, Borjan, et al. "The emergence of clusters in self-attention dynamics." Advances in Neural Information Processing Systems 36 (2024).

---

> > > > > > ### Comment · Reviewer_6HF1 · 2024-11-26
> > > > > > **The score is maintained**
> > > > > >
> > > > > > Thanks for the response. The score is maintained

---

> > > > > > > ### Author Response · Authors · 2024-11-29
> > > > > > > **Possible review thread mixup**
> > > > > > >
> > > > > > > We thank Reviewer 6HF1 for their detailed feedback and taking time taking time to review our paper.
> > > > > > >
> > > > > > > There might be a mixup in that this is the thread for Reviewer q17K.
> > > > > > >
> > > > > > > Nonetheless, please let us know (reviewer 6HF1 and q17K) if there is anything we can do to further improve the paper and your review of it! We believe our work provides strong preliminary evidence on the benefits of SigmoidAttn over SoftmaxAttn, both from a practical and theoretical point of view, and our open source implementation of FlashSigmoid will enable immediate practical impact.

---

> ### Author Response · Authors · 2024-11-14
> **Response 2/3 to Reviewer q17K: On SigmoidAttn's Novelty, Theoretical Foundations, and Performance Analysis**
>
> * **Activation parity for vision tasks:**
>     * To paint a complete and thorough picture we highlight various activation alternatives in vision. Here we find that many activations perform equivalently. However when tried with language we observe a disparity (also observed inHua, Weizhe, et al, see Table 2).
> * **“Putting aside your FlashSigmoid hardware aware attention. Could you explain what is the real benefit of using sigmoid over softmax? Is there any real benefit in terms of training/performance? Could you also clearly explain the disadvantages of using sigmoid attention?“**
>     * Thank you for this thoughtful question about sigmoid versus softmax attention. While the benefits of sigmoid extend beyond our FlashSigmoid implementation, the fundamental advantage lies in sigmoid's local, element-wise nature versus softmax's row-wise operations. This locality enables superior parallelization, particularly valuable for multi-device scenarios where we eliminate one round of node-to-node communication. From a computational perspective, softmax attention requires two reductions along the key/value sequence length (one for softmax normalization, another for V vectors summation), while sigmoid attention needs only one. On modern hardware like the H100, which scales FP32 performance more than bandwidth and warp communication compared to A100, softmax's non-matrix-multiply reduction is less efficient than standard reductions. Importantly, our empirical results (detailed in section 5 and provided as summary tables below) demonstrate that these computational benefits come without compromising model performance, and in some cases even improve it. We believe this work opens valuable directions for exploring alternatives to the conventional softmax monopoly in transformer attention mechanisms.
>     * SigmoidAttn is faster than SoftmaxAttn without Flash applied to either variant. Table 3 uses JAX with XLA without any Flash variant for either formulation. Even here we see a 20% speedup.
> * **Comparison to prior works:**
>     * We compare SigmoidAttn with [5] in Figure 9 (a) — this is $n^{-\alpha}$ without layerscale or qknorm. We also ablate multiplicative sequence length normalization in Appendix G.1.1.
>     * We compare SigmoidAttn with [6] in Figure 9 (c) [this is ReLU$^2$] —  parity is only achieved with QK Norm and LayerScale]. The authors of [6] already have results that highlight that going from softmax → ReLU$^2$ results in a performance degradation (see Table 2 in [6] where perplexity increases). In contrast we show parity in our language modeling results at a much larger scale.
> * **Hybrid-Norm for longer sequence lengths:**
>     * The core issue to mitigate to ensure a high performant SigmoidAttn is that of attention norms. While the local Lipschitz constant shows the worst case scenario for SigmoidAttn being lower than for SoftmaxAttn, this is not informative of the average-case behavior, and we do observe instabilities in practical training. Other reasons beyond the Lipschitz constant that can cause issues:
>         * The Lipschitz constant is calculated on a per-layer basis and does not account for interaction in deep transformer networks with other components like normalization layers and MLPs.
>         * As the sequence length increases there can be a large attention norm which is not properly tempered by the b = -ln (max seq len) approximation. A stronger normalization with hybrid-norm helps mitigate the instabilities at larger sequence lengths. Note that hybrid-norm is applied on the embedding dimension of models (typically much smaller than sequence length for modern models).
>     * To help simplify the exposition we have attached the relevant runs comparing Sigmoid and Softmax Attention below in markdown. Here you can see that SigmoidAttn outperforms SoftmaxAttn on 5/9 tasks at 1B n=4096.
>     * The number of parameters is neglible compared to the rest of the network: for a regular pre-norm transformer block using an embed dim d this increases total parameters by 1/(6d + 3) — so for d=2048 this is 0.0081% extra parameters. We also found that in some situations normalization without learnable affine terms works well (making the extra parameters 0).  There is a slight tradeoff to throughput, however at sequence length 1024 we still observe a 8.36% causal attention throughput improvement.

---

> > ### Author Response · Authors · 2024-11-14
> > **Response 3/3 to Reviewer q17K: On SigmoidAttn's Novelty, Theoretical Foundations, and Performance Analysis**
> >
> > Lower WER% is better in the table below:
> >
> > | 255M ASR Model | test-clean | test-other | Ted 00-10s | Ted 10-20s | Ted 20-30s | Ted 30s+ |
> > |----------------|------------|------------|------------|------------|------------|----------|
> > | Softmax (RoPE) | **2.2** | **5.5** | 12.7 | 10.6 | 12.8 | 9.5 |
> > | Sigmoid (RoPE) | **2.3** | **5.4** | **12.3** | **10.1** | **12.3** | **8.6** |
> >
> >
> > Higher % is better for LM results below:
> >
> > | 1B Lang Model | Seq. Len | ARC Easy (%) | ARC (%) | Hellaswag (%) | Piqa (%) | Sciq (%) | Winogrande (%) | Lambda (%) | TriviaQA (1-shot) (%) | WebQS (1-shot) (%) | AVG (%) |
> > |--------------|----------|--------------|---------|---------------|-----------|----------|----------------|------------|-------------------|-----------------|---------|
> > | Softmax (ALiBi) | 2k | 62.2 | 26.8 | 42.4 | 59.0 | **72.3** | 88.1 | 58.4 | **19.9** | **15.4** | 49.4 |
> > | Sigmoid (ALiBi) | 2k | **62.8** | **28.8** | **42.5** | **59.7** | 70.3 | **88.6** | **59.7** | 19.1 | 13.8 | **49.5** |
> >
> > | 1B Lang Model | Seq. Len | ARC Easy (%) | ARC (%) | Hellaswag (%) | Piqa (%) | Sciq (%) | Winogrande (%) | Lambda (%) | TriviaQA (1-shot) (%) | WebQS (1-shot) (%) | AVG (%) |
> > |--------------|----------|--------------|---------|---------------|-----------|----------|----------------|------------|-------------------|-----------------|---------|
> > | Softmax (ALiBi) | 4k | **62.6** | 27.7 | 42.4 | 58.6 | 71.1 | 88.2 | 58.6 | 18.9 | **14.7** | 49.2 |
> > | Sigmoid (ALiBi) | 4k | 60.5 | 27.3 | 41.3 | 57.8 | 70.5 | 87.0 | 57.6 | 18.9 | 12.6 | 48.2 |
> > | Softmax (ALiBi + H-Norm) | 4k | 61.7 | 26.8 | 43.4 | 59.4 | 70.6 | 88.6 | **60.8** | **20.5** | 12.9 | 49.4 |
> > | Sigmoid (ALiBi + H-Norm) | 4k | 62.4 | **28.9** | **43.5** | **60.8** | **71.3** | **89.6** | 59.2 | 20.2 | 14.3 | **50.0** |
> >
> >
> >
> > [1]: Geshkovski, Borjan, et al. "The emergence of clusters in self-attention dynamics." Advances in Neural Information Processing Systems 36 (2024).
> >
> > [2]: Geshkovski, Borjan, et al. "A mathematical perspective on transformers." arXiv preprint arXiv:2312.10794 (2023).
> >
> > [3] Hua, Weizhe, et al. "Transformer quality in linear time." International conference on machine learning. PMLR, 2022.
> >
> > [4] Patel, D., & Ahmad, A. (2023, February 9). The inference cost of search disruption – large language model cost analysis. semianalysis.com. https://www.semianalysis.com/p/the-inference-cost-of-search-disruption
> >
> > [5] Wortsman, Mitchell, et al. "Replacing softmax with relu in vision transformers." arXiv preprint arXiv:2309.08586 (2023).
> >
> > [6] Hua, Weizhe, et al. "Transformer quality in linear time." International conference on machine learning. PMLR, 2022.

---

### Official Review · Reviewer_sBpo · 2024-10-27

**Soundness:** 3
**Presentation:** 4
**Contribution:** 2
**Rating:** 6
**Confidence:** 4

**Summary:**

This paper revisits sigmoid activations in attention mechanism and point out that the main problem with sigmoid attention is that of large initial attention norms. Then they propose solutions with in-depth theoretical and empirical analysis. Also, authors introduce FLASHSIGMOID, a hardware-aware and memory-efficient implementation yielding 17% inference kernel speed-up over FLASHATTENTION2. Furthermore, authors demonstrate that performance across language, vision, and speech are comparable with softmax attention.

**Strengths:**

**Originality:** This paper provides in-depth mathematical analysis of sigmoid attention specially in Universal Approximation Property and Regularity. Also, authors identify stabilization of large initial attention norms during  the early stages of training as a crucial factor for sigmoid attention.

**Quality:** This paper provides detailed mathematical proofs and very extensive experiments and ablation study on sigmoid attention including supervised image classification, self-supervised image representation learning as well as automatic speech recognition and auto-regresive language modeling. The performance is comparable with softmax attention. Furthermore, I appreciated the implementation of FLASHSIGMOID for efficiency and speedup

**Clarity:** The paper is written very well and easy to follow.

**Significance:** Challenging softmax and understanding how activation works in attention mechanism is appreciated. Softmax attention is computationally cost and authors revisit sigmoid activation function to speedup inference time. I believe this paper will inspire the community to rethink attention mechanism.

**Weaknesses:**

1. In Sec 3.2, the authors state that the Lipschitz constant provides insight into the robustness of the network and the ease of optimizing it. They then present a theorem stating that, in $\mathbb{R}^2$, the local Lipschitz constant of SigmoidAttn is much lower than the worst local Lipschitz constant of SoftmaxAttn. This suggests that sigmoid attention should be easier to train than softmax. However, this is inconsistent with the experiments presented by the authors in Fig. 2, Fig. 3, and Fig. 4.

2. Table 2 shows that sigmoid attention causes unstable training, whereas softmax does not. I believe a theoretical analysis of training stability is necessary.

**Questions:**

1. It would be helpful if the authors could make Fig. 2 clearer.

2. It would be better if the authors conducted experiments on large-scale language models (around 7B parameters).

---

> ### Author Response · Authors · 2024-11-14
> **Response to Reviewer sBpo: Clarifying Theoretical Bounds, Training Stability, and ASR Performance**
>
> Thank you for your detailed review and recognition of our paper's depth and potential impact. We appreciate your positive comments on the presentation and analysis. We'd like to address your concerns and questions:
>
>
> * **“the local Lipschitz constant of SigmoidAttn is much lower than the worst local Lipschitz constant of SoftmaxAttn. This suggests that sigmoid attention should be easier to train than softmax.”**
>     * We thank the reviewer for this insightful remark. First, we want to clarify what these theoretical results mean. Theorem 3.2 shows that the local lipschitz constant of sigmoid attention is $O(R^2)$ for any input sequence that has vectors of norm <= R. On the other hand, the results of [Castin et al. 23] show that there exists a sequence of vectors of norm <= R such that the local lipschitz constant of softmax attention grows like $R^2 \exp(R^2)$. This is a worst case result, and it is not informative of the average case, which might be closer to what happens in practice. Note that we never intend to imply that these results lead to more stable training for sigmoid attention in practice. However, these results regarding the regularity of sigmoid attention are of great interest to theoretically study the dynamics of attention (see e.g. [1, 2]). We have made these points clearer in the revised pdf.
> * **“This suggests that sigmoid attention should be easier to train than softmax. However, this is inconsistent with the experiments presented by the authors in Fig. 2, Fig. 3, and Fig. 4.”**
>     * Fig 2 demonstrates that SigmoidAttn matches SoftmaxAttn on various domains and tasks. We have improved the figure as you suggested, which should enhance the paper presentation.
>     * Fig 3 and Fig 4 are to highlight that a naive application of SigmoidAttn results in instabilities, particularly when naively using SinCos or RoPE embeddings. Note that Figure 5 and Figure 6 highlight that this is completely resolved using (a) ALiBi in Figure 5, (b) RoPE with b=-10 in Figure 6. With Figure 6 (RoPE) the 1:1 comparison is between Sigmoid RoPE and Softmax RoPE. With the bias in place the norm of SigmoidAttn matches that of SoftmaxAttn (RoPE). We also see that Sigmoid slightly outperforms RoPE in terms of the train loss.
> * **“Table 2 shows that sigmoid attention causes unstable training, whereas softmax does not. I believe a theoretical analysis of training stability is necessary.”**
>     * Table 2 provides a thorough ablation of all the components of SigmoidAttn, some of which are indeed unstable. The goal of this table is to be as transparent as possible and ablate what does and does not work with SigmoidAttn.
>     * The suggested approach [ QK Norm with b =-ln(n) ] is called just “Sigmoid” in the table [the default]. Sigmoid + RoPE (“Sigmoid” in the RoPE section) performs better sequence level generalization than SoftmaxAttn on the ASR task that uses TEDLIUM-v3 that evaluates on ever growing sequence lengths [see summary markdown below]. Lower word error rate (WER %) is better.
> * **"It would be helpful if the authors could make Fig. 2 clearer."**
>     * Thanks for the feedback! We have improved the plot in the updated manuscript with a wider line for SigmoidAttn and set the z-level of SoftmaxAttn to be in front as well as apply smoothing to the language modeling train loss. It is now easy to see that SigmoidAttn matches the train loss of SoftmaxAttn on a wide variety of domains and tasks (the main goal of the figure).
> * **"It would be better if the authors conducted experiments on large-scale language models (around 7B parameters)."**
>     * We appreciate the reviewer's suggestion to consider larger models. Our study's primary objective is a methodical comparison between sigmoid and softmax attention, necessitating precise control over all variables. Given that training a single 1B LLM requires about 11,300 A100 40G GPU hours and we train 5 (Tab 3) + 16 (Appendix Table 11) we believe this already presents a fair evaluation of SigmoidAttn vs. SoftmaxAttn [237,300 GPU hours total].
>
>
>
> Lower WER% is better in table below:
>
> | 255M ASR Model | test-clean | test-other | Ted 00-10s | Ted 10-20s | Ted 20-30s | Ted 30s+ |
> |----------------|------------|------------|------------|------------|------------|----------|
> | Softmax (RoPE) | **2.2** | **5.5** | 12.7 | 10.6 | 12.8 | 9.5 |
> | Sigmoid (RoPE) | **2.3** | **5.4** | **12.3** | **10.1** | **12.3** | **8.6** |
>
>
> [1]: Geshkovski, Borjan, et al. "The emergence of clusters in self-attention dynamics." Advances in Neural Information Processing Systems 36 (2024).
>
> [2]: Geshkovski, Borjan, et al. "A mathematical perspective on transformers." arXiv preprint arXiv:2312.10794 (2023).

---

> > ### Comment · Reviewer_sBpo · 2024-11-15
> > **Response to authors**
> >
> > I thank the authors for their detailed explanation. I appreciate that the authors conducted extensive ablation experiments and provided in-depth analysis. Improving the figures makes the paper more presentable. I believe the FlashSigmoid, which improves speed by 17%, is beneficial for both the community and industry. I would like to improve my scores.

---

> > > ### Author Response · Authors · 2024-11-15
> > > **Thank you and welcome further feedback**
> > >
> > > We thank the reviewer for taking time to review our paper and help us improve the work. Please let us know if there is anything we can do to further improve the paper and your review of it!

---

### Official Review · Reviewer_6HF1 · 2024-11-02

**Soundness:** 3
**Presentation:** 3
**Contribution:** 2
**Rating:** 6
**Confidence:** 5

**Summary:**

The paper provides theoretical and empirical analysis for sigmoid attention. The theoretical contribution is the proof that transformers with sigmoid attention are universal function approximators. Authors also empirically linked the training instability of original sigmoid attention to large  attention norm, and proposed  a few options to improve it (switch to Alibi, or adding attention bias. They also did ablation study of the effect of layerScale and QK normalization on training stability. Since sigmoid attention is theoretically more expensive than regular soft-max, authors proposed new Flash-sigmoid  - efficient implementation for GPU. Combining multiple techniques they demonstrated that sigmoid attention can potentialy replace the  classical softmax attention on multiple tasks without loss in accuracy or speed.

**Strengths:**

The paper is very detailed analysis of sigmoid attention vs softmax attention. The sigmoid attention is not very original, but authors did a solid job by exploring its theoretcial foundations and execution solid ablation study to decide if sigmoid attention is viable replacement for  soft-max attention .

The first theoretical contribution - the proof that sigmoid attention can be used as universal approximation to continuous function- is executed well, but has limited novelty, and not very interesting . For me the most interesting theoretical  contribution was the analysis of regularity for sigmoid attention. Btw, I think  the proof of Theorem 2 is missing the key assumption for reqularity is that  ||Wq, Wk"|| and ||Wv|| should be limited. So we have to  make special measure during training to control or normalize  them.

The experimental part (Section 5) is very good. Most interesting parts
1.  observation that training of sigmoid-attention based transformers is very sensitive to initial norm of |sigma (QK)"|v| .
2. demonstarion that combination of layer-scale with  QK norm can help to  stabilize training in some cases
3. very solid ablation study (CV, ASR, LM)

I also liked that authors spent time and effort to rewrite Flash-attention to support Flash-sigmoid and its description in Appendix F. This clearly helped to speed-up whole model (Step-time in Table 3)

The paper is well writen and it is easy to read.

**Weaknesses:**

The main weakness of the paper that it doesn't answer main question  "Why should we switch from original softmax attention to sigmoid attention"?:
- will Sigmoid-Attention  be more stable than original Softmax Attention?
- will it help with long context?
On the positive  side, thanks to flash-sigmoid we can see some speed-up for LLM inference (Table 3)

More details:
The paper started with explanation that original "softmax in SoftmaxAttn is not without limitations. For instance, the softmax function can sometimes lead to a concentration of attention on just a few features (Yang et al., 2018; Ganea et al., 2019), potentially neglecting other informative aspects of the input data." As far as I remember Yang's 2018 paper was about soft-max attention work as low-rank factorization, and Ganea's 2019 was about last soft-max layer.  How this is related to sigmoid-attention?
The paper says that the second reason to explore sigmoid-attention was " applying SoftmaxAttn requires performing a row-wise reduction along the length of the input sequence, which in the case of efficient attention kernels (Dao et al., 2022; Dao, 2023), slows down computations." You will still need gather operation along previous sequence to sum  V vectors scaled with attention weights over previous keys, so it's not clear how using  sigmoid attention can do noticable difference in this case (see e.g.  "Online normalizer calculation for softmax" by M. Milakov, 2018 for detailed analysis)

The first theoretical contribution -- the proof that sigmoid attention can be used as universal approximation to continuous function--  has limited novelty. This is mostly  re-write of original Yun's 2020 proof.  The main  addition is  the proof that sigmloid-based transformers can do contextual mapping (Appendix C2)  . The observation in C1 that sigmoid can approxiamte Heaviside step function when lambda --> infty is  obvious.

The analysis of computational complexity is based on computing flops. But the real complexity should take into consideration memory access (read/write ) instead.

A few comments related to the experimental part:
-  it looks like the proposed remedy ( bias b=-10  only partially helps to control the  increase in the amplitude of |act * V} (graph 6)
- ASR: sigmoid (RoPE with bias) performs badly on long audio ( Table 2)
- LLM: Sigmoid (ALibi) underperform vs Softmax (Alibi) . No results for Sigmoid (RoPE)

**Questions:**

Very solid experimental study, but I am still not convinced with paper conclusion: "Our findings establish sigmoid attention as a viable alternative to [soft-max].
1) is it worth to  switch from original softmax attention to sigmoid attention?
2) will training Sigmoid-Attention be more stable than original Softmax Attention?

---

> ### Author Response · Authors · 2024-11-14
> **Response 1/2 to Reviewer 6HF1: Addressing SigmoidAttn's Advantages, Long-Context Performance, and Technical Clarifications**
>
> We thank you for the positive review and recognition of our paper’s strengths. We appreciate your insightful comments about our work and would like to address your questions and concerns:
>
> * **"Why should we switch from original softmax attention to sigmoid attention?"**
>     * As the scale of modern ML models grows so does the energy consumption.  For example [1] suggests that the cost of inference is in the range of hundreds of thousands of dollars per day. Saving 17% of this can significantly reduce cost and energy, which becomes a more critical concern as model size increases. There are also associated inference speedups.
>     * We also want to highlight that SigmoidAttn does outperform SoftmaxAttn on:
>         * Language: 6/9 tasks on 1B n=2048 [see summary markdown below]
>         * Language: 5/9 tasks at 1B n=4096 [see summary markdown below]
>         * Sigmoid + RoPE performs better sequence level generalization using RoPE on the TEDLIUM-v3 ASR task evaluates the model on ever-growing sequence lengths [see summary markdown below]. This is “Sigmoid” in Table 2 (the default setting is QK Norm + b = -ln(n) that we propose). Here lower WER% is better.
>     * Concurrent work has suggested that Softmax attention produces suboptimal representations [2, 3] and faces issues when working with multi-modal learning scenarios [4]. We hope to explore SigmoidAttn in these contexts in the future.
> * **“will it help with long context? ”**
>     * We do have preliminary evidence that it does well in this regard. See the summary ASR table below w/ TEDLIUMv3 that evalutes on growing sequence lengths (lower WER% is better).
> * **Computational complexity of reductions.**
>     * Mathematically there are 2 reductions in SoftmaxAttn along the seq length (once for the softmax, once for the sum of V vectors) - while SigmoidAttn only has one along the sum of V vectors. In the case of efficient kernel implementations, with SoftmaxAttn, the first reduction for the softmax happens over tensor-core formats while not being a tensor core matrix-multiply-accumulate operation (the second one with V can use the mma) - that leaves it to be fairly inefficient, especially when compared to a regular reduction. This gap is exaggerated on machines like the H100 which scale their fp32 performance more than their bandwidth and warp communication times as compared to the A100.
> * **UFA proof:**
>     * We agree with the reviewer that the UAP proof of sigmoid attention stems from an adaptation of the one for softmax attention, and indeed we highlight this in the manuscript. However we argue that, also in the original case, proving the architecture can represent a contextual mapping indeed represents the main challenge (the remainder in fact reuses established results from UAP of MLPs and approximation of continuous function via piecewise constant ones), and as such is not trivial. Moreover, the softmax operation lends itself well as a building block for building a contextual mapping, thanks to its ability to extract individual values (as it tends to hardmax); sigmoid provides additional challenges in this sense as it can only extract ranges of values (as it tends to heaviside). This makes the proof rather more convoluted, as testified by the lengthiness of C2.
> * **“I think the proof of Theorem 2 is missing the key assumption for reqularity is that ||Wq, Wk"|| and ||Wv|| should be limited."**
>     * The theorem does provide an explicit relationship between the Lipschitzness bound of sigmoid attention and the norms of its parameters ||Wk||, ||Wq||, ||Wv||. To clarify, we will further highlight that the bound on the Lipschitzness constant degrades as the norm of the weight matrices increases.

---

> > ### Comment · Reviewer_6HF1 · 2024-11-24
> > **On non-locality and parallelization of Softmax**
> >
> > The claim that sigmoid is better than softmax, since "Sigmoid itself is a local operation (applied element-wise), as opposed to the non-locality of Softmax (applied row-wise) with direct implications on the parallelisability of the algorithm" and that it requires two passes over sequence is not correct.
> > There is well known parallel algorithm to compute attention based on "Online normalizer calculation for softmax" by Maxim Milakov and Natalia Gimelshein. Their algorithms 1)  (key, value) sequence is partitioned in chunks (k,v)_n and distributed over multiple nodes 2) each node computes A_n partial sum of attention expression + denominator factor s_n 3) sends (A_n, s_n) to the node which combines these pieces into final Attention  score. Completely parallel and requires only one "gather" pass over nodes.

---

> > > ### Author Response · Authors · 2024-11-26
> > > **Clarifying Sigmoid's advantage.**
> > >
> > > Thanks for pointing out the "Online normalizer calculation for softmax" paper; we are indeed aware of this work, which is at the base of the first FlashAttention paper. Our FlashSigmoid implementation, however, builds on the more optimized FlashAttention2 implementation, which has considerable improvements in terms of the reductions and involved operations.
> > >
> > > For softmax attention, we need to compute the maximum of each row block for numerical stability. These "rowmax" variables m_i must be updated per block as shown:
> > >
> > > $m^{(1)} = \text{rowmax}(S^{(1)}) \in \mathbb{R}^{B_r}$
> > >
> > > $\ell^{(1)} = \text{rowsum}(e^{S^{(1)}-m^{(1)}}) \in \mathbb{R}^{B_r}$
> > >
> > > $\tilde{O}^{(1)} = e^{S^{(1)}-m^{(1)}}V^{(1)} \in \mathbb{R}^{B_r \times d}$
> > >
> > > $m^{(2)} = \text{max}(m^{(1)}, \text{rowmax}(S^{(2)})) = m$
> > >
> > > $\ell^{(2)} = e^{m^{(1)}-m^{(2)}}\ell^{(1)} + \text{rowsum}(e^{S^{(2)}-m^{(2)}}) = \text{rowsum}(e^{S^{(1)}-m}) + \text{rowsum}(e^{S^{(2)}-m}) = \ell$
> > >
> > > $\tilde{P}^{(2)} = \text{diag}(\ell^{(2)})^{-1}e^{S^{(2)}-m^{(2)}}$
> > >
> > > $\tilde{O}^{(2)} = \text{diag}(e^{m^{(1)}-m^{(2)}})^{-1}\tilde{O}^{(1)} + e^{S^{(2)}-m^{(2)}}V^{(2)} = e^{S^{(1)}-m}V^{(1)} + e^{S^{(2)}-m}V^{(2)}$
> > >
> > > $O^{(2)} = \text{diag}(\ell^{(2)})^{-1}\tilde{O}^{(2)} = O$
> > >
> > > Additionally, FlashAttention2 requires computing the (log of) normalization factor of softmax for accelerating the forward pass, as you have correctly pointed out. However, note that even this "rowsum" variable l_i must also be computed per block and needs to be correctly updated so that the outputs are consistent. These two variables - rowmax and rowsum - are key to our claim about sigmoid's advantages.
> > >
> > > As detailed in Algorithms 1 and 2 (Appendix F), sigmoid attention eliminates both computations: we don't need rowmax variables since sigmoid is inherently stable, and being point-wise, we don't need rowsum variables. This removes the need for reduction and tracking across blocks entirely, making sigmoid attention more efficient than softmax attention.
> > >
> > > We hope this clarifies our claims. We're happy to provide any additional details if needed. Thank you again for your review!

---

> ### Author Response · Authors · 2024-11-14
> **Response 2/2 to Reviewer 6HF1: Addressing SigmoidAttn's Advantages, Long-Context Performance, and Technical Clarifications**
>
> * **“it looks like the proposed remedy ( bias b=-10 only partially helps to control the increase in the amplitude of |act * V} (graph 6)“**
>     * The correct comparison in Fig 6 is Sigmoid + RoPE b=-10 and Softmax + RoPE. In this case the norms are approximately overlapping. Softmax + ALiBi does have a lower norm, but that model has objectively different learning dynamics.
>     * Sigmoid + RoPE + b=-10 does outperform (in terms of train loss) the Softmax + RoPE baseline in Fig 6. This is also the (practically) infinite data regime, so train loss \approx test loss.
> * **“ASR: sigmoid (RoPE with bias) performs badly on long audio ( Table 2)“**
>     * “Sigmoid” in the table refers to “SigmoidAttn uses LayerScale, QK norm, b=−log n, and no sequence normalization.” (lines 473-474). This model actually outperforms the softmax baseline (see summary ASR table below). We will further clarify this detail in the updated manuscript.
> * **“LLM: Sigmoid (ALibi) underperform vs Softmax (Alibi) . No results for Sigmoid (RoPE)”**
>     * Sigmoid (ALiBi) outperforms Softmax (ALiBi) on 5/9 tasks at sequence length 2048 (see summary table below).
>     * There is a gap at sequence length 4096, but this is resolved via hybrid-norm. We also highlight this below for reference where Sigmoid (ALiBi) + H-Norm outperforms the Softmax (ALiBi) baseline  on 5/8 tasks.
>     * We have many more results (incl Sigmoid (RoPE)) in Appendix G.4 as well as in the ASR results.
> * **Relation to Yang et al., 2018; Ganea et al., 2019:**
>     * We highlight these two works to demonstrate the limited expressivity of Softmax [independent of attention]. The two works above demonstrate this relationship through rank bottlenecking in matrix factorization as the reviewer highlights. There are newer works such as [5] and [4 — empirical] that also highlight softmax as being problematic w.r.t. expressivity — we will include these references in the paper.
>
> Higher % values are between in the two LM tables below:
>
> | 1B Lang Model | Seq. Len | ARC Easy (%) | ARC (%) | Hellaswag (%) | Piqa (%) | Sciq (%) | Winogrande (%) | Lambda (%) | TriviaQA (1-shot) (%) | WebQS (1-shot) (%) | AVG (%) |
> |--------------|----------|--------------|---------|---------------|-----------|----------|----------------|------------|-------------------|-----------------|---------|
> | Softmax (ALiBi) | 2k | 62.2 | 26.8 | 42.4 | 59.0 | **72.3** | 88.1 | 58.4 | **19.9** | **15.4** | 49.4 |
> | Sigmoid (ALiBi) | 2k | **62.8** | **28.8** | **42.5** | **59.7** | 70.3 | **88.6** | **59.7** | 19.1 | 13.8 | **49.5** |
>
> | 1B Lang Model | Seq. Len | ARC Easy (%) | ARC (%) | Hellaswag (%) | Piqa (%) | Sciq (%) | Winogrande (%) | Lambda (%) | TriviaQA (1-shot) (%) | WebQS (1-shot) (%) | AVG (%) |
> |--------------|----------|--------------|---------|---------------|-----------|----------|----------------|------------|-------------------|-----------------|---------|
> | Softmax (ALiBi) | 4k | **62.6** | 27.7 | 42.4 | 58.6 | 71.1 | 88.2 | 58.6 | 18.9 | **14.7** | 49.2 |
> | Sigmoid (ALiBi) | 4k | 60.5 | 27.3 | 41.3 | 57.8 | 70.5 | 87.0 | 57.6 | 18.9 | 12.6 | 48.2 |
> | Softmax (ALiBi + H-Norm) | 4k | 61.7 | 26.8 | 43.4 | 59.4 | 70.6 | 88.6 | **60.8** | **20.5** | 12.9 | 49.4 |
> | Sigmoid (ALiBi + H-Norm) | 4k | 62.4 | **28.9** | **43.5** | **60.8** | **71.3** | **89.6** | 59.2 | 20.2 | 14.3 | **50.0** |
>
> Lower word error rate percent (WER%) is better in the table below:
>
> | 255M ASR Model | test-clean | test-other | Ted 00-10s | Ted 10-20s | Ted 20-30s | Ted 30s+ |
> |----------------|------------|------------|------------|------------|------------|----------|
> | Softmax (RoPE) | **2.2** | **5.5** | 12.7 | 10.6 | 12.8 | 9.5 |
> | Sigmoid (RoPE) | **2.3** | **5.4** | **12.3** | **10.1** | **12.3** | **8.6** |
>
> [1] Patel, D., & Ahmad, A. (2023, February 9). The inference cost of search disruption – large language model cost analysis. semianalysis.com. https://www.semianalysis.com/p/the-inference-cost-of-search-disruption
>
> [2] Xiao, Guangxuan, et al. "Efficient streaming language models with attention sinks." arXiv preprint arXiv:2309.17453 (2023).
>
> [3] Sun, Mingjie, et al. "Massive activations in large language models." arXiv preprint arXiv:2402.17762 (2024).
>
> [4] Team, Chameleon. "Chameleon: Mixed-modal early-fusion foundation models." arXiv preprint arXiv:2405.09818 (2024).
>
> [5] Veličković, Petar, et al. "softmax is not enough (for sharp out-of-distribution)." arXiv preprint arXiv:2410.01104 (2024).

---

### Author Response · Authors · 2024-11-14
**Response to Reviewers: Clarifying SigmoidAttn's Benefits, Theoretical Foundations, and Performance Results**

We thank the reviewers for their time and thoughtful reviews. In addition to each response we want to highlight a few core details that are common across reviewers and address them here:


* **Why SigmoidAttn?**
    * SigmoidAttn improves attention throughput by 17% (in non IO bound scenarios) while maintaining (or improving on) the performance of SoftmaxAttn, this can translate to significant real-world impacts, especially for large-scale deployments. Most ML workloads are dominated by inference. An 17% inference speedup (in non IO bound scenarios) is a substantial energy and cost savings for cloud providers running millions of inferences daily. Common commercially available LLMs use upwards estimates of 100,000 dollars per day [1]. A 17% reduction in this workload is  17,000 dollars per day. This also translates to real world energy reduction.
    * The gains reported by FlashSigmoid are not only due to it being a hardware-aware implementation, but rather by the fact that Sigmoid itself is a local operation (applied element-wise), as opposed to the non-locality of Softmax (applied row-wise) with direct implications on the parallelisability of the algorithm.
    * **Theoretical Contributions:**
        * We provide a proof demonstrating UAP for SigmoidAttn, matching the one in [Yun et al.] for SoftmaxAttn. The adaptation is non-trivial, and highlights the equivalence of the two architectures at the theoretical level.
        * The provided regularity bound provides a worst case analysis which are useful for theoretical analysis and situations like adversarial guarantees.
    * **Empirically SigmoidAttn outperforms SoftmaxAttn on:**
        * Language Modeling: 6/9 tasks at 1B n=2048 [truncated summary table in next message]
        * Language Modeling:  5/9 tasks at 1B n=4096 [truncated summary table in next message]
        * ASR: Sigmoid + RoPE  (“Sigmoid” in Table 2 under “RoPE” row — note that this is SigmoidAttn + QKNorm + b = -ln(n), our recommended setting ) outperforms SoftmaxAtttn on all sequence generalization tasks evaluated on TEDLIUM-v3 [truncated summary table in next message]. Lower WER is better here. In the manuscript we provide a transparent view of various changes to highlight what works and what does not.
        * Matches performance in all vision tasks (supervised and self-supervised learning)

* **Stability / Optimality / Lipschitz constant**
    * SoftmaxAttn has had numerous years of research to help stabilize training and tune hyper-parameters. In this work we don’t tune any of the SigmoidAttn runs and simply keep the exact same hyper-parameters as SoftmaxAttn. The objective here is to show that SigmoidAttn really is a drop in replacement for SoftmaxAttn. Future work will focus more on finding more specific optimizations for SigmoidAttn.
    * We thank the reviewers for highlighting the apparent discrepancy between the theoretical regularity results of SigmoidAttn and its training behaviour in some experiments. To resolve this, we point out that the Lipschitzness bounds shield us from the worst-case scenario (which is worse in SoftmaxAttn: see also [Castin et al], which identifies a failure case of $R^2 \exp(R^2)$ growth for softmax). In general, we can't expect said bound to offer a clear picture of the average behaviour observed in practice, as this is affected by numerous interactions with other components in the architecture (MLPs, normalisation, etc). We will make sure to adapt the paper to adjust the expectations accordingly.

* **Paper improvements:**
    * We have removed the duplicate figure 10 and 11 which were older variants of figure 9 and were mistakenly added.
    * Completed Table 11 to include all relevant variants of SigmoidAttn and SoftmaxAttn [relevent 1:1 summaries in markdown table in next response].
    * Validated speedups with hybrid-norm (Table 18) using optimal layernorm implementation. At 1024 tokens with hybrid-norm we can achieve 8.36% speedup vs. SoftmaxAttn without hybrid-norm on causal attention.
    * Add missing terms in Table 1 for softmax computational complexity (missing counts for max subtraction). Mathematical complexity wise softmax is as expensive as sigmoid however not all FLOPs are created equal.


[1] Patel, D., & Ahmad, A. (2023, February 9). The inference cost of search disruption – large language model cost analysis. semianalysis.com. https://www.semianalysis.com/p/the-inference-cost-of-search-disruption

---

> ### Author Response · Authors · 2024-11-14
> **High level summary results for language modeling (1:1 comparisons)**
>
> Higher % values in the LM tables below are better.
>
> | 1B Lang Model | Seq. Len | ARC Easy (%) | ARC (%) | Hellaswag (%) | Piqa (%) | Sciq (%) | Winogrande (%) | Lambda (%) | TriviaQA (1-shot) (%) | WebQS (1-shot) (%) | AVG (%) |
> |--------------|----------|--------------|---------|---------------|-----------|----------|----------------|------------|-------------------|-----------------|---------|
> | Softmax (ALiBi) | 2k | 62.2 | 26.8 | 42.4 | 59.0 | **72.3** | 88.1 | 58.4 | **19.9** | **15.4** | 49.4 |
> | Sigmoid (ALiBi) | 2k | **62.8** | **28.8** | **42.5** | **59.7** | 70.3 | **88.6** | **59.7** | 19.1 | 13.8 | **49.5** |
>
> | 1B Lang Model | Seq. Len | ARC Easy (%) | ARC (%) | Hellaswag (%) | Piqa (%) | Sciq (%) | Winogrande (%) | Lambda (%) | TriviaQA (1-shot) (%) | WebQS (1-shot) (%) | AVG (%) |
> |--------------|----------|--------------|---------|---------------|-----------|----------|----------------|------------|-------------------|-----------------|---------|
> | Softmax (ALiBi) | 4k | **62.6** | 27.7 | 42.4 | 58.6 | 71.1 | 88.2 | 58.6 | 18.9 | **14.7** | 49.2 |
> | Sigmoid (ALiBi) | 4k | 60.5 | 27.3 | 41.3 | 57.8 | 70.5 | 87.0 | 57.6 | 18.9 | 12.6 | 48.2 |
> | Softmax (ALiBi + H-Norm) | 4k | 61.7 | 26.8 | 43.4 | 59.4 | 70.6 | 88.6 | **60.8** | **20.5** | 12.9 | 49.4 |
> | Sigmoid (ALiBi + H-Norm) | 4k | 62.4 | **28.9** | **43.5** | **60.8** | **71.3** | **89.6** | 59.2 | 20.2 | 14.3 | **50.0** |

---

> ### Author Response · Authors · 2024-11-14
> **High level summary results for automatic speech recognition (1:1 comparisons)**
>
> Lower WER% is better.
>
> | 255M ASR Model | test-clean | test-other | Ted 00-10s | Ted 10-20s | Ted 20-30s | Ted 30s+ |
> |----------------|------------|------------|------------|------------|------------|----------|
> | Softmax (RoPE) | **2.2** | **5.5** | 12.7 | 10.6 | 12.8 | 9.5 |
> | Sigmoid (RoPE) | **2.3** | **5.4** | **12.3** | **10.1** | **12.3** | **8.6** |

---

### Meta-Review · Area_Chair_iiMK · 2024-12-26

**Metareview:**

There is a disagreement between reviewers whether the paper should be accepted. On one had the paper does a good set of experiments establishing the advantages and drawbacks of sigmoid over softmax for attention. On the other hand there is no strong evidence to prefer sigmoid over softmax. Further the discussion of theoretical results for sigmoid only caused more confusion for the reviewers. I believe the paper does a good exploration, even though doesnt definitely address the softmax alternate question, and provides valuable code that others can build on towards giving more concrete answers to this question. Hence I suggest acceptance.

**Additional Comments On Reviewer Discussion:**

Reviewers mainly questioned on the utility of sigmoid over softmax, authors provided more clarification on the experiment results and potential compute savings from not having to do normalization.

---

### Decision · Program_Chairs · 2025-01-22

Accept (Poster)